# Divergent convective outflow in ICON deep convection-permitting and parameterised deep convection simulations

Edward Groot[1,2], Patrick Kuntze[1], Annette Miltenberger[1], and Holger Tost[1]

[1]Institut für Physik der Atmosphäre, Johannes Gutenberg Universität, Johannes-Joachim-Becher-Weg 21, Mainz, Germany
[2]Atmospheric, Oceanic and Planetary Physics, University of Oxford, Sherrington Road, Oxford, UK

**Correspondence:** Edward Groot (large.edward.simulations@gmail.com)

**Abstract.** Upper-tropospheric deep convective outflows during an event on 10th-11th of June 2019 over Central Europe are analysed in ensembles of the operational numerical weather prediction model ICON. Both a parameterised and an explicit representation of deep convective systems are studied. Near-linear response of deep convective outflow strength to net latent heating is found for parameterised convection, while different but physically coherent patterns of outflow variability are found in convection-permitting simulations at 1 km horizontal grid spacing. We investigate if the conceptual model for outflow strength proposed in our previous idealised LES-study is able to explain the variation in outflow strength in real-case scenario. Convective organisation and aggregation induce a non-linear increase in the magnitude of deep convective outflows with increasing net latent heating in convection-permitting simulations, consistent with the conceptual model. However, in contrast to expectations from the conceptual model, a dependence of the outflow strength on the dimensionality of convective overturning (2D versus 3D) cannot be fully corroborated from the real-case simulations.

Our results strongly suggest that the interactions between gravity waves emitted by heating in individual deep convective elements within larger organised convective systems are of prime importance for the representation of divergent outflow strength from organised convection in numerical models.

## 1   Introduction

It is well known that the process of (deep) convective organisation and clustering is an important actor in physics and dynamics of the Earth's atmosphere (e.g. Houze, 2004, 2018; Schumacher et al., 2004). Local heating by clusters of convective clouds can drive a flow that diverges away from the convective heat source in the upper troposphere. The divergent upper-tropospheric flow is accompanied by convergence at low levels. In recent work, Groot and Tost (2023b) have shown that on the one hand geometry and on the other hand clustering, organisation and aggregation of clouds within a convective system strongly affect the intensity of the induced divergent flow in the upper troposphere (when expressed per equivalent unit heating in a column). Idealised Large eddy simulations (LES) show that the amount of divergence differs between infinitely long squall lines and for instance regular multicells at fixed latent heating rates. Differences in strength of mesoscale divergent winds at a fixed (area average) column integrated heating rate as shown in the results of Groot and Tost (2023b) can be explained by variability in storm morphology and convective aggregation and these findings can be synthesised in a conceptual model (Figure 1), which

is introduced later in this introduction. In this work, we aim to identify if convection-permitting and convection parameterised simulations of a real-case scenario display patterns of variability of outflow divergence with storm morphology, convective clustering and aggregation that are consistent with the conceptual model.

Comparing different representations of deep convection (i.e., LES, convection-permitting simulations and deep convection parameterisations) is important as forecast products are increasingly based on high-resolution simulations, while global en-

sembles of weather and climate simulations are currently treating deep convection as a parameterised process (e.g. Bechtold et al., 2014; Ollinaho et al., 2017; Palmer, 2019). Moreover, one could assume that convective aggregation and organisation is less thoroughly represented in convection-parameterised simulations than in convection-permitting simulations (e.g. Done et al., 2006; Keane et al., 2014; Satoh et al., 2019; Lawrence and Salzmann, 2008).

In this work, the state-of-the-art numerical weather prediction (NWP) model ICON (Zängl et al., 2015; Giorgetta et al., 2018)

is analyzed, with a focus on two ensembles with different spatial resolution and therefore representation of convection that cover a single convective event. An extensive methodology will be presented to investigate if the conceptual understanding (Figure 1; Groot and Tost (2023b)) can explain the coupling between dynamics and latent heating in ICON. If successful, the methodology could be useful for applications in further cases and regions around the world. In the following the conceptual model that links storm morphology, convective clustering and convective aggregation to different outflow geometries (accom-

panied by relative differences in the upper tropospheric divergence) is explained, based on Groot and Tost (2023b). After the explanation, the relation of the associated processes of gravity waves and convective organisation is shortly reviewed.

**Conceptual model**

Divergent and convergent flows can be interpreted as results of gravity wave emissions at the location of convective heating

(e.g. Pandya and Durran, 1996; Houze, 2004).

Work fundamental for the interpretation of the conceptual model has been done in the late 1980s and 1990s (Bretherton and Smolarkiewicz, 1989; Nicholls et al., 1991; Mapes, 1993; Pandya et al., 1993; Pandya and Durran, 1996), with some further studies appearing recently (e.g. Bierdel et al., 2017, 2018; Adams-Selin, 2020a, b; Weyn and Durran, 2017). The basic concept is that a warming tendency, representing the latent heating by cumulonimbus clouds as a localised heat source, continuously

creates temperature and pressure, hence density, perturbations. The thereby modified atmospheric states often do not return to a background state immediately, but are maintained for some time: the perturbed state persists because updrafts can last for hours in a well-organised convective system. The outflow pattern is maintained, until the convective heating source ceases. The role of fluctuations in the intensity of a convective system has recently been documented and explained in Adams-Selin (2020a, b).

A continuous stream of upward moving parcels in a convective system results in continuously generated perturbations, leading to gravity wave adjustment within the convective system and in the surrounding atmosphere. The upper branch of the flow following such an adjustment mechanism in the plane perpendicular to a quasi-2D convective system (e.g. a squall line) is the divergent outflow from deep convection, which has been investigated in Nicholls et al. (1991); Pandya and Durran (1996). In Nicholls et al. (1991) explicit expressions for the linear component of the gravity wave response to basic localised heating

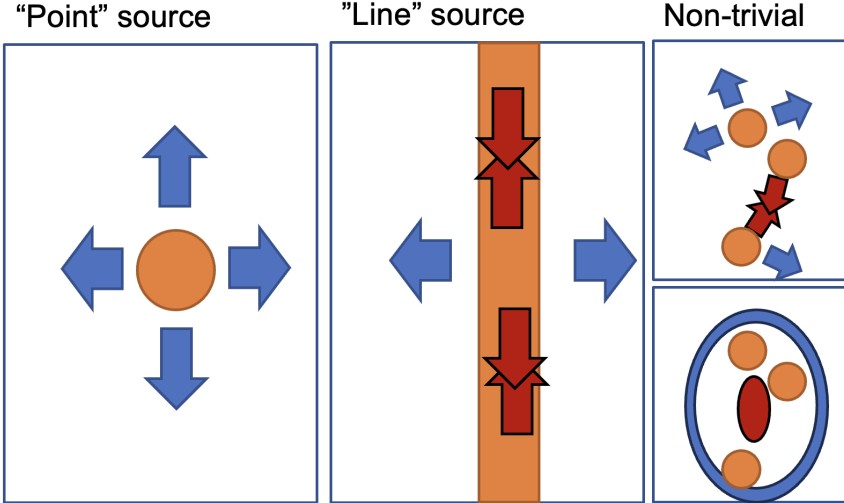

| "Point" source | "Line" source | Non-trivial |

**Figure 1.** Top view of flow resulting from latent heating in convective updrafts (orange shading), as occurring in the upper troposphere. Left: a point source of heating with radially diverging flow. Center: line source of heating associated with squall lines leads to laterally diverging flow (blue arrows), but in the longitudinal direction, individual heating patterns along the squall line compensate: the diverging winds away from individual cells converge (red arrows) and neutralise divergent winds along the convective line. Right (top): complex situation with several updrafts in irregular positions relative to another, (bottom) which leads to a more complex pattern of divergence and convergence zones as outflows collide, simplified in the schematic with an oval orientation of those zones. Reality will be even more complicated. The conceptual understanding is based on Nicholls et al. (1991); Groot and Tost (2023b).

geometries are derived: "point" sources (Figure 1a) and "line" sources (Figure 1b) of latent heating have different outflow intensities. Work by Pandya and Durran (1996) using a more advanced numerical simulation technique shows that the linear model by Nicholls et al. (1991) contains the dominant contributions to the resulting flow (linear approximation) and that the omitted non-linear terms are comparatively less important. Furthermore, Pandya and Durran (1996) argue based on the superposition principle that if a prototype heating pattern is inserted in a model of the atmosphere, the model contains a flow response closely resembling the flow expected from the linear model. Conceptually, this implies that local heat sources associated with updrafts behave as sketched in Figure 1.

From the perspective of Pandya and Durran (1996) any geometric pattern of heating can be seen as a superposition of a pattern of point sources (Figure 1). The most important notion is that radial divergence away from a small updraft ("point source") leads to more divergence at a given latent heating rate than lateral divergence, as associated with idealised line sources (and therefore very elongated squall lines), consistently with findings of Morrison (2016a, b). Nicholls et al. (1991) derived separate expressions for the two geometric flow patterns. In recent work, Groot and Tost (2023b) identified the significance of the differences in idealised LES of different convective systems. In essence, this results in a variability in the amount of upper tropospheric divergence away from a convective system at the cloud top.

Most of the above mentioned works focus on short time scales of several hours, but usually a fraction of a day. On time scales beyond about 6-10 hours, it is thought that the distinction between instantaneous and integrated divergence response is important. The instantaneous divergence is produced by gravity waves, which while propagating on longer time scales (or, equivalently, in a system that rotates faster) are affected by system rotation (Shutts and Gray, 1994; Bierdel et al., 2017, 2018). The heating perturbations that drive the gravity waves are thought to undergo geostrophic adjustment, which would then modify the (balanced) large-scale atmospheric flow. The corresponding length scale of the adjustment process is set by the internal Rossby radius of deformation. In this work however, we solely focus on the instantaneous and local outflow divergence, at the location of convective systems.

The variability in the instantaneous rate of upper tropospheric divergence is governed by the combination of either quasi-two-dimensional or quasi-three-dimensional vertical overturning (or a mixture of those) and, herewith connected, the morphology of organised convective systems. In the current study, we examine whether variability in instantaneous outflow strength from realistic convective systems in NWP may be explained by the same concepts.

### Convective organisation

One of the mechanisms that can organise convection is actively driven by gravity wave dynamics (e.g. Mapes, 1993; Lane and Reeder, 2001; Stechmann and Majda, 2009; Grant et al., 2018; Adams-Selin, 2020a, b). The propagation velocity of gravity waves is inversely proportional to their vertical wavenumber (e.g. Grant et al., 2018). A metric for the vertical wavelength is the count $n$ of wave crests over twice the vertical depth of the troposphere. In this metric waves with $n = 1$, $n = 2$ and $n = 3$ are nearly always or usually fast enough to propagate away from the convective system, in the presence of a typical background flow. These first few vertical modes of propagating gravity waves create regions of preferred upward motion and of preferred positive / negative temperature perturbations in the lower and upper troposphere. The perturbations from the background state thereby increase / reduce the tendency of deceleration of upward moving parcels in certain layers (i.e. convective inhibition (CIN)) and modifying the convective available potential energy (CAPE); see for instance Adams-Selin (2020a), Figure 6). As the gravity waves propagate (also) horizontally, conditions can alternate between more and less favourable conditions for the initiation of deep convection (compared to the background state). Consequently, moving spatial patterns of locations favourable and unfavourable for convective initiation occur around pre-existing convective systems.

As gravity waves simultaneously impact the spatial distribution of convective updrafts and downdrafts and are generated by heating (cooling) signals produced in updrafts and downdrafts, complicated mutual interactions can occur (e.g. Groot and Tost, 2023a, b; Houze, 2004; Adams-Selin, 2020a, b; Grant et al., 2018). These interactions may disturb the simple but typical point sources of divergence and convergence resulting from gravity waves emitted by a local latent heating maximum. The conceptual model proposed by Groot and Tost (2023b) provides an explanation: as divergent winds of convective clouds appear in the form of wave signals, the waves may collide in the upper troposphere. Therefore, convergence may occur locally upon collisions of the wave signals at cloud top levels (Figure 1), which presumably closes the instantaneous upper tropospheric divergence budget over larger scales (Groot and Tost, 2023b; Nicholls et al., 1991). It is thought that this interaction causes a non-linear response of divergence to increasing latent heating.

Other mechanisms, like vertical wind shear, cold pool propagation and (related) moisture convergence also impact the organisation and clustering of convective systems. These mechanisms may interact with the gravity wave dynamics that (co-)organises the convection. In this work it is not of relevance which mechanisms cause convective organisation and aggregation, but it is important to be aware that those factors interact. A comprehensive review of convective organisation and relevant mechanisms is provided by Muller et al. (2022).

Furthermore, convective momentum transport (CMT) may modify upper-tropospheric flow perturbations induced by deep convection (Rodwell et al., 2013). Rodwell et al. (2013) found that mesoscale convective systems over the North-American continent could affect European weather predictability. CMT may not only play a crucial role in the organisation of convective systems, but also in downstream perturbation development. Groot and Tost (2023b) noted that the effect of CMT could be separated into a direct and an indirect effect: firstly, CMT affects divergent flow and associated horizontal acceleration directly, resulting in flow perturbations around convective systems. Secondly, as CMT affects the convective organisation and precipitation rates, this results in an indirect modification of upper-tropospheric flow. A direct effect on instantaneous upper-level divergent outflows was not identified in LES (Groot and Tost, 2023b), possibly due to too weak upper-tropospheric shear.

**Analysis and hypotheses**

The following hypotheses are investigated here:

– The geometry of a convective system is statistically related to the local divergent outflow strength, where updrafts approximately in line produce comparatively less divergent outflow than those that resemble a point source of heating at given heating rates (as in Figure 1a versus b).

– While convective systems aggregate, grow upscale and organise, the precipitation rate tends to increase, but the ratio between instantaneous mass divergence rate and precipitation rate decreases on average (compare Figure 1a to 1c).

– Variability in CMT does not alter the typical (i.e. mean) ratio between instantaneous mass divergence rate and precipitation rate, as found by Groot and Tost (2023b).

Furthermore, it will be investigated whether comparable effects on instantaneous mass divergence variability are represented in ensembles with parameterised moist deep convection. Nevertheless, representation of such effects would not be expected in the first place, since convective organisation is represented in a much more simplified way in convection parameterised simulations than in convection-permitting setups. The methodology will be detailed in Section 3.2.1.

Therefore, we firstly investigate if sub-linear increases of instantaneous mass divergence rate occur at increasing precipitation rates (corresponding to latent heating rates) in convection-permitting and convection parameterised ICON ensembles of a single event. The event is exemplary and will demonstrate whether the methodology is useful, as well as indicate first conclusions on whether the conceptual understanding is likely correct and represents dynamical feedbacks of convective aggregation in state-of-the-art NWP at mid-latitudes. Another aim is to investigate whether patterns resembling line and point sources may be separated, using our proposed methodology. If both of the leading aspects of storm morphology, resulting from line and

point sources of heating on the one hand and convective organisation and clustering on the other hand, are connected to the instantaneous divergence variability, simulation setups are able to represent gravity wave interactions and the impact of storm morphology on instantaneous upper-level divergence patterns. Supposedly this is possible at 1 km grid spacing, but not at 13 km resolution, when convection is parameterised.

In Section 2, the investigated event is characterised in terms of synoptic conditions. In Section 3, the simulations and the data-analysis methods are described. Subsequently, Section 4 illustrates the key parameters derived from the simulations output by discussing their evolution in an exemplary convective system. Then, the instantaneous deep convective outflow strength is compared between convection-parameterised and convection-permitting ICON in Section 5.2. In Section 6, we analyze the convection-permitting ICON by characterizing the relation between key parameters and the strength of divergent deep-convective outflows. Thereby, the representation of deep convection during the event is investigated following the hypotheses formulated in this introduction.

Afterwards, we reflect on the results and their coherence in the discussion of Section 7, as well as their implications. This is followed by the main conclusions (Section 8).

## 2 Synoptic conditions of the case study

The organised convection over Central Europe on June 10th and 11th is notorious for the Munich Hail Storm (Wilhelm et al., 2021). An upper-tropospheric low pressure system was located over Western France on June 10th 2019 (Figure 2; grey isolines with geopotential height patterns), with a southerly flow advecting warm, moist air northward over Central-Europe (high $\theta_e$, red). The associated pattern with cold air west of the upper low pressure system led to strong baroclinicity over France, the Alps and (later) Germany. Cold surface air creeping northeastward directly ahead of the collocated cold front supported the initiation of strong convective systems. These systems are present in nearly all simulations, albeit at slightly different locations than in reality, including east of the front in the region of warm near-surface air. Storms generally appeared further to the west in the convection-permitting ensemble and even more so in the parameterised set-up than in reality.

After all, several systems with mesoscale convective activity developed over Germany and the Alps during the afternoon and evening, which were co-located within the parameterised deep convection ensemble. Similarly, convection was relatively active in convection-permitting simulations over Southern-Germany in the (late) afternoon of the 10th of June (e.g. Figure 7a; observed convective systems are also shown there). Well organised convection occurred over regions with strong relief in the southwest. In the east initially surface-based convection occurred in the late afternoon to early evening. The strong south to southwesterly upper-level flow helped to organise convection in convection-permitting simulations to a varying degree: a few convective systems in the east of Southern-Germany developed squall line-like structures. On the contrary, other structures only organised into smaller multicells. This mixture may be very suitable for assessments of instantaneous divergent outflow rates from deep convection, since idealised LES simulations suggest that convective organisation, geometry and aggregation may be crucial aspects for the outflow rate. These aspects determine the normalised outflow strength with respect to net latent heating (Groot and Tost, 2023b). A more detailed discussion of the synoptic configuration and actual convective evolution around this

**Figure 2.** Equivalent potential temperature at 600 hPa (blue-white-red), isotachs at 250 hPa (30 to 60+ m/s at 5 m/s intervals, transparent colors) and geometric height of the 250 hPa surface at ca. 11 km height and with 50 m intervals forecasted for 22 UTC on June 10th over Western Europe.

event is provided in Wilhelm et al. (2021).

## 3    Methods

### 3.1    Model set-up

#### 3.1.1    Domains, grids and parameterisations

This study investigates numerical simulations with ICON 2.6 (Zängl et al., 2015; Giorgetta et al., 2018), which is developed and operated by the German Weather Service and Max Planck Institute for Meteorology. Simulations have been conducted and analysed in the following configurations:

   – Global simulations, with a nest over Europe ("PAR")

– Convection-permitting simulations over Southern-Germany ("PER") using the local area mode (LAM)

The PAR-simulations have been initiated at 12 UTC on June 10th 2019, whereas PER-simulations over Southern Germany have been initiated at 03 UTC on the same day (Figure S1 in the Supplement). For details on simulation settings, see Table 1. We refer to Prill et al. (2020) for the mostly similar default parameterisation settings.

**Table 1.** Simulation settings within the three domains.

| Domain | Global domain | European Nest | LAM, Southern Germany |
|---|---|---|---|
| **Model version** | 2.6.0 | | 2.6.2.2 |
| **Grid spacing (km)** | 26 (R03B06) | 13 (R03B07) | 1 (R05B09) |
| **Time step (s)** | 100 | 50 | 10 |
| **Domain top altitude (km)** | 75 | | 22.5 |
| Number of vertical levels (-) | 90 | | 90 |
| **Deep convection parameterisation** | Tiedtke (1989), Bechtold et al. (2014) | | None |
| Time step deep convection (s) and subgrid orography | 1200 | 600 | |
| Time step gravity wave drag (s) | 1200 | | |
| **Microphysics parameterisation** | 1M (Seifert, 2008) | | 2M (Seifert and Beheng, 2006) |
| **Radiation parameterisation** | Ritter-Geleyn[1] | | RRTMG |
| Time step radiation (s) | 1800 | | 600 |
| Grid spacing radiation (km) | 52 | 26 | 1 |
| **Rayleigh damping height (km)** | 22 | | 12.5 |
| **Initial conditions** | DWD analysis | | KENDA, provided by Matsunobu et al. (2022) |
| Initial condition time | 12 UTC | | 03 UTC |
| Initial condition set | Deterministic | | 20 member ensemble |
| Ensemble perturbations | Surface dataset ($n = 6$; 2015-2018) | | Initial conditions |
| **Boundary conditions** | None (two-way nested) | | ICON EU ensemble forecasts (20 km) |
| **Additional perturbed simulations (number)** | Rescaled: - Latent heating $\pm5\%,\pm10\%,\pm20\%$ (6) - CMT tendencies (none, $\pm50\%$) (3) Deep convection scheme: - No parameterisation (6, ensemble perturbations) - Adjusted calling frequency (2) | | None |
| **Output time step (min)** | None | 10 | 5 |
| **Total integration time (hours)** | 33 | | 16 |

### 3.1.2 Ensemble and perturbation settings

Ensembles have been used with the aim to sample an unspecific form of background convective variability within a similar large-scale flow configuration. To further sample the variability of the PAR-set-up, additional experiments with adjustments following Groot and Tost (2023b) have been done (Table 1). Global nested simulations have been perturbed with alternative surface tile datasets (outdated, spanning various dates over 2015-2019), whereas the 20 member ICON-PER initial condition ensemble closely resembles the operational ICON D2-ensemble of DWD.

The combined variability imposed by selecting various convective systems over a time range and through the dimension of ensemble members allows us to study the characteristics of convective variability in a precipitation-conditional framework.

The results presented will mostly be focused on the comparison of the PAR and PER ensemble and on the PER ensemble itself.

### 3.2 Extracting convective system properties in ICON simulations

For a fair comparison, the divergence in convection-permitting simulations are low-pass filtered, whereby the variability in the wind field at scales up to 45 km is removed using a discrete cosine transform. Thereby, the convection-permitting and convection-parameterised simulations obtain roughly the same effective resolution in the divergence field. Therefore the divergent outflows are well intercomparable and there is no problem of small-scale divergence patterns in convection-permitting simulations (lacking in the parameterised configuration). The filtering step assures that the box integrations that we carry out

are applied to datasets with very similar truncation scales.

Extraction of properties of individual convective systems (shape, area, etc) can be achieved in the PER-simulations. On the contrary, parameterised treatment does not lend itself very well to such an extraction procedure, because it assumes that a statistically averaged effect of convection over larger scales exists and is represented (e.g. Done et al., 2006). Furthermore, the geometry of convective heating cannot be assumed to be well represented, which is a problem mathematically similar to the

coastline problem: an island of 100 square km in a 10 km grid spacing always has a coastline of 40 km, but as soon as the coast is represented more accurately, the coastline can have any other length (larger than the minimum of $20\sqrt{\pi}$ km). In other words, the potentially complicated geometry of gravity wave sources, hence their emission, has to be under-resolved. Therefore, any description of (sub-grid) variability induced by convective cells and convective organisation is represented substantially less accurately than in convection-permitting simulations with finer grid (if at all represented), the latter of which in the analogy of

the coastline problem allows for a range of "inlets" and "bays" while the former does not. The extraction procedure of organised convective systems and associated metrics from the PER-simulations is described in section 3.2.1, followed by discussion of metrics from PAR in 3.2.2.

### 3.2.1 Convection-permitting simulations "PER"

In order to single out the expected outflow regime (2D-like or 3D-like), the dataset with properties of convective systems

must be able to describe the degree of convective clustering, orientation and the relative state of elongation of convective

systems in time and space. These factors have been found to determine the relative (instantaneous) magnitude of outflow from deep convection (Groot and Tost, 2023b, and Figure 1). Parameters describing the elongation and state of aggregation for any convective system are estimated with an ellipse fitting algorithm which has been designed for this purpose (Figure 3, blue boxes on the left). In parallel, a moving box is initiated to track a convective system (Figure 3, red arrows towards the red box). The box conserves a moving integration volume, relative to the convective system's main updraft, over which instantaneous divergence rate and instantaneous precipitation rate are integrated. After the independent boxes have been determined, the following steps lead to a dataset of convective systems and ellipse parameters:

1. A moving box is initiated to track each convective system in a simulation

2. Ellipse fitting (blue boxes no. 1-4 in Figure 3)

3. Validation and ordering of obtained ellipse parameters (blue boxes no. 5-6 of Figure 3)

4. Matching between ellipse parameters and moving box diagnostics, as obtained from a specific convective system and that specific simulation (brown arrow and first brown box at the bottom of Figure 3)

5. Final check of the matched records (second brown arrow at the bottom of Figure 3)

Before the ellipse fitting and validation procedure is detailed, the following two paragraphs describe the procedure to derive box diagnostics.

**Box volumes**

The box (step in red) is used for integration of precipitation and divergence over a horizontal subspace that is constant in time (with respect to the moving box centre). The convective systems propagate with relatively constant velocity north- or north-eastward and only 1-3 systems have been tracked in each simulation (see also Figure 7a). Manually defined boxes moving at a constant velocity could therefore be used to define integration outlines.

For each box and time step the following variables are calculated: Firstly, the strength of convective momentum transport (CMT) is computed to determine whether and how this acceleration (deceleration) affects the upper-tropospheric divergence. The estimate of CMT is based on the cross-correlation products of flow deviation vectors ($u' = u - u_{mean}$, $v'$, $w'$) from the domain mean. Separate estimates of the meridional and zonal correlations with vertical velocity representing convective momentum transport fluxes are computed at model level 25, to estimate the vertically integrated CMT acceleration up to this vertical level (located at 315 hPa or about 9 km altitude). This level is selected, because the eddy flux in the troposphere has a maximum at or near this level during the studied event. The box mean values of $u'w'$ and $v'w'$ represent the vertical integral of CMT acceleration over all levels below the selected level. Both CMT and divergence rate are normalised with respect to mean surface precipitation rate, i.e., proxies of box mean latent heating (from here on called $C$ for normalization of CMT and $D$ for normalization of instantaneous mass divergence rate), to investigate the connection between anomalies in both quantities (conditioned on precipitation rate) in a more robust way. Secondly, the mean precipitation intensity and the filtered mean

divergence rate (wavelengths $> 45$ km in both horizontal directions; top parts of Figure 3) is computed. These three quantities can only be computed if the box is fully contained within the simulation domain.

The moving boxes are initiated and then track the systems independently from the ellipse fitting procedure, because merging of ellipses occurs frequently in the ellipse dataset. In case of a merging event, ellipse parameters will weakly vary in time, but the spatial integration mask of the moving box should not change accordingly. If ellipse parameters vary strongly, the ellipses cannot be validated. The signal of instantaneous divergence rate and precipitation rate within a box should predominantly be affected by the main, central convective system within the box and only be weakly affected by small/shallower neighbouring

cells that develop from time to time around some of the systems.

**Ellipse fitting and constraining the final PER-dataset**

Ellipse fitting and verification are used to quantify the geometry of convective systems, in line with Grant et al. (2020). However, a new methodology tailored at our hypotheses is developed. The ellipse fitting is applied to any area larger than about

400 km$^2$ with an average precipitation rate over five minutes exceeding 10 mm/h in PER-simulations (blue box no. 1). Before fitting the ellipses, a binary representation of convective precipitation is smoothed spatially ($\sqrt{r}$ dependence) over a 20 km radius (blue boxes no. 2-3 in Figure 3). A module named CV2 (as part of OpenCV, utilized in 2022) is used for the ellipse fitting procedure, and for the technical details, we refer to the code (Groot and Kuntze, 2023). Subsequently, after fitting ellipses, an initial validation procedure assesses the stability of the ellipse parameters over an one hour window (next boxes, no.

4-5, Fig. 3). Short lasting very strong fluctuations are filtered out. Only fluctuations that match any prior and successive record within one hour are kept (Groot and Kuntze, 2023). The result is a track of each convective system, which may contain one or more gaps of one or several time steps (blue box 6, Figure 3).

Subsequently, after integrating the instantaneous precipitation rate, divergence rate and convective momentum transport spatially (red arrows and boxes), additional validation measures check the distance between an ellipse center (set to be $< 20$ km)

and the corresponding box center (first brown arrow at the bottom Figure 3). A list of ellipse parameters extracted in the procedure is provided Table A1 (Appendix).

Finally, the ellipse characteristics of the ellipses contained within each box (elongation $A$: length ratio between two ellipse axes; $O$ orientation; area of the ellipses) are matched with the integrated divergence, CMT and precipitation rate as computed over the moving box volume. It should be noted that the ellipse parameters and the corresponding box-integrated diagnostics

are matched within one simulation and for one specific convective system within that simulation. An example of the path of a moving box and (contained) ellipses with corresponding ellipse parameters for one convective system is provided in Section 4.

**Dataset**

The ellipse dataset fulfilling all conditions of quality control contains 456 records, in which the time evolution of 22 of a total

of 28 convective systems is represented (following the validation procedure). This dataset is the basic dataset for the assessment in Sections 5.2 and 6. With a slightly weaker box-center-to-ellipse-center distance criterion, a second dataset of 866 records is obtained. For this larger dataset, the distance criterion was set individually for each convective system (based on characteristics

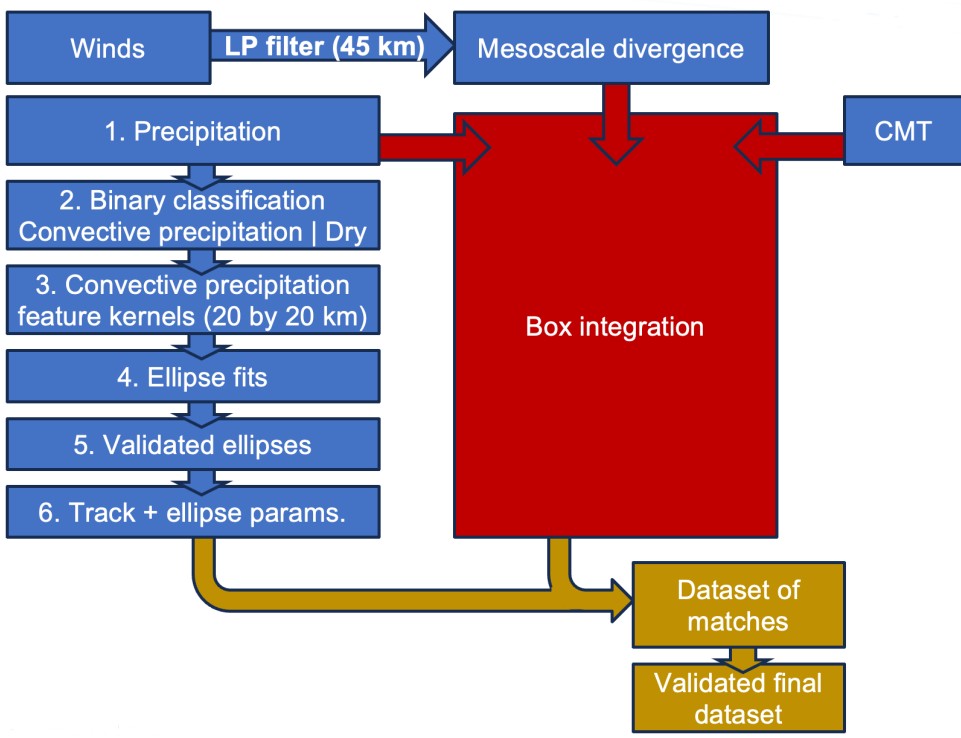

**Figure 3.** Processing of the raw ICON PER simulation data to obtain the dataset used in the analysis of section 4 and onward. The input fields displayed in the four blue boxes at the top. The pre-processing steps, numbered 2-6, are also displayed in blue boxes. They consist of three parallel streams of data, derived from precipitation rate, eddy flux of CMT and filtered instantaneous mesoscale divergence rate and serve as inputs to the box area/volume integration (which itself is displayed in red). These streams merge into a dataset of the instantaneous precipitation rate, divergence rate and momentum transport (CMT) and ellipse parameters. A match occurs based on center distance between ellipse centre and box centre at a given time. The ellipse records and box records can merge in the first brown arrow's step if the distance is under a threshold (main processing step 2), followed by another final verification step of post-processing before a dataset is fixed for final use in this work. These steps are further detailed in section 3.2.1.

The workflow as documented here is carried out by scripts published in Groot and Kuntze (2023).

such as box size) or replaced with an ellipse area criterion. In the larger dataset, all 28 convective systems are present. In a few cases, duplicates fulfill all validation criteria based on area and centre location of the ellipse at one specific time stamp.
Duplicates have manually been selected before finalising both datasets: five additional duplicates were in the dataset of 866 (+5); two occurred in the dataset of 456 (+2) records.

### 3.2.2 Convection parameterising simulations "PAR"

Computationally feasible NWP resolutions require the application of a parameterisation to represent deep convection, although mostly just in current global models. Only recently, global convection-permitting simulations have been utilised for research purposes (e.g. Judt, 2020) (and even in such a setup, shallow convection is often parameterised).

The philosophy behind the representation of deep convection is and has been generally different between parameterised convection and convection-permitting simulations approaches - for simulations with parameterised deep convection:

– Convective cells are not advected with the background flow, but have their full life cycle within a cell: there is a split between larger-scale, explicitly represented dynamics and the parameterisations of sub-grid-scale motion (including deep convection) in each grid cell (Lawrence and Salzmann, 2008; Prill et al., 2020)

– An equilibrium assumption is done (e.g. Done et al., 2006; Becker et al., 2021) , where (deep) convection represents the adjustment mechanism of the atmosphere to the presence of static instability. Adjustment occurs under the condition that convection can be triggered. However, grid cells in numerical models are often so small nowadays that the equilibrium, between convective forcing and the adjustment on a separate scale, is questionable.

– The temporal resolution of the full life cycle of convection within an individual cell is either represented within a full time step (typically in climate models), or the adjustment process and reduction of CAPE takes place over several consecutive time steps.

As a consequence, the representation of deep convection by parameterisation does not only tend to smoothen precipitation through its coarser resolution, but also through underestimated spatial and temporal variability (Keane et al., 2014) . Even though there are small differences between the assumptions applicable to different convection schemes (Arakawa, 2004), the assumptions outlined above and the comparatively large grid size imply that convective organisation is weakly represented in simulation configurations with parameterised deep convection and weakly coupled to the engine of numerical models (the dynamical cores); we could say it is clearly underrepresented. Therefore, categorisation by convective organisation is weakly justifiable (if at all) (see also Satoh et al., 2019). Consequently, an application of a complex tracking algorithm following parameterised deep convective systems is not suitable. A statistical sample of convective cells technically regenerates anyway, while corresponding precipitation moves together with conditionally unstable or lifted air masses.

Furthermore, the LAM-domain is small (400 by 500 km), whereas the parameterised convection simulations cover most parts of Europe with a grid spacing of 13 km. A typical (mesoscale) convective system is easily contained within a box of several to tens of grid cells in each horizontal direction for ICON-PAR, which means the system starts to get resolved if it grows sufficiently large (e.g. Skamarock, 2004). However, can still be assumed to be affected by the regenerative, and other, parameterisation assumption, which likely induces biases in the coupling between parameterisation and dynamics. For the comparison, three static boxes are chosen and compared among the PAR-simulations. These boxes are designed such that the dominant precipitation rate and divergence rate signals associated with convective systems fall within the boxes. Three very different deep convective systems are systematically compared across six ensemble members.

  **4    Example of a track in ICON-PER**

The track of one of the two convective systems in ensemble member 14 of the PER simulations is illustrated in Figure 4a. The box centre is indicated as a red line, with bi-hourly markers along the way. The first snapshot at 12:30 UTC shows that the ellipse algorithm detects an aggregated convective system at the edge of the box. This large system does not fully fall into the box. The validation procedure automatically reports a failure (represented by an X) because of a too large distance between the box centre and the ellipse centre (which is surrounded by the convective system).

During the next hours, small convective cells develop near the center of the box (14:30 UTC). The easternmost system obtains a surrounding ellipse located within close range of the box centre. Another one to the west also obtains an ellipse, but the distance to the box centre is larger. Therefore the latter match is rejected.

Two hours later, again two matches are found: one very near the box centre and one to the north of the centre, but within the box. The larger central one matches through the distance rule, but the northern one gets rejected.

At 18:30 UTC, an elongated convective system develops in association with the earlier central system (14:30, 16:30 UTC). Still sitting close to the box centre, it is the only ellipse within the box.

In Figure 4b the evolution of the ellipses over short time intervals is illustrated. The differently colored precipitation and box features move to the northeast slowly. However, the ellipses undergo various changes, which is associated with a slight convective reorganisation. The reorganisation is induced by new cell formation in a close proximity of the older system. The northeastern feature is detected throughout, but blue crosses demonstrate that the match is initially rejected. The box is slowly closing in on the system, as revealed by the possible match (square marker) at the sixth and last time step. For the southwestern system, the initial system (larger purple ellipse associated with it) breaks up into smaller pieces for two successive time steps and eventually disappears. One of the ellipses of the southwestern system (blue circle) matches with the box at one instance (green), when the ellipse is closest to the box centre.

However, the northeastern system matches at just one instance: the last time step. This match is only valid for the larger dataset with relaxed conditions. This illustrates how convective (re-)initiation and small displacements can affect the ellipse parameters. Corresponding jumps in evolution of ellipse parameters are filtered out. The wobbly interval is indicated by the dark purple rectangle at 17:30-18:00 UTC. Most ellipses in this interval are rejected due to wobbly ellipse parameters, but some are retained during the interval. A temporary shrinking in the axes lengths is seen (without consequent rejection in the validation), due to the stability criteria and interpolation from any prior and successive records within an hour. Another jump within the time window is seen in the offset parameters (Figure 5). Nonetheless, the evolution of ellipse parameters is mostly smooth over the five hours. This evolution illustrates that the regenerating systems can successfully be detected, covering their temporal evolution.

The evolution of upper-tropospheric divergence, CMT and precipitation rate over the moving box is found in the supplement (Figure S3). Around 13 UTC no records of the system are validated: the validation criteria haven not been fulfilled (green solid outline in Figure 5).

Between 14 and 15 UTC two convective systems have been matched with the box (Figure 4). One is travelling at a distance of

about 20 grid cells from the box centre and the other at about 4-9 cells (10-20 km).

The distance between the centre of an ellipse and the associated convective box centre is maximum 9 grid cell distances (20 km) for the strict dataset of 456 records (purple line versus pink solid rectangle in Figure 5).

## 5    Intercomparison of divergent convective outflow rates in ICON-PER and ICON-PAR

The representation and variability of instantaneous convective outflow rates in ICON-PER and ICON-PAR ensembles is compared here. In particular, the mean mass divergence rate over moving boxes and in corresponding areas of persistent thunder-

storm activity is investigated. First, the spatial-temporal characteristics of instantaneous divergent outflow rates are broadly assessed for the selected systems in both ICON-PER and ICON-PAR. This provides a basis for the quantitative intercomparison of the divergent outflow rates between both configurations, for which we condition on the precipitation rate (equivalent to net latent heating rate). Case-related information on the convective organisation and plausible assumptions on the outflow characteristics are used to further characterise the dataset.

After this comparison, Section 6 will analyse the upper tropospheric mass divergence rate versus precipitation rate and the corresponding ellipse parameters (ICON-PER only; this is motivated in the current section), which is a verification of the conceptual understanding presented in the introduction (e.g. Figure 1).

### 5.1    Convective systems and associated patterns in instantaneous divergence (variability)

The time evolution of mean horizontal divergence rate over the moving boxes in ICON-PER is displayed in Figure 6. Boxes

without deep convective activity have been omitted. Furthermore, the boundary between mass divergence and mass convergence has also been highlighted. The evolution of upper and lower quartiles of the pressure level of this boundary has been marked by a grey dashed line, while the median is added in black. Deep convective inflow (mass convergence) predominantly occurs in the boundary layer (bottom boundary up to about 800 hPa) initially. Subsequently, the convection tends to become elevated (16-19 UTC) in ICON-PER: dominant inflow levels lift to about 600-800 hPa. Furthermore, weaker inflow, and en-

trainment, typically occurs up to about 450 hPa. Above 400-450 hPa (roughly the boundary between net convergence and divergence), the main outflow region extends upward. A strong vertical gradient in the mean divergence rate is found around 180-190 hPa, close to the tropopause. Near to this level, many convective systems have another level of neutral divergence, i.e., the upper boundary of the divergent outflows in ICON-PER (as expected).

PAR-profiles also reveal a strong divergence maximum directly beneath the tropopause (Figure 6b and Figure S5 in the supple-

ment), just below the 200 hPa level. The typical level of neutral divergence is shifted downward by about 15-20 hPa compared to ICON-PER. However, the variability of the lower level of zero divergence, between 550 and 350 hPa, increases when utilising a deep convection parameterisation in our case (see also Figure S5). Furthermore, the mean lower bottom level of the divergent outflows is located about 50 hPa lower in ICON-PAR than in ICON-PER.

According to Figure 6, the maximum of the instantaneous mass divergence rate occurs between 200 and 300 hPa. The spatial

variability of horizontal divergence rate for ICON-PAR and ICON-PER ensembles is illustrated in Figure 7 b-d at approxi-

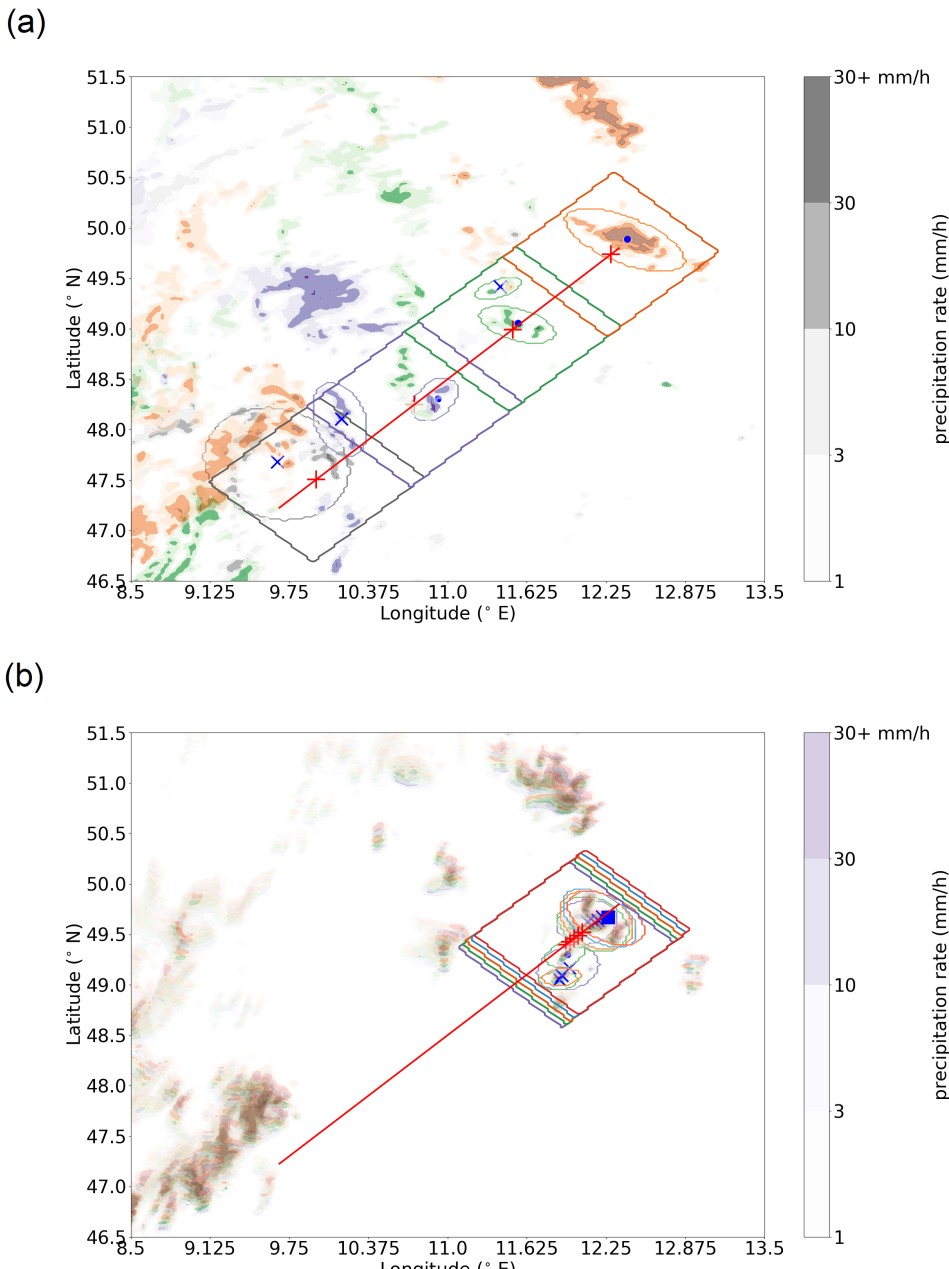

**Figure 4.** Precipitation rate (mm/h) in ensemble member 14 of the PER simulations at (a) 12:30 UTC (grey color scale), 14:30 UTC (purple), 16:30 UTC (green) and 18:30 UTC (orange). The color intensity represents precipitation rate according to the color bar shown for 12:30 UTC. Same for 17:30-17:55 UTC with 5 minute intervals (b). The box outline (tilted rectangles) designed to track the convective system is displayed in the same color. The edges of ellipses matched with the box outline are also indicated. The track of the box with time is indicated by a red line and its centre location is indicated by a +. The distance from that red plus-sign to the ellipse center (blue markers) is evaluated and marked with an x for distances larger than 11 grid cells (about 25 km), a blue circle for those within 20 km and a blue square for those at 20-25 km distance.

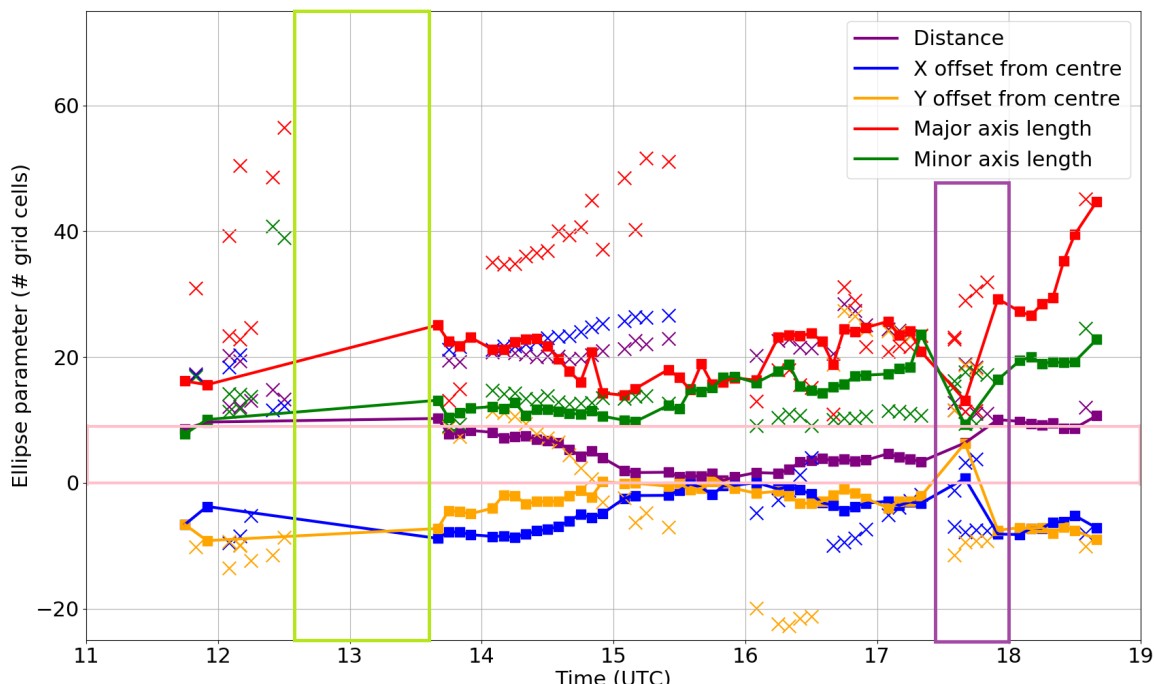

**Figure 5.** Example of time evolution of ellipse parameters in the dataset for the same convective system as shown in Figure 4, halfway through the validation process. Different colors indicate various ellipse parameters. Red: Major axis length, green: minor axis length, purple: distance from box center (+ in Figure 4); blue and orange: offset in x- and y direction from box center). Crosses represent rejected ellipse records for any of the final datasets. Square markers with a line indicate accepted records. The subset within the pink solid rectangle defines which records are in the small subset of 456 records.

mately the level of maximum divergence. Surface precipitation rates of ICON-PAR are also shown. Figure 7a shows the tracks of the convective systems (as derived from the ellipse dataset) passing over southern Germany in PER. The convective systems generally move from southwest to northeast through the domain. Furthermore, their mean intensity increases gradually over time (as manifested by their cross-correlation coefficient with time; as to be shown later, in Figure 11 of Section 6). While
the overall mean precipitation rate over the moving boxes is 3.1 mm/h, the value increases to 4.4 mm/h between 17:30 and 19:00 UTC. Furthermore, the average position of the box centres moves toward the northeastern quadrant of the simulation domain in the last 1.5 hours, coincident with the largest ensemble variability in upper level mass divergence rate (Figure 7 a). Summarised, the large variability in the instantaneous divergence rate is associated with the proximity of increasingly active convective systems.
Results for PAR simulations are shown in Figure 7c and 7d for two different simulation time steps: maxima in instantaneous upper-tropospheric divergence variability are again co-located with enhanced convective precipitation. However, not all regions with (strong) precipitation are directly connected to enhanced upper-tropospheric divergence variability. One possible

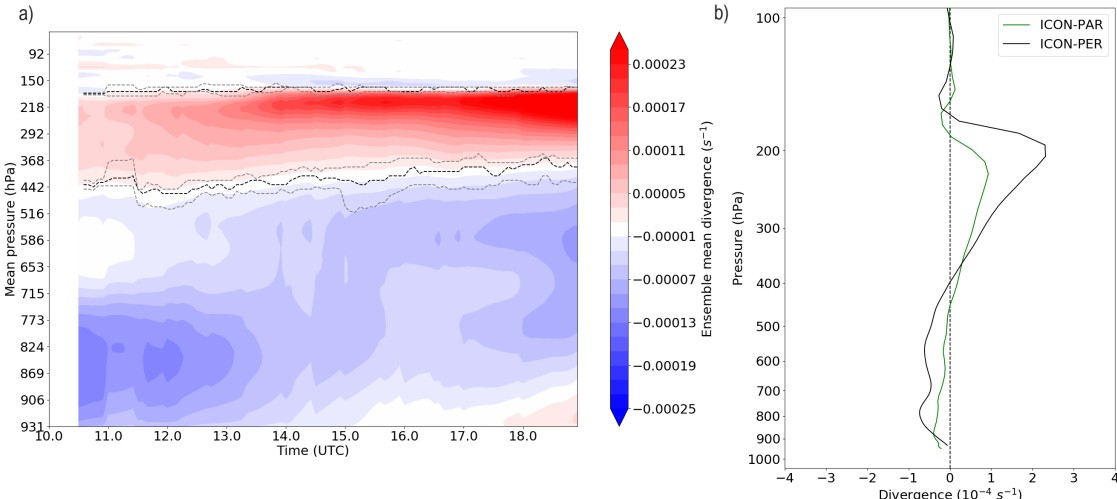

**Figure 6.** (a) Evolution of box-mean divergence (convergence) rate along tracks, as a function of mean pressure. Note that at each instance only a subset of the 28 convective systems is active. Black dashed lines indicate the median level of neutral divergence at any time and grey dashed lines the corresponding 75th and 25th quantiles (nearest to the vertical maximum of instantaneous divergence). The solid purple outline indicates the levels between which the divergent outflow has been integrated in PER. (b) Mean instantaneous vertical divergence profile over the last seven hours of panel (a) (ICON-PER), as well as for ICON-PAR across the analysed systems and ensemble members.

explanation may be the release of latent heat predominately at lower tropospheric levels, which would lead to divergence in the middle instead of the upper troposphere (within the regions with surface precipitation and no or weak upper-tropospheric 405 divergence variability). Another reason for weak connectivity are overall small deviations from the ensemble mean, both in precipitation rate and mass divergence rate.

In Figure 7 the convective system over the Swiss Alps ($9°E$, $47°N$, panel c) and those over Northern-Central Germany ($51°N$, $10°E$ and $53.5°N$, $12°E$, panel d) are the regions dominating instantaneous divergence variability. Consequently, rectangular boxes (purple) define the integration mask for the following analysis.

**5.2 Comparison of relationship between net latent heating rate and instantaneous outflow divergence rate in ICON-PER and ICON-PAR**

Figure 8 shows the relation between outflow mass divergence rate and precipitation rate in all of the analysed ICON-simulations. Note that we switch to instantaneous mass divergence units from now on, since Figure 7b, c and d showed divergence at a fixed pressure level (i.e. nearly constant densities on that level), whereas from here on vertically integrated values divided by the 415 integration depth are used (giving the vertical mean value of the integrand in kg per cubic metre per second). The line of low D corresponds to the slope of highly organised squall lines in an LES-study, which approaches the limit of 2D convection from Groot and Tost (2023b). Only records ($n = 456$) with validated ellipses are included in Figure 8a. Furthermore, the temporal

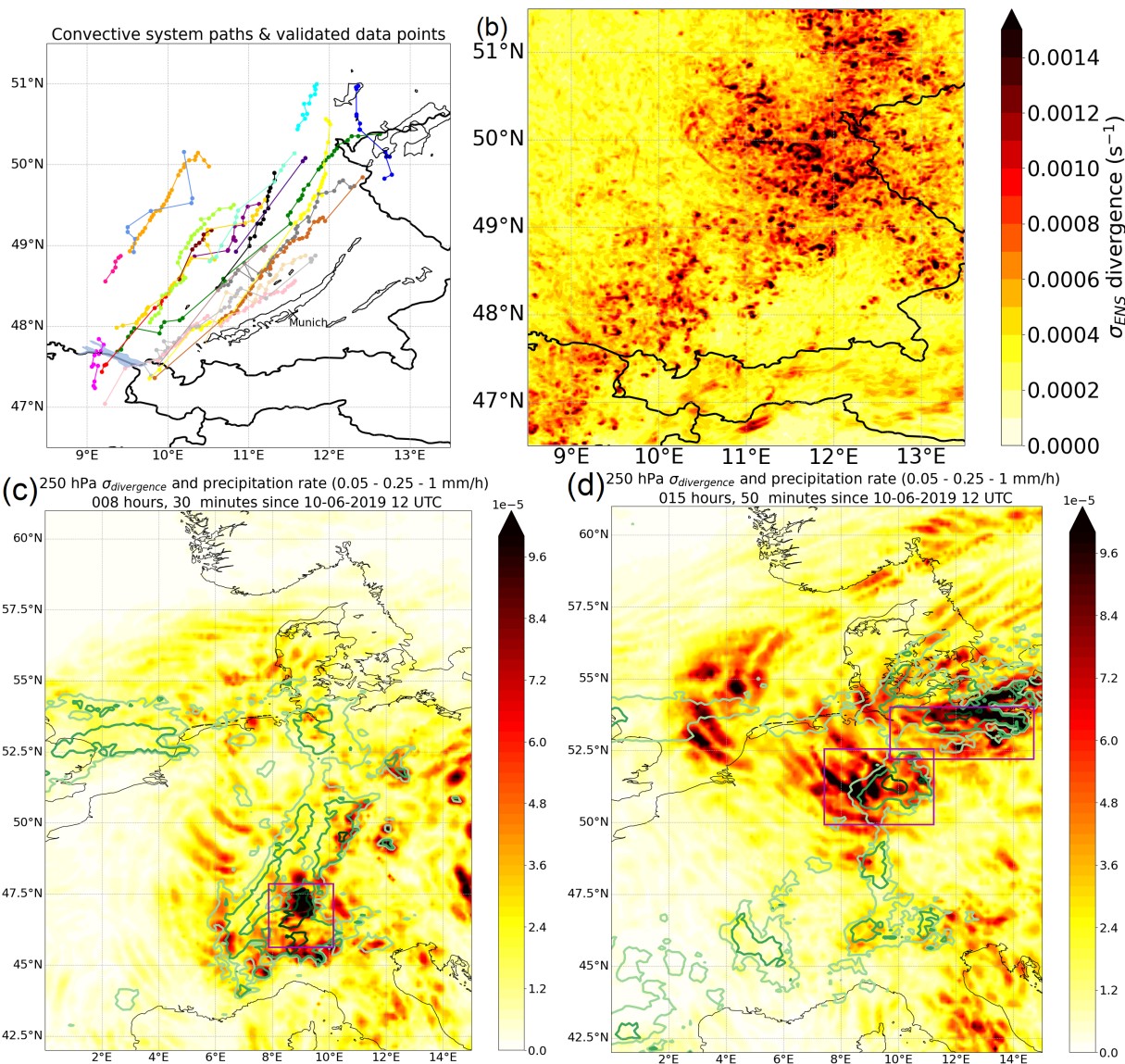

**Figure 7.** (a) Paths of convective systems over Southern-Germany as included in the dataset of 456 records for ICON PER. In black contours, tracks of observed convective systems with > 55 dBz reflectivity are shown for the same day, which generally appear further to the southeast than those in ICON. The color shading in b-d shows the ensemble standard deviation of the instantaneous divergence. (b) $\sigma_{ENS}$ at 255 hPa and the 10th of June 2019 18:00 UTC for ICON-PER. (c) $\sigma_{ENS}$ at 250 hPa and the 10th of June 20:30, resp., (d) 11th of June 3:50 UTC, both for ICON-PAR. Isolines in light green to dark green indicate precipitation intensity over 0.05, 0.25 and 1 mm/h (ensemble mean; bottom plots); the boxes surrounding three convective systems are outlined in purple.

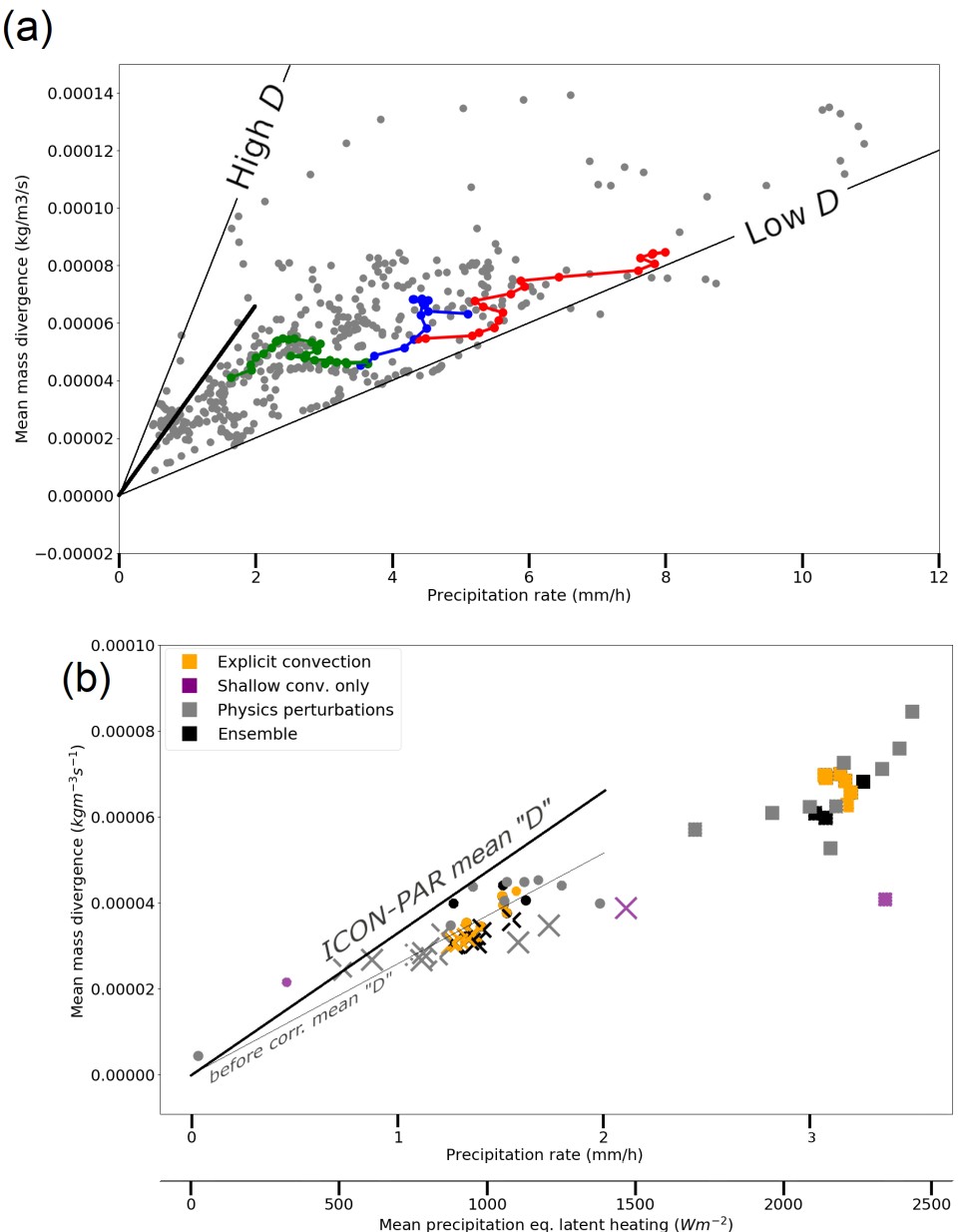

**Figure 8.** (a) Divergence-precipitation rate relationship (convertible to divergence-column net latent heating relationship) for ICON-PER-simulations in the validated dataset of 456 records (grey), plus the time evolution of three convective systems that form a short squall line in three ensemble members (colors). Divergence is integrated over the 380 to 180 hPa layer. (b) Same relation integrated over model levels from 420-430 hPa up to 175 hPa for ICON-PAR (black: ensemble and parameterisation calls at lower frequencies; grey: perturbed latent heating or convective momentum transport; orange: no deep convective parameterisation; purple: shallow convection parameterisation only). In (b), three different markers correspond to three different convective systems, which correspond to the three purple boxes of Figures 7c and 7d. In both panels, the mean value of quantity "D" (ratio of instantaneous mass divergence rate and precipitation rate) based on ICON-PAR is also annotated as bold black line, and both before and after correction for the difference in outflow layer thickness in the bottom panel b (before the correction: thin black line; see annotating text of b).

evolution of three separate convective systems that eventually develop into squall line-like structures are highlighted by colored symbols. In particular, these highlighted points correspond to the part of their evolution, during which squall line-like development gradually occurs. Note time in general increases with increasing precipitation intensity. These systems are thought to resemble two-dimensional convection much closer than isolated three-dimensional-like systems.

The ratio between instantaneous mass divergence rate and precipitation rate effectively represents the normalised mass divergence rate $D$ (if the intercept at 0 mm/h corresponds to 0 divergence, which is a reasonable assumption for *cumulonimbus* clouds, but this assumption does not hold for the non-precipitating stages of clouds). Our hypotheses and the conceptual model (Figure 1 and the accompanying paragraphs of Section 1) suggest that this ratio is not expected to be constant over time, unless the convective overturning remains either two-dimensional or three-dimensional as a result of constant geometry of the convective system. Hence, when noise is removed, the time evolution of a convective system in the "precipitation rate"-"divergence rate" space is expected to correlate with the change in convective overturning as a result of changing convective organisation. If a squall-line-type of organisation develops, the geometry is gradually expected to become increasingly two-dimensional. Accordingly $D$ should become rather low (at least when systems developing squall-line-like characteristics are separated from the sample mean of $D$). Therefore, we expect that systems while developing squall-line-like characteristics gradually move towards lower $D$ as the systems grow and their precipitation rate increases. In the following, we regress the instantaneous mass divergence rate with the precipitation rate for the three selected systems, to assess whether this hypothesis is true.

Figure 8a shows that, while squall line structures develop (colored lines), $D$ (i.e., normalised divergence rate) indeed moves towards typically lower values over time. Consequently, mass divergence rates become comparatively low compared to a fitted mean mass divergence rate at a given precipitation intensity taking all data points into account. For the green system with lowest precipitation intensities, the slope of a linear least squares fit to its evolution in the precipitation rate - mass divergence rate space is negative. Therefore, it develops (in the diagram) towards a low normalised divergence rate, while squall line-like structures develop. The fit to the evolution of the other two systems in the precipitation rate - instantaneous mass divergence rate space (blue and red markers) has a positive slope, as typical for the background scatter. However, the intercept of the linear fit at 0 mm/h precipitation rate lies below the intercept value representative of the background fit (see Table 1 in the Supplement for the intercept and slope parameters of the linear fits). Therefore, these systems are at lower $D$ than the background, too. Furthermore, Figure 8a suggests that for the latter two systems the gradient of mass divergence rate with precipitation rate decreases as precipitation rate increases, i.e. over their lifetime and the increasing resemblance to a squall line-like structure.

The propagation of these squall line-like systems towards lower than expected $D$ at a given precipitation rate in Figure 8a fits the expected type of outflow source on convective outflows (Groot and Tost, 2023b): namely that systems that resemble line sources of heating emit gravity waves causing reduced divergence rates at the same heating rates, compared to point sources (corresponding to isolated cumulonimbus updrafts). In Section 6 we will further investigate if the variability in ratio $D$ within the ICON-PER dataset aligns with the conceptual model of Figure 1. First, the corresponding character of the variability within the ICON-PAR dataset is discussed and then compared to ICON-PER.

The PAR-simulations, illustrated in Figure 8b and representing three different convective systems as marked with crosses, squares and dots, suggest a roughly linear relation between instantaneous mass divergence and net latent heating rates. If

neutral divergence is assumed in the layer excluded from the vertical extent of PER-integration masks, PAR and PER can be compared, even though the integration depth differs by about 50 hPa between the two (about 200 vs. about 250 hPa pressure thickness). The expected impact, based on an assumption of neutral mass divergence in layers excluded from the analysis, would translate to a $\approx 25\%$ stronger outflow in PER. Based on this assumption, the corrected mean PAR-divergence rate would correspond with a steeper slope in the Figure than the slope of the regression line of $D$ in ICON-PER. The corrected and uncorrected sloping lines from ICON-PAR are illustrated in Figure 8b, while only the corrected line of ICON-PAR is visualised in 8a. Hence, a direct visual comparison between PAR and PER is possible.

On average, enhanced outflow rates occur in PAR compared to PER at given net latent heating rates. The relationship for unperturbed parameterised deep convection (black, Figure 8b) appears to be very close to linear, as little or no information on geometry of the convective systems can be represented with a coarse grid spacing (especially of geometric heating structures within, which are crucial for accurate and complete spectral representation of gravity wave sources), possibly further limited by parameterisation (see also Section 3.2.2). The appearance of a squall line structure is missing; precipitation structures of such a squall line would not be convective and would closely resemble the dynamics of tilted lifting, as typically associated with a frontal zone (see Appendix C2 of Groot, 2023). The relationship is also linear for the ensemble without any convection parameterisation (orange), suggesting that the effect of grid spacing dominates the effect of parameterisation (although the impact of parameterisation itself may depend on factors as grid spacing) . If only shallow convection is parameterised (magenta markers), the outflow of one system deviates substantially from the linear relationship. This can be explained by a considerable downward shift of the outflow layer. Consequently, the integration mask as defined in Figure 6 misses the dominant levels of convective outflow, as the outflow is redistributed from upper levels towards mid-levels.

In short, Figure 8b suggests that the coarse resolution linearises the precipitation rate-outflow relationship and that the spread is only represented at higher convection-permitting resolution, in ICON-PER, presumably by representing refined cloud heating, which results in gravity wave emission and dynamical interactions between these gravity waves. Furthermore, physical perturbations in the parameterised configuration cause only slight deviations from this approximately linear relationship. The detection of minor conditional outflow spread (when conditioned on heating rates) agrees well with expectations from the weakly represented convective organisation in parameterised convection or coarsely resolved explicit deep convection. Explicitly "resolving" deep convection at 13 km grid does not affect the suggested linear relationship. On the other hand, in convection-permitting simulations at 1 km horizontal grid spacing the ratios between latent heating and outflow rates vary. The envelope of variation is roughly consistent between ICON-PER and the idealised LES-simulations of Groot and Tost (2023b). As there are no systematic patterns of a residual instantaneous outflow rate-net latent heating rate relationship obvious in the ICON-PAR simulations, the analysis of such patterns is restricted to the ICON-PER configuration in Section 6.

## 6 Dependence of divergent deep convective outflow rates on properties of convective systems in convection-permitting ICON simulations

This section discusses the representation of instantaneous divergent convective outflow rates in ICON-PER, following the conceptual model outlined in the introduction and then discussing the role of CMT. The conceptual model in the introduction suggests that divergent outflow strength depends

- linearly on net latent heating rate (hence, also on precipitation rate)

- on the storm geometry (point or line heating source)

- on interactions between outflows from individual convective cells as a result of convective clustering, through outflow collisions

The main diagnostics used in this section are: (i) the ratio $D$ between instantaneous mass divergence rate in the upper troposphere and the corresponding net precipitation rate (see Table A1) and (ii) the ratio $C$ between the eddy momentum flux and precipitation rate. In addition, we use ellipse parameters to describe the geometry and mean-flow relative orientation of the 495 convective elements.

### 6.1 Elongation of convective systems

The elongation of convective systems is quantified by the ratio $A$, which is defined by the ratio of two axes of the ellipses fitted to the convective systems. From the LES study and the conceptual model of Groot and Tost (2023b) lower $A$ is expected for systems with a lower $D$, at a given precipitation rate. Furthermore, during the evolution of a convective system $A$ is expected 500 to correlate positively with $D$. Finally, for two-dimensional convection it is expected that the convective inflow and outflow are mostly parallel to the tropospheric mean winds, resulting in a typical ellipse orientation $O$ perpendicular to these winds.

At lower precipitation rates below 6 mm/h, where ratio $A$ is distributed over each of the three classes, no clear relationship between $A$ and $D$ is found. This suggests that the elongation of convective systems is not the only parameter accountable for anomalies in the "outflow strength" - "latent heating" space. The classification into three $A$-classes can be sensitive to 505 thresholds of $A$, but Figure 9 shows that sorting of $A$ and $D$ is not supported.

The ellipse dataset is split in subsets for further investigation. After selecting the subset with $> 2.5$ mm/h precipitation rate first, within this subset, two subsets of strong $D$ anomalies are created: one subset that exceeds $115\%$ of the conditional mean of quantity $D$ ("low D" class) and another where $D_{sample} < 0.85 \times D_{con.mean}$ ("high D" class). The "high D" class is associated with an average $A$ of 0.602, versus 0.542 for the "low D" class (see also Table B1, Appendix B). The mean value over the whole 510 dataset is 0.56 with $\sigma$ of 0.18. Therefore, the difference of the expected (i.e. positive) sign is significant at 95% confidence. Nevertheless, the difference in $A$ between the classes is lower than expected based on Groot and Tost (2023b).

Furthermore, variability in ellipse orientation $O$ within the low $D$ subset is strongly reduced compared with the high $D$ subset: $\sigma = 32°$ for low $D$ versus $\sigma = 44°$ for all records and $\sigma = 45°$ for the high $D$ subset. In short, very similar ellipse orientations (reduced variance in ellipse orientation) occur at low normalised divergence rates (Table B1, Appendix B).

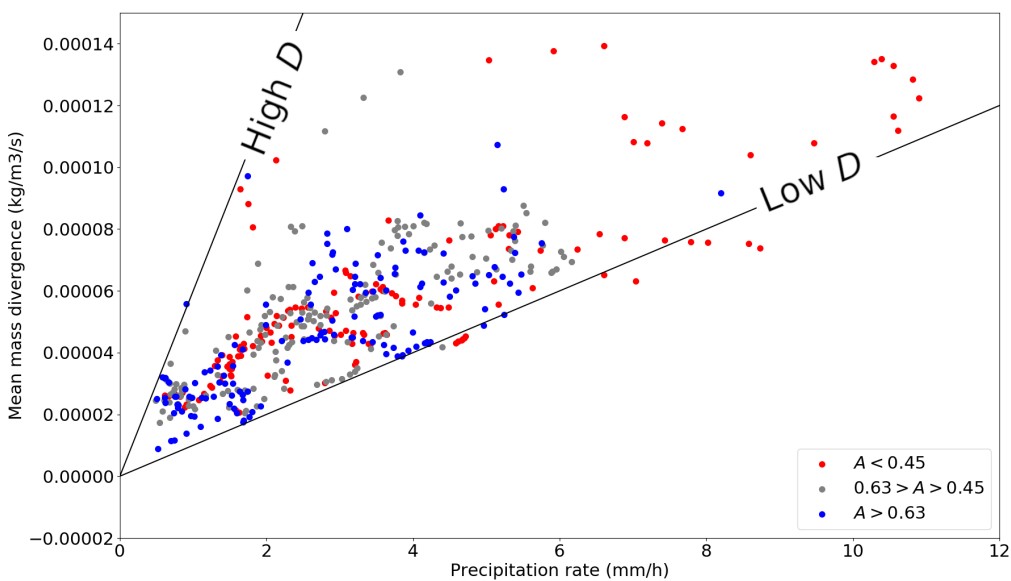

**Figure 9.** Divergence rate-precipitation rate dataset in ICON PER simulations with colors indicating three similarly sized classes of axes ratios. Added are two black lines of constant $D$: those with 6e-5 $\frac{kg \cdot mm}{m^3 s^1 h^1}$ and 1e-5 $\frac{kg \cdot mm}{m^3 s^1 h^1}$.

## 6.2 Aggregation of convective systems

Figures 8a and 9 suggest a deviation from the linear relationship towards larger precipitation rate, i.e. reduced $D$ for convective systems with increasing precipitation rates. In this subsection, we quantify the off-linearity of the relationship between instantaneous mass divergence rate and precipitation rate, which has been found for ICON-PER, but not for ICON-PAR.

A reduction in mass divergence rate may be caused by the collision of individual three-dimensional outflows from individual cells, as induced by convective aggregation. Hence, convective aggregation may reduce divergence rates, relative to isolated convective cells, as more precipitation cells develop within an area. Measures that can indicate the presence of developing and clustering convective systems are ellipse area and area of high ($> 10$ mm/h) precipitation rate (Table A1 in the Appendix; Figure 11). Furthermore, precipitation intensity itself generally increases with an increasing number of mature precipitation cells in a small area, also an indicator of convective clustering.

The expected negative correlation of the instantaneous mass divergence per unit precipitation intensity $D$ with increasing size and (precipitation) intensity of the convective systems is found in the dataset (11). The most important relation connects precipitation intensity and the ratio $D$ with a Pearson correlation coefficient of -0.59 in the fully validated dataset and -0.52 in the larger dataset ($n = 866$). The negative correlation bends off the scatter in Figure 9 towards lower divergence rates than in case of a continued linear relationship (like in Figure 8b).

The robust negative correlation coefficient between $D$ and precipitation rate implies the non-linear behavior within the envelope of Figure 9 is partially predictable: a non-linear best fit between divergence rate and precipitation rate is expected. A power law with power $< 1$ could optimally fit the relation between instantaneous mass divergence rates and precipitation rates. Indeed, a best fit for the smaller dataset is obtained with an exponent of 0.704. The lowest least squares residual to predict the divergence rate from precipitation rate uses this exponent. For the larger dataset, the exponent is 0.606. With bootstrapping the

uncertainty in the transformed fit of the smaller dataset is investigated. The 95% confidence of the power transform was estimated at 0.526 to 0.851. However, since multiple highly correlated parameters contribute to the fit (intercept, slope, exponent), the actual parameter uncertainty is likely smaller.

   Conditional correlations between the ellipse area and $D$ are evaluated within precipitation rate bins. These correlations support the representation of convective clustering and its consequences for outflow collisions within ICON-PER, consistently with

LES (Groot and Tost, 2023b). The conditional correlations are summarised in Figure 12: the (sample size) weighted mean correlation coefficient is -0.32, which is significant at 95% confidence. Furthermore, the area $> 10$ mm/h precipitation rate reveals the same pattern, with a weighted mean correlation coefficient within precipitation bins of -0.29.

   The analysis suggests the following evolution of the convective characteristics: increasing precipitation intensity forces a linear increase in instantaneous mass divergence rate in the upper troposphere, initially. However, beyond a certain precipitation

intensity the instantaneous mass divergence does not keep up with the initially linear relation anymore. At higher precipitation rates, instantaneous mass divergence tends to grow comparatively slower (i.e. negative feedback). This signal was exemplified by the developing squall-line-like structures in Figure 8a. The non-linear divergence rate reduction is stronger in squall line-like structures than in the average of all sampled convective systems. These convective systems move towards the right lower corner in Figure 8a.

In the supplement (Figure S6) surface-based and mixed/elevated convection subsets are analysed separately, where a fingerprint of convective aggregation is present, too. .

## 6.3    Role of convective momentum transport

   For the larger dataset with less strict matching criteria, the effect of convective momentum transport on mass divergence rate has been investigated by normalising both quantities with the precipitation rate ($C$ and $D$) and analysing conditional

correlations of $C$ and $D$ within precipitation rate bins. Thereby, the first order effects of precipitation intensity on instantaneous mass divergence rate (Section 6a,b) are filtered out. Figure 10a shows zonal (x-axis) and meridional (y-axis) components of CMT, while Figure 10b shows the relation between quantities $C$ and $D$ over two separate ranges of precipitation rate. The precipitation rate bins to diagnose the conditional correlations are constructed such that the ratio between upper and lower bound of each precipitation rate bin is about 4 to 5 and the combination of all bins cover the interval 0.6-6.25 mm/h (Figure

12).

   Over the 11 bins containing 39-150 samples each, the weighted average of the conditional correlation coefficient is 0.31. The equal weight average is 0.34. There are exclusively positive correlations with values up to 0.7-0.8 across the range of bins. Given these statistics, the true correlation coefficient lies probably within the interval 0.2-0.5. Therefore, a small fraction of

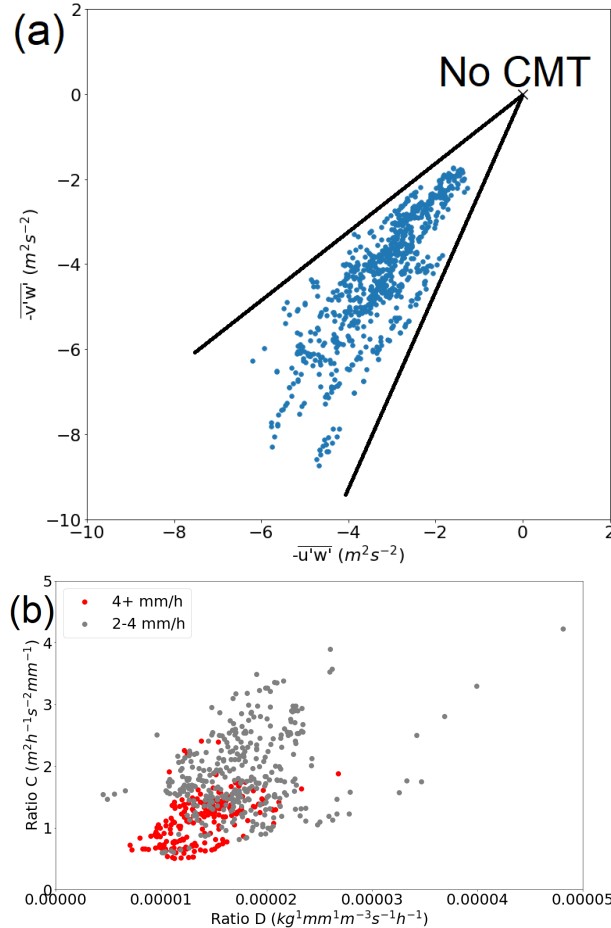

**Figure 10.** Top (a): two components of the diagnosed vertical CMT integral at 315 hPa (overbar denotes mean operator). Bottom (b): relation between upper-tropospheric mass divergence rate and the absolute acceleration derived from the CMT diagnostic, both normalised to precipitation rate and for two different classes of precipitation rates (red and grey).

outflow variability in the convective systems can probably be explained by variability in CMT (Figure 10b).

No single data point with upgradient transport occurs within the dataset (Figure 10a), since the CMT-fluxes oppose the predominantly southerly wind shear vector of this event, hence reducing the vertical gradient in the wind speed, and therefore, the wind shear. The sample of 866 records is not fully independent, as only 28 independent convective systems are represented with records at small time lags being correlated. The temporal evolution of several convective systems in a single synoptic environment is of course somewhat biased towards a specific scenario. On the other hand, the coherent background flow

supports identifying subtle patterns in the dataset. This contrasts strongly with the experimental method in Groot and Tost (2023b) to study CMT effects on mass divergence rate.

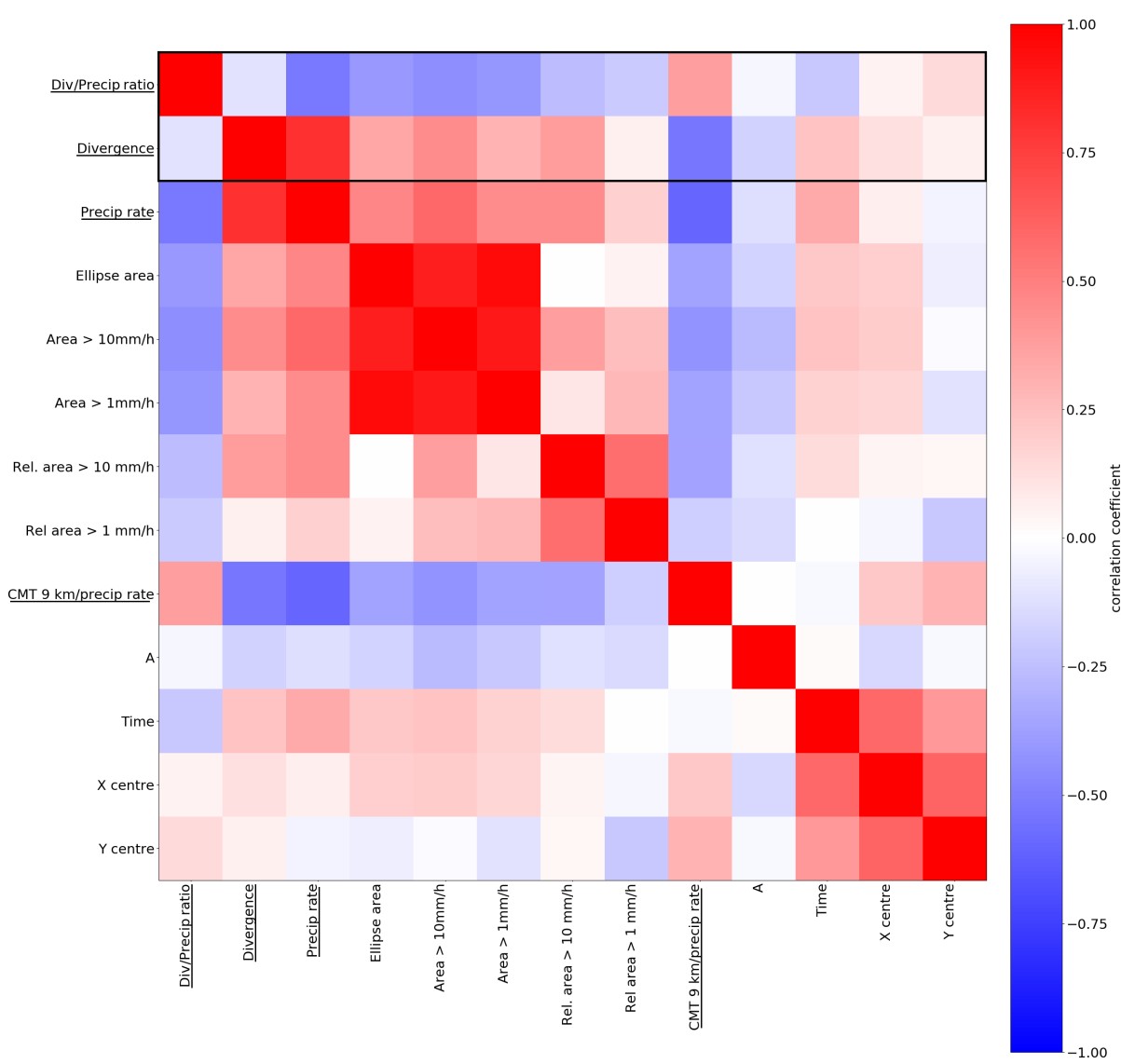

**Figure 11.** Overview of the correlation structures assessed in Section 6. Underlined variables indicate those that have been derived from the box integration, those without the line indicate variables extracted from ellipse parameters. Based on larger dataset with $n = 866$ samples.

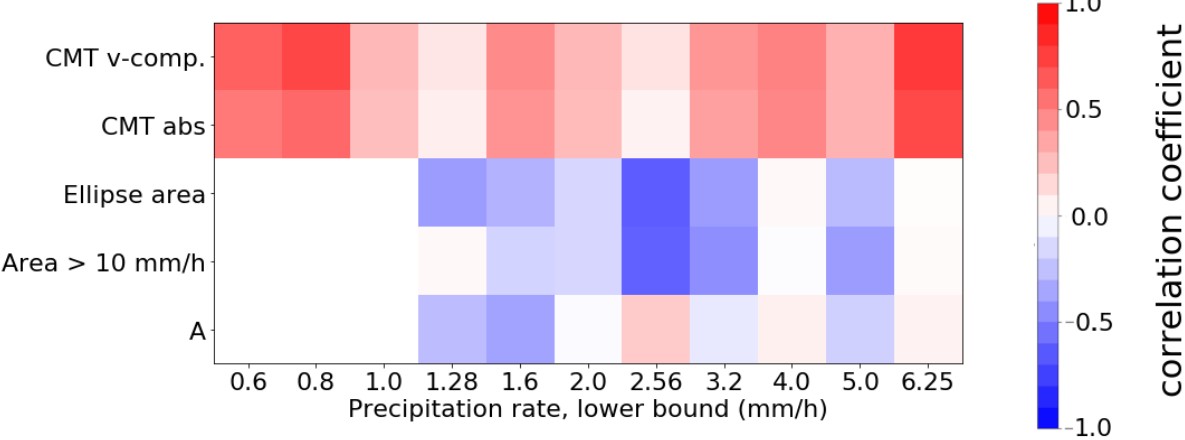

**Figure 12.** Overview of the correlation structures conditional on precipitation rate as assessed in Section 6. Based on the smaller dataset with $n = 456$ samples. Low precipitation signals are partially omitted due to small sample sizes and weak convection (under 1 mm/h mean precipitation rate over integration box).

## 7 Synthesis

### 7.1 ICON representation of divergent outflows

This study has investigated instantaneous divergent outflow variability from deep convection in ICON, conditional on the pre-
cipitation intensity in ensembles with convection-parameterised (PAR) and convection-permitting (PER) setups.

PAR-simulations show an approximately linear relationship between precipitation rate (a close proxy for vertically integrated latent heating rate) and the instantaneous outflow strength with little spread. Conversely, in PER a non-linear relation between between these two quantities is found, accompanied by substantial scatter away from the mean relationship.

The convection-permitting simulations have been utilised to explore hypotheses on the controlling factors in the relationship
between convective latent heating / surface precipitation and upper-level divergence rate derived from idealised studies (Groot and Tost, 2023b) in a real-case simulation: In particular, we investigated the impact of convective organisation, clustering and convective momentum transport. The expected impact of convective clustering and organisation on the divergent outflows is illustrated in Figure 1: A point source of heating and a linear heat source in the horizontal plane can be viewed as conceptual extremes of convective clustering and organisation. The instantaneous outflow strength from these two scenarios strongly varies
even at identical precipitation rate (Groot and Tost, 2023b; Nicholls et al., 1991). Therefore our first hypothesis ("dimensionality hypothesis") is that with increasing elongation of a convective system outflow strength decreases (at identical precipitation rate). Furthermore, clustering of convective cells leads to collisions of upper-level outflow, which reduces the net instantaneous mass divergence through convergence and compensating vertical circulation ("convective clustering") over mesoscale regions. The second hypothesis ("clustering hypothesis") tested in this paper is therefore that increasing convective clustering (e.g.
during temporal evolution a convective cloud field) decreases outflow strength (at identical precipitation rate). To focus on the

variability of instantaneous outflow strength conditional on precipitation rate, we use the ratio of upper-level divergence rate to surface precipitation rate, $D$. In this work, it has been investigated whether each of these two controlling factors on area average mass divergence rate contribute to variability in outflow strength in the convection-permitting configuration of ICON, during a selected convective event over Germany. Mixed results are identified with respect to the first aspect: the geometric shape of the convective heat source. On the one hand, substantial spread is clearly identified in the divergent outflow intensity as a function of precipitation rate (Figure 9). Subcategories with high and low $D$ are found to consist of statistically different cell geometries. Furthermore, contrasts between the two subsets in ellipse orientation relative to the background flow are consistent with expectations. On the other hand, no clear correlation between elongation $A$ of the cells and $D$ is found.

The evidence for the "clustering hypothesis" is strong: the reduction of the ratio $D$, between instantaneous net mass divergence and precipitation rate, with increasing area mean precipitation rate was confidently identified in the realistic convection-permitting ICON configuration. A consistent correlation signal is identified among several additional ellipse parameters (Section 6.2) and $D$, including total ellipse area and ellipse area fraction of convective precipitation ($> 10$ mm/h) and $D$. The sub-linear increase of instantaneous outflow strengths with precipitation rates signifies a negative feedback from increasing diabatic heating onto upper-level divergence rates, as a result of outflow collisions and compensating (adjacent) convergence in the upper-troposphere.

As a third and final hypothesis we investigate the direct impact of convective momentum transport on instantaneous outflow strength. Given a certain precipitation rate, it is suggested by the findings here that flow perturbations induced as a result of convective momentum transport can likely impact the mass divergence rate slightly in ICON-PER. However, details of the interactions cannot be derived from this study. Furthermore, note that the indirect impact of convective momentum transport is technically included when convective geometry and clustering are investigated, as convective momentum transport can impact convective organisation and therefore precipitation rates (see also Groot and Tost, 2023b).

## 7.2 Discussion

Our analysis provides insight into the instantaneous divergent outflow variability in ICON for the selected case-study and the mechanisms that can govern the variability. The amplitude of instantaneous divergent outflow is proportional to net latent heating rates, which has been known (Nicholls et al., 1991). However, at a certain latent heating intensity, it is now also clear that LES and convection-permitting NWP allow for a substantial variation in mean divergent upper-tropospheric outflow rates (see also Groot and Tost, 2023b). Our ICON case study indicates that the variation of instantaneous outflow magnitudes at a given heating rate is determined by the geometric structure of the convective systems, consistently with earlier results from LES, but likely along with small direct modulations by convective momentum transport in ICON-PER.

### 7.2.1 Conceptual understanding of divergent outflows

The "dimensionality hypothesis" is not strongly supported by our analysis, albeit some indirect evidence points to an impact of dimensionality on divergent outflow rates. Three suggestions are made, why the dimensionality hypothesis is not strongly supported by the statistics:

- The chosen metric is sub-optimal: it is not able to distinguish nearly-two-dimensional ("line source of heating") and nearly-three-dimensional convection ("point source") well (Figure 1); furthermore, real cases often cannot be unambiguously categorised into the two classes (e.g. Trier et al., 1997).

- The (elevated) shear profile of this case does not induce the maximum possible variation in dimensionality of the deep convective overturning (from nearly-two-dimensional to nearly-three-dimensional)

- Opposing statistical relations between ellipse parameter estimates (e.g. ellipse elongation $A$) and the potential effect on instantaneous mass divergence compensate each other, even within precipitation bins.

Each of these will be discussed in the following paragraphs one by one.

The first possibility is that our metric for system dimensionality, namely the ellipse elongation, does not adequately map the variability in geometry of small convective systems. In our case study only few systems develop clear structures. Future assessments involving more extensive datasets of convection-permitting simulations from across the globe and various cases or, alternatively, different algorithms (guidelined by Groot and Tost (2023b); Trier et al. (1997) and quantifying geometry based on storm-relative flow) could clear-up on this issue A second explanation of improper sorting of overturning characteristics could be insufficient variability in the overturning dimensionality in the investigated case-study. We essentially may not sample nearly-two-dimensional convective overturning, found in well-organised squall lines (e.g. Moncrieff, 1992; Trier et al., 1997; Groot and Tost, 2023b) with strong updrafts along a concentrated line of strong near-surface convergence and accompanying density current. For such squall lines, we would expect strong similarities in orientations of elongated ellipses.We found such a pattern, but squall line segments have lengths of only up to about 100 km. Long and narrow squall lines forming in very strong near-surface shear (like in Groot and Tost (2023a); Rotunno et al. (1988); Coniglio et al. (2006)) probably resemble two-dimensional convective overturning much better than our systems. Well-organised squall-lines are typically associated with strong shear at low levels perpendicular to the orientation of the convergence line (also associated with a strong cold pool), causing nearly two-dimensional overturning. However, in our case strong wind shear is concentrated in a somewhat elevated layer (in PER: at 2-4 km a.g.l.) and the angle to the frontal boundary is small (Section 2), possibly resulting in nearly-three-dimensional overturning.

Lastly, estimated ellipse elongation parameter $A$ and convective aggregation may very well be anti-correlated. However, based on our conceptual understanding both can plausibly impact the mass divergence. Therefore, their effects on the mass divergence could potentially compensate each other (see Figure 11). Therefore, a third possibility for the weak evidence obtained for the "dimensionality hypothesis" is that ellipse parameter $A$ hides the direct dimensionality signal in the outflow variability. The signal may even be hidden if conditional correlations between $A$ and $D$ within precipitation rate bins are considered (Figure 12). This is possible if $A$ is a sub-optimal proxy for the dimensionality of convective outflows. The two correlation effects expected (based on conceptual understanding) could oppose another within precipitation bins, unless these bins approach a limit of near-zero width. Consequently, analysed patterns may seemingly be explained by convective clustering only, even if the dimensionality hypothesis explains a substantial proportion of the examined instantaneous outflow variability.

Summarised, the outflow geometry and dimensionality of convective overturn seems to contribute only weakly to outflow

variability. Still, the signal associated with squall line development supports the dimensionality argument, consistently with findings by Bretherton and Smolarkiewicz (1989); Nicholls et al. (1991); Groot and Tost (2023b) and others.

### 7.2.2 CMT hypothesis

The impact of CMT on the instantaneous outflow strength is may be much more pronounced based on this study than based on the LES study of Groot and Tost (2023b). This might be due to the difference in the wind shear profiles between the LES study and the real case investigated here: Upper-tropospheric wind shear was completely absent in (the initial conditions of) the LES-configuration, whereas any real world has non-zero shear in the upper troposphere. Shallow shear configurations (e.g. all shear at levels below 3 km altitude) can reduce the average height that parcels in convective cells reach (Coniglio et al., 2006). The associated reduced vertical overturning would subsequently suppress the interaction between the shear layer and the divergent outflows (in agreement with Brown (1999)), such that eddy momentum transport through the upper half of the troposphere may hardly occur. Therefore, effective local acceleration or deceleration of the flow might not reach the upper-troposphere. Hence, the impact of CMT on the instantaneous outflow strength is likely suppressed in Groot and Tost (2023b) and CMT was not found to (directly) affect the convective outflows in the LES-simulations. A comparison of LES-simulations where the shear is more evenly distributed over a deeper layer, with a setup similar to (Groot and Tost, 2023b), would be beneficial to assess if this hypothesis is true. The presence of shear over a much deeper layer is a more realistic scenario and therefore should be assessed in a complementary study with LES-configuration.

Additionally, the ICON-PER configuration is arguably most suitable for detecting subtle (reasonable, real case) CMT impacts on the divergent outflow rates. Conversely, the setup does not allow in-depth understanding of the mechanisms behind instantaneous outflow variability, due to the complexity of the scenario and the amplitude of the systems in close spatio-temporal proximity.

Overall, it is clear that further research is needed for a basic understanding of the interaction between the characteristics of divergent convective outflow and their relation to convective momentum transport, as well as to further describe the two-fold (direct and indirect) role of convective clustering and organisation therein. The different role of CMT for instantaneous upper-level divergence between the LES and NWP study provide little foundation for more than speculation of the mechanisms that may act. Similarly, to investigate how the CMT-accelerations could mechanistically affect flow predictability is beyond the scope of this work and would require advanced, specifically tailored methods.

## 7.3 Implications for predictability in NWP

### 7.3.1 Predictability and uncertainty in divergent convective outflow strength

Previous studies investigating the predictability of the atmosphere from a dynamical perspective have identified that ensemble spread amplifies strongly in regions of precipitation, and in particular convection (e.g. Zhang, 2005; Zhang et al., 2007; Selz and Craig, 2015a, b; Baumgart et al., 2019; Selz et al., 2022). Baumgart et al. (2019) have investigated the sequence of dynamical processes that (on average) contribute to mid-latitude growth of flow perturbations. They proposed that latent heating

tendencies from their deep convection scheme in ICON may induce differential divergent winds in the upper troposphere, which may interact with a nearby jet stream to constructively amplify perturbation growth. Subsequently, further non-linear growth of flow perturbations in the upper troposphere is driven by differential advection. Practical cases where precipitation systems importantly reduce atmospheric predictability are nevertheless thought to be rare, at typical state-of-the-art initial state uncertainty amplitudes (see Lorenz, 1969; Rodwell et al., 2013; Durran and Gingrich, 2014; Zhang et al., 2019; Selz et al.,
2022).

    An open key question is whether the variability in flow perturbations associated with convective outflows in the upper troposphere is comparably (and reliably) represented in simulations with resolved and parameterised deep convection. The findings of this work suggest that this is only the case when deep convection is explicitly resolved at sufficiently small grid spacing. In parameterised setup, or explicitly resolved, with coarse grid spacing (i.e. larger 10 km), our results suggest that the ensemble
is underdispersive in terms of instantaneous outflow variability with a strong linear correlation between divergent outflow rate and precipitation rate variability.

    This work once more confirms the strong link between precipitation variability and flow variability in an ensemble. This close connection may lead to perturbation growth in a forecast or spread in an ensemble. The downstream propagation of perturbations is not directly addressed here. Nevertheless, the spatial-temporal distribution of instantaneous divergence variability
(Figure 7) is consistent with a potential role for divergent outflows propagating precipitation variability and other convective variability (mostly: CMT) to uncertainty in dynamics at large scales, in line with Baumgart et al. (2019).

### 7.3.2    Flow variability, convective organisation and the model representation of deep convection

    This work suggests that the convective contribution to flow variability can be separated into a component of precipitation variability (i.e. along the x-axis of Figure 8) and another component of superposed conditional divergence rate variability ($D$
variability), induced by convective organisation, which may relate to the results of Rodwell et al. (2013). Extensive examination in an event with upscale impact of the convection is needed to assess the mechanisms acting.

    The separation of divergence variability into the above mentioned components has significance for weather and climate modelling: only the former component seems to be accounted for at coarse grids ( 10 km spacing). The latter component is (nearly) absent in parameterised and coarse-grid configurations, but is accounted for in a convection-permitting setup. Assessing deep-
convective variability, conditioned on precipitation rate, is an important tool to illustrate the two components of variability (e.g. Groot and Tost, 2023a, b). Thereby, the feedback between deep convection and its environment, for instance, can be better discerned.

    The systematic differences between ICON-PAR and ICON-PER regarding the relation between surface precipitation and upper-level divergence rates (Fig. 8) suggest that the feedback from deep convection to its surroundings at larger scales is likely not
accurately represented with parameterised deep convection, even if the precipitation climatology is well represented: parameterised ensembles seem to be underdispersive in terms of corresponding dynamical variability at a given precipitation rate (Figure 8). Nevertheless, explicit deep convection at the same coarse grid spacing (13 km) manifests the same conditional properties as parameterised deep convection. Furthermore, the convective flow feedback to larger scales is likely on average

overestimated in ICON-PAR. An on average overestimated instantaneous deep convective outflow feedback at given global average precipitation rates may substantially impact regional circulation patterns in weather and climate models, potentially contributing to subsequent regional circulation biases.

Groot and Tost (2023b) shows that collisions of divergent outflows cause sub-linear increases of instantaneous divergent outflow rates with increasing precipitation intensities, in LES. Here, we investigate the presence of this effect in two ICON configurations, and find it only in convection-permitting ICON. At low precipitation rates, the ratio between instantaneous mass divergence and precipitation rate in ICON-PAR overlaps with the upper range of the same ratio in ICON-PER and the LES configuration of (Groot and Tost, 2023b). Nevertheless, both ICON-PER and the aforementioned LES study do not maintain these high values of $D$ up to higher precipitation intensities: the values of $D$ decrease at these precipitation rates.

Convective organisation can also affect the vertical extent of mesoscale heating patterns, and thereby change the intensity of the local heating divergent wind forcing, as vertical background stratification changes. Although the results in the current study and Groot and Tost (2023b) suggest that the magnitude of the vertically integrated divergence rate is not substantially affected by the vertical stratification in the outflow region (just the vertical distribution is). However, as we have only investigated a subset of deep convection environments, the full population of convective clouds (including for instance mostly stratiform MCS, medium size precipitating cumulus and weakly-sheared tropical deep convection) might show additional dependencies of the divergent outflow response.

It is known that models represent convective organisation imperfectly, especially whenever a parameterisation scheme and coarse grid spacing is used. Our work suggests that it is important to increase the understanding of convective organisation biases in models. These biases may interact with biases in the precipitation climatology, and even cause compensating errors in NWP. However, these compensating errors may be hidden, unless instantaneous mesoscale mass divergence spread produced at given precipitation rates is specifically included in an analysis. Apart from conditioning on precipitation rates, conditioning on e.g., the diurnal cycle and regional convective characteristics can importantly contribute to improved simulations across resolutions (e.g. Bechtold et al., 2014; Becker et al., 2021, and references therein). Convective organisation biases are known to affect squall line representation in convection-permitting models (Becker et al., 2021) and could likely contaminate succinct results in ICON setups like ours. The conceptual model of (Groot and Tost, 2023b), verified against ICON in this work, provides a possible pathway of how grid scale storms could introduce mesoscale circulation biases despite an accurate precipitation climatology.

Upscale growth and clustering of convective systems are found to be key players for the instantaneous magnitude of divergent outflows in practice, which is properly accounted for only by convection-permitting simulations at about 1 km horizontal grid spacing. This is probably because the small-scale gravity waves emitted by individual deep convective elements, and their interactions after collisions, are only well-resolved at this grid spacing. Based on the second and to a lesser extent third hypothesis of this work, convective clustering affects dynamics. Divergence rates associated with convective heating increase non-linearly with the heating rate as convective systems grow. Therefore, it is needed to include non-linear increments of divergence rates with increasing intensity of convective systems into error growth studies, assisting these studies to extend all the way from the

convective to the planetary scales. Consequently, the conditional convective perspective shaped here can be connected with the Baumgart et al. (2019)-perspective.

## 8   Conclusions

The multivariate exploration of instantaneous divergent outflow strength of deep moist convection in real case weather prediction shows that their controlling processes are rather complex and cannot easily be distinguished and assigned to individual mechanisms. However, based on the analysis of a single convective event, major variability of the relationship between precipitation rate and upper-tropospheric divergent outflow rate is explained by effects that were also present in LES-analyses (Groot and Tost, 2023b). The following can be concluded on variability in upper-tropospheric divergent outflows from deep convection (as to this case study):

- The outflow is responsible for major ensemble spread in the divergent part of the upper-tropospheric wind during a convective event.

- Convection-permitting (1 km horizontal grid spacing) simulations represent the effect of aggregation on instantaneous divergent outflow rates from deep convection and substantial spread of divergent outflow rates exist at a given net latent heating rate.

- Using simulations at coarser resolution probably implies assuming a (near-)linear relationship between the outflow rate and net latent heating rate.

- Various indications show that the fingerprint of dimensionality is represented in variability of instantaneous convective outflow strength in ICON convection-permitting settings, but a case study comparing squall lines that highly resemble 2D-convection with less organised convection is needed to increase the confidence in this finding.

- Convective momentum transport seems to weakly affect this outflow strength directly.

- To understand convectively induced flow perturbations better, a separation into two components of convective variability is necessary: 1. variability in predicted mesoscale precipitation rates; 2. representation of the residual (conditional) flow perturbations, which depend on the cloud-scale dynamics.

The results of this work strongly suggest that the interactions between gravity waves emitted by heating of individual clouds is likely of prime importance for the representation of instantaneous divergent outflow rates from organised convection, which can successful be achieved at convection-permitting resolution. Additional case studies are needed to revisit the role of the dimensionality of convective overturning.

*Code and data availability.* The code used in this work and the output for one PER and one PAR simulation are available in Groot and Kuntze (2023) (last accessed: 16-02-2023).

## Appendix A:  Table of parameters in ellipse dataset

**Table A1.** List of parameters in the dataset of ellipse records, with their descriptions.

| Name of parameter | Unit | Explanation (if necessary) | Symbol |
|---|---|---|---|
| Ellipse id | # | Each ellipse comes with an id | |
| Time stamp | # 5 min | Each corresponding output time step has a time stamp | |
| Axis ratio | - | Ratio between major and minor axis of ellipse | A |
| Mean precipitation rate | mm/h | Mean surface precipitation rate over convective system's track following box | |
| Mean mass divergence | $\frac{kg}{m^3 s}$ | Mean upper tropospheric (380-180 hPa) mass divergence rate over convective system's track following box | |
| Div/precip ratio | $\frac{kg \cdot mm}{m^3 s^1 h^1}$ | Ratio between box mean mass divergence and box mean precipitation rate | D |
| U-component of CMT eddy flux | $m^2 s^{-2}$ | Mean value of $u'w'$ at model level 25, about 315 hPa, where $'$ indicates domain average relative velocity perturbations | |
| V-component of CMT eddy flux | $m^2 s^{-2}$ | Mean value of $v'w'$ at model level 25, about 315 hPa | |
| Absolute vertical integral of CMT acceleration | $m^2 s^{-2}$ | Mean value of $[(u'w')^2 + (v'w')^2]^{\frac{1}{2}}$ at model level 25, about 315 hPa | |
| Ratio absolute integral of CMT acceleration over precip rate | $\frac{m^2 h}{s^2 mm^1}$ | As above, but relative to precipitation rate in mm/h (also computed for U and V component separately) | C |
| Xcentre | grid cell # | Centre of the fitted ellipse in zonal direction | |
| Ycentre | grid cell # | Centre of the fitted ellipse in meridional direction | |
| Ellipse angle/orientation | ° | Orientation of the major axis of the ellipse with respect to a reference direction | O |
| Major axis length | km | | |
| Minor axis length | km | | |
| Ellipse area | # grid cell$^2$ | Area of the ellipse that has been fit, can be converted to square km (1 grid cell is approximately 4 km$^2$) | |
| Mean precipitation rate over ellipse | mm/h | Mean precipitation rate over ellipse only | |
| Area >10 mm/h precipitation rate | - | Fraction of ellipse exceeding 10 mm/h ("convective") precipitation | |
| Area >1 mm/h precipitation rate | - | Fraction of ellipse exceeding 1 mm/h ("stratiform + convective") precipitation | |
| X distance ellipse and box centre | km | Distance between box centre and fitted ellipse in x-direction | |
| Y distance ellipse and box centre | km | Distance between box centre and fitted ellipse in y-direction | |
| Total distance ellipse/box | km | Total distance obtained from its x- and y-component | |

## Appendix B: Table of two subsets in ellipse dataset

**Table B1.** Ellipse parameters associated with two subsets of the full dataset analyzed in Section 6.1 are sorted separately in the below table.

| Parameter | Full dataset | Subset low $D$ | Subset high $D$ |
|-----------|--------------|----------------|-----------------|
| $A_{mean}$ (-) | 0.56 | 0.54 | 0.60 |
| $O_{st.dev.}$ (°) | 44 | 32 | 45 |

*Author contributions.* EG (ICON-PAR) and PK (ICON-PER) carried out the simulations for this work, under the supervision of HT and AM. EG designed the study, developed the ellipse fitting algorithm, carried out the analysis and wrote the manuscript with contributions from all co-authors.

*Competing interests.* The authors have no competing interests to declare.

*Acknowledgements.* The research leading to these results has been done within the subprojects 'A1 - Multiscale analysis of the evolution of forecast uncertainty' and 'B1 - Microphysical uncertainties in hailstorms using statistical emulation and stochastic cloud physics' of the Transregional Collaborative Research Center SFB / TRR 165 'Waves to Weather' funded by the German Research Foundation (DFG). The authors would also like to acknowledge the computing time granted on the supercomputer MOGON 2 at Johannes Gutenberg-University Mainz (hpc.uni-mainz.de, last accessed: 02-02-2023).
We would also like to thank the reviewers for sharing their opinions on the manuscript and the (co-)editor(s) for handling the manuscript.

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
