# Peer review of "Divergent convective outflow in ICON deep convection-permitting and parameterised deep convection simulations"

_EGUsphere, 2023_

## Referee Comment (RC2)

Review of egusphere-2023-664

Summary:
This is a study about deep convective outflow and differences between simulations with resolved vs. parameterized convection. Overall, I don't think this paper has the quality needed to be a journal article. I have a difficult time understanding the motivation, the methodology and the results. The text leaves me with an overall impression of having been crafted without much care and suffers from muddled scientific writing. I do think there is some worthwhile science hiding, but it would need to be brought out by completely rewriting the manuscript. I'd also recommend having a professional editing service go over the manuscript to improve readability.

Recommendation:
Reject

Paper Strengths:
The experiment set up is interesting and, when the manuscript is much improved, the results warrant publication.

Major Comments:
1. Introduction: Why is the introduction centered around predictability? This isn't really the topic of the study. I recommend centering the introduction around properties of convection; or focus on the predictability aspect in your results. In a similar vein: you need to better motivate why convective outflow is worthy of study. I also don't really understand what you mean by "flow variability" in the introduction in the context of this manuscript. Sensitive dependence on initial conditions? Variability in terms of space and time? These are very different concepts. In summary, the introduction is very weak and doesn't fit the purpose of the paper in my opinion.
2. Methodology: You're tracking convective systems in the convection-permitting simulations, but you're calculating box averages in the simulations with parameterized convection. This is not a fair or consistent comparison and the results aren't really apples-to-apples comparisons. I recommend focusing on box averages also in the case of PER so you can better compare with PAR.
3. Presentation of figures: As one of the goals of the paper is to compare properties of convection-resolving and parameterized-convection runs, I'd recommend comparing PER and PAR in each individual figure. Fig. 5 is an example where you sort of do this, but in an awkward way (tracks of PAR in Fig. 5a, divergence of PER in Fig. 5b, divergence of PAR in Fig. 5c,d without a clear correspondence to Fig. 5b). A direct comparison between, such as directly comparing Fig. 5b with 5c, would help the reader immensely with digesting the results in terms of what the differences between PER and PAR are. Along similar lines, Fig. 6a and b are difficult to compare because of different axes and markers, making understanding difficult.
4. Sections 6 and 7 have the biggest potential to bring out some new findings, but in the current manuscript are rather short. I recommend expanding these sections and using them to "build" the paper. Important questions that can be asked are, for example, does

the difference in organization between PER and PAR lead to fundamental differences in CMT? How does this matter for predictability? Does it matter at all? This seems to be part of your motivation (L516: "This close connection may lead to perturbation growth in a forecast or spread in an ensemble")

5. Section 7: I am not sure how the "Dimensionality hypothesis" fits into this study.

6. Conclusions: Now it seems like you're talking about things that haven't been shown before…for example, "The outflow is responsible for major ensemble spread in the divergent part of the upper-tropospheric wind during a convective event." I don't find any figures showing ensemble spread? It seems buried somewhere but I can't find it. "Using simulations at coarser resolution probably implies assuming a (near-)linear relationship between the outflow and net latent heating." – Does this refer to Fig. 6b?

Minor Comments:
1. L1: What kind of event? This should be made clear (i.e., "convective event")
2. L26+: You should specifically state the hypotheses. It's unclear what you're after.
3. L30: "based on their linearised gravity wave adjustment model an idealised expression of outflow strength from deep convection was constructed (Nicholls et al., 1991)" – Not clear what this means.
4. L32: What do you mean by the "slope of dependency"?
5. L35: What is "Outflow dimensionality?"
6. L36: " idealised point ("3D") and idealised line ("2D") sources" – I don't understand this concept.
7. L39: "such behavior" – What behavior?
8. L134: What day were the PER simulations initialized?
9. L138: You should state somewhere before that this is an ensemble study.
10. L141: What is the "alternative surface tile dataset"?
11. L141: "20 member initial condition ensemble closely" — Does this refer to PER or PAR?
12. Methodology (L190+, conditioning on precipitation rate): Do you take into account that the precipitation reflects atmospheric processes from some time before the instantaneous CMT is calculated? What I mean by that is that precipitation takes time to form and to fall and therefore reflects convection from some time ago. Or is there some time averaging going on? Or do you compute precip rate over some time interval?
13. L214: "In past times" – Not only in the past, this is still mostly true today. All operational global NWP models as well as climate models use some form of cumulus parameterization.
14. L238+: Why do you reject features farther than 25 km from being a match? This seems like a very strict criterion given the low predictability of convective features.
15. L285: "PAR-profiles also reveal a strong divergence maximum directly underneath the tropopause (see supplementary material: Figure S5) – This figure should be in the paper as one of the main points is comparison between PER and PAR…you could make a two-panel figure for example.
16. Fig. 6a. The caption says "divergence-latent heating relationship", but the figure shows divergence vs. precip rate. Please reconcile.

Editorial Suggestions:
1. L2: no comma after "both"
2. L132: local area model
3. L194: delete "mostly"
4. L319: space is missing before "Consequently"

---

## Author Comment (AC1)

**Replies to all Referee Comments on "Divergent convective outflow in ICON deep convection-permitting and parameterised deep convection simulations. " [1]**

Edward Groot, Patrick Kuntze, Annette Miltenberger, Holger Tost

June 26, 2023

**1 General**

We thank both referees for reading our manuscript and formulating their summary and feedback [2, 3]. The manuscript will be revised drastically (especially the introduction), since the reviewers have noted that a lot of information is buried and convoluted: this is also clear from their written reviews, which means that for improved readability, the introduction is the starting point to make the entire manuscript more accessible. Furthermore, we will work on the existing analysis and work on the presentation of the results and discussion to more clearly state the main points. The new manuscript will be heavily revised along these lines.

In particular, we will add a figure illustrating the current conceptual understanding of divergent outflows from high resolution large eddy simulations, which will (hopefully) bring the reader in a more convenient way to the starting point: the hypotheses. Furthermore, another figure illustrating the workflow (and emphasizing the key points thereof) will hopefully enhance the clarity and structure of the methodology (Section 3.2, mostly 3.2.1). The presentation of the results and the discussion will be strongly restructured and rewritten, but mainly using the existing figures.

**2 Referee 1: reply to RC1 [2]**

1. *'" please make sure from the beginning, already in the Abstract that you are dealing with Ensembles (otherwise reader, seeing just one case would think this is not significant)"'*
We will emphasize this more in the updated manuscript.

2. *'"your sample size is still very small. From the modelling point of view it should not be much work to add more days - but dont know how much additonal work it is for your classification method. If you can do so, please add at least a few days with larger systems, this would be easily achievable for US systems or African squall lines with large and long-lived systems, but you can also add more days over Europe with prefrontal convection. You can still keep your main figures for Germany but for the scatter plots this would be extremely helpful. As it is I dont think you can quantitively talk about CMT or divergence in unorganized vs 2D squall line or Derecho type convection;*
*As said above best would be to add more samples (days in Europe US or Africa) and also to rewrite section 6 and adapt all Figure from Figure 6"'*
We agree with the reviewer that analysis of a larger variety of convective systems would be very interesting and would strengthen the insight gained. Unfortunately this involves a significant amount of computational and manual efforts, that we are not able to conduct within the scope of the present manuscript. In particular for our current analysis methodology, many manual analysis steps are required to process the data: the ellipse fitting process is automated, but the selection of convective systems is not. This means that one would have to go through a dimension of ensemble members and cases/regions/set-ups to compare, before the dataset can be finished. It may be possible to further automate those processes, but at this point this is not feasible. In the present manuscript we therefore want to focus on presenting the analysis methods and some first results as to the variability of upper-level convergence in a real-case NWP simulation.

For the future, we have several ideas for re-applying the method to a large-spectrum of cases. In the manuscript we already mention that an ensemble with even more strongly organised squall lines at mid-latitudes would be particularly suitable for the first further assessment. Furthermore, we have

other plans to further investigate (mesoscale) convective systems with our methodology, and would also encourage others to do so. However, it needs to be clear that just the modelling centers are likely have the datasets in their archives that could allow them to do a very comprehensive assessment of outflow climatologies.

3. *'" instead of using precipitation, could it be more useful to use low values of OLR or BT to charcaterize outflow, this might produce smoother results and better identify the upper level circulation?"'*
In previous work on large eddy simulations [4], we have discussed that the upper tropospheric divergence is produced right at the location of the precipitation cell under conditions of storm relative winds that do not slant/tilt the convective system overly. In other words, the anvil (or OLR fingerprint) is not the first feature to map, when investigating the divergent outflows, but the updraft location is. In this work, rectangular horizontal masks of integration have been set such that they explicitly match the precipitation cells AND the divergent wind source, namely the region of divergent acceleration of the flow. The divergent winds produce the outflow and the divergent wind source does not move along with this flow. However, slanting/tilting of convective updrafts in the model can have an effect on the displacement of the outflow, with respect to surface precipitation. This effect of tilting has been taken into account by choosing a sufficiently large mask for the computation of divergence and surface precipitation(sometimes important for ICON-PAR indeed: as the high percentiles of vertical velocities at the 13 km grid-scale are much lower [i.e. up to 1-2 m/s] than in ICON-PER [sometimes 10 m/s], this contrast results in horizontal large horizontal displacements in ICON-PAR during the "convective" overturn). It has been discussed in [4] that if we extend the mask too far beyond the core area of precipitation, that subsidence is probably going to compensate for some of the divergent winds.

However, we do think that OLR can have an additional small impact on the divergence (less than 20%). But the leading order terms will be the precipitation (or, more specifically, its latent heating pattern) and convective organisation and aggregation. So there could be very small benefits of accounting for long wave radiation. Our point is however that, firstly, the main diabatic source of divergence, and the heating rate plus the role of organised convection therein, should be quantified, before much smaller terms are accounted for.
In summary, defining the extent of the convective cells by using OLR has intentionally not been done to avoid the near cloud subsidence effects and concentrate on the divergence of the main cells, which are better characterised by the precipitation latent heating and therefore the precipitation field itself.

4. *'" I am unsure it is helpful to normalize momentum transport by precipitation, yes it is natural to normalize divergence by heating but momentum transport should be normalized by shear"; '"l12-13 "as opposed to expectations, CMT". I think this part also in result section should be revised or could be dropped in absence of sufficient cases/samples with shear;"' and '"l195-200 normalize CMT with precipiation, see main comments"'*
The hypothesis that CMT does not have a significant impact on upper-level divergence is currently based on LES results. We therefore believe it is a valid / interesting point to quantify the relations of CMT and upper-level divergence in a real-case scenario. We therefore keep this section but more clearly state that a larger sample of convective systems is need to firm up conclusions.

The choice was made was based on

- The work of [4] (no direct impact of convective momentum transport on the magnitudes of divergent outflow) and
- the fact that we investigate one case, where shear profiles are very similar to each other (a domain of only a few hundred by a few hundred kilometers and less than half a day are covered).

In Figure S3 of the supplement it can be seen that the CMT diagnostic and precipitation rate are strongly correlated. Therefore, to see potential impacts on the conditional mass divergence/outflow that might exist (we start with the hypothesis that there is none, based on [4]), the co-variability of the CMT diagnostic and precipitation rate has to be removed: no precipitation results in no CMT.
Now, we can test the effect of CMT on divergence with the null hypothesis that there is none. And given the statistical coherence and strong correlation, this null hypothesis has to be rejected. However, future inspection of more extensive datasets might change the conclusion and this will be discussed explicitly.

5. *'"I found section 6 of your manuscript not convincing and not well written, please redo"'*
   We will work on strong modifications of the sections 5 and 6, following up on the lines of modification in the introduction and methods section of the paper. The new results sections will tie in much more closely with the conceptual perspective to be introduced in the introduction.

6. *'" also in Figure 6a the divergence is so scattered, in the explicit run, corresponding to small scale systems="noise" in Figure 5b that I dont think it makes sense to discuss linear regressions (couloured lines) as a function of lead time or system evolution. In that sense it would even make more sense to discuss better the parametrized runs with larger outflow"'*
   The regression analysis is not applied to suggest any form of linear relation here. It is used to obtain designated statistical diagnostics for the fact that we find the green, red and blue systems in Figure 6 at the lower end of the distribution of divergence to precipitation rate ratio D. The diagnostics provide a quantitative interpretation of where the three convective systems are located in the divergence-precipitation rate space.
   Furthermore, we apply a low-pass filter to the windfields from the convection-permitting simulation (removing all wavelengths smaller than 45 km) to remove the gravity wave "noise". Thereby, the convection-permitting and convection-parameterised simulations obtain roughly the same effective resolution in the divergence field. After this operation, the divergent outflows are well intercomparable and there is no problem of small-scale noise in convection-permitting simulations (lacking in the parameterised configuration).

7. *'"Specific points: l12-13 "as opposed to expectations, CMT". I think this part also in result section should be revised or could be dropped in absence of sufficient cases/samples with shear;"' and '"l195-200 normalize CMT with precipiation, see main comments"'"'*
   We refer to the answer above, on CMT, point #4 in this reply.

8. *'l19 "initial condition flow variability". In you manuscript you actually use physics perturbation"'*
   Yes, both are used. The physics perturbations are used to further confirm earlier results [4]. The new element is the initial condition perturbations, in the convection-permitting set up: we want to focus at the role of initial condition variability. This is also where the hypotheses of the work are focused on. Indeed ideally both types should be mentioned here, even if we think one of them is more relevant, for instance, to predictability and the novel elements of the manuscript than the other.

9. *'" l76-86 you formulate here 3 hypotheses. you might drop some of these in absence of larger samples and systems, as the main (also expected outcome) is a broadly linear relationship of divergent outflow and precipitation or better mass flux which should also hold for larger area (opposed to individual system)"'*
   The convective aggregation mechanism [4] shows that the linear relationship does not effectively hold at high precipitation intensities in LES. The work here confirms that the same is found for convection-permitting simulations. The simple linear relationship holds for parameterised deep convection only.

   We may have to substantially reformulate and repeat some of the points on the significance of the convective aggregation mechanism of our previous studies to make it more clear, that the aggregation can lead to a non-linear mean divergence - mean precipitation rate relationship in convection-permitting simulations.

   By revising the introduction drastically, and the rest of the manuscript accordingly, we hope to address the shortcoming of buried main results in the manuscript. This should lead to a clearer overview of the main points and how to find them, in terms of scientific analysis.

10. *'" l188 "satellite systems" avoid this expression as "satellitee" misleading here"'*
    We will change the wording. We mean accompanying, smaller, convective clouds of the main systems.

11. *'" l195 "vertical integral of CMT over all levels below". Not sure this is intended as over the whole troposphere CMT integrates to zero so in that case if you decide to keep CMT you might need to consider upper and lower troposphere separately "'*
    This is exactly what we intended to do, but it seems that our formulation did not pin it properly to

the point. We split up the troposphere into the part of the lower- and mid-troposphere (below 315 hPa) and the upper troposphere and integrate over the two parts separately.

12. *"l227 you migth add reference to https://doi.org/10.1002/qj.4185 here as actually the organisation is not so simple and sometimes too spotty in explicit compared to parametrized convection Also you might add refernce in the text to studies that explained that the convective outflow only has an effect to the circulation and subsequent predictability if the outflow occurs in the vicinity of the jet"'*

Thanks for your suggestion. We will consider adding this reference along with some discussion of the potential impact of organisation biases. However, we would also like to use this opportunity to point out that organisation biases in itself are not a problem: we aim to show with diagnostics which processes play a role. In other words, the first question is which processes are acting and are represented in the model. The follow-up question would consequently be, whether there is a bias or not compared to the real-world in the representation or activity of the processes ICON, and possibly other models. We focus on the first question in this work.

13. *"' l297 "increase over time" this looks like the diurnal cycle evolution"'*

The diurnal cycle is not the main actor, as we deal with a strongly dynamically forced case of convection. However, of course we cannot rule out that the diurnal cycle might have an impact. Convective organisation and aggregation is influenced by the diurnal cycle, but any organisational type of convection could in principle occur throughout the diurnal cycle. The life time of convective clouds could be of more significance, as it may be suggested by our results (Figures 4 and 6a in the initial submission, although this remains merely speculative). Based on the available evidence, we think that the diurnal cycle is of minor relevance compared to the lifecycle of the convective cases in the considered case.

14. *"' Figure 5: add (a), use same legend bar for (b) as for (c) and (d). Here and in Figure use everywhere units of [1/s] for divergence (not multiplied by density in some places, eg Figure 6) and scale legend bars by 1.e5"'* Figure 5 cannot use the same color bars for each panel, as parameterised deep convection and convection-permitting simulations have very different amplitudes in their divergence signal (which would make the patterns in one of the two configurations invisible). The convection-permitting divergence in Figure 5 (b) is still unfiltered (before the filter operation, applied to obtain figure 6 for convection-permitting configuration) and hence much stronger, but we focus here on the spatial distribution of strong ensemble variability. It does not matter whether it is in mass units or in inverse time units in Figure 5: we look at 250 hPa and the density is nearly the same everywhere.

However, once we want to compare outflow strengths, it is important to understand how much mass is transported in each case. Therefore, we need mass/density units in Figure 6!

This will be discussed explicitly in the updated manuscript.

15. *"' Figure 6a, coloured slopes not convincing. Figure 6 b more convincing, but why is in Figure 6b shallow (magenta) so different form no convection run (organge)?"'*

The coloured lines in Figure 6a a show the evolution of individual convective systems in the shown parameter space. In that respect we are not sure, what the issue of the reviewer is as we cannot change the data-base. We will make it more clear what the colored lines represent in the revised manuscript. In Figure 6b the magenta data points are off, because they have a deep convection parameterisation switched off and the outflow is created from shallow convection instead, maximizing near 500-600 hPa. Therefore, this physics perturbation is slightly unrealistic, but it shows that the outflow levels can be altered by this unrealistic setting of the parameterization (e.g. probably as a result of unrealistic parameterisation mixing and entrainment processes and cloud system geometry, but this remains speculative).

16. *"' l439-440 Why do you say "larger-scale flow feedback in PAR likely not accurately represented"? Isn't supported by anything in manuscript and how do you know that PER effect is not overestimated etc? "'*

It is found that the ICON-PAR simulations only follow the upper range of mass divergence to precipitation rate ratio ($D$ in our work) of the convection-permitting ICON, which initially occurs at low precipitation rates for the convection-permitting configuration (in this comparison, a correction for differences in outflow layer thickness is considered to make both configurations comparable!).

[4] shows that collisions of divergent outflows cause sub-linear increases of divergent outflow with increasing precipitation intensities, in LES. Here, it is investigated whether outflow collision effects are also detected in the patterns of deep convective outflow intensities in ICON. The hypothesis on the

presence of this convective aggregation mechanism on divergent outflow variability is supported by our data and shows that the divergent outflow increases with precipitation rates in a sub-linear fashion, towards high precipitation intensities. The same is found in LES [4]. The ratio $D$ in the linear dependence for the parameterised deep convection configuration in ICON overlaps with the high range of divergent outflows (and $D$, there) in the LES configuration. Nevertheless, both the convection-permitting ICON and the LES do not maintain these high values of $D$ up to higher precipitation intensities: the values of $D$ decrease at these precipitation rates.

Therefore, since $D$ in ICON-PAR turns out to be practically constant (linear relationship of precipitation rate/divergence), both LES and convection-permitting ICON analyses suggest that the diabatic warming by precipitation "pushes" too strong outflows in ICON-PAR (when statistically averaged over areas of high and low precipitation intensities): constant $D$ up to high precipitation rates are not found when convection is resolved better, so the high $D$ suggests a positive outflow bias. Therefore, both LES and convection-permitting ICON suggest that in ICON-PAR a too strong feedback from convective clouds towards the larger-scale upper-tropospheric flow is represented at high precipitation rates.

This effect is described in this paragraph and its description will be improved in the revised manuscript (along the lines of the discussion here).

**3 Referee 2: reply to RC2 [3]**

17. *'"Major Comments: 1. Introduction: Why is the introduction centered around predictability? This isn't really the topic of the study. I recommend centering the introduction around properties of convection; or focus on the predictability aspect in your results. In a similar vein: you need to better motivate why convective outflow is worthy of study. I also don't really understand what you mean by "flow variability" in the introduction in the context of this manuscript. Sensitive dependence on initial conditions? Variability in terms of space and time? "'*

We agree that the choice made for the focus in the introduction is not the best possible choice. However, we have had the following reasons to use predictability as starting point:

- This has been done because predictability is the motivation to do our investigation.

- The authors do not fully agree that sensitive dependence on initial conditions and variability in space and time are completely unrelated with regards to the posed hypotheses. Since different modes of convective organisation may occur in close proximity during some weather events of e.g. highly organised deep convection (e.g. supercells and squall lines), divergent outflows may also vary very strongly in close proximity. And building supercells versus very long squall lines are suggested to end up in very different $D$ regimes in our work, based on [4]. The ICON simulations also reveal some variability in terms of convective organisation, as needed for the investigation. Although strong very extensive squall lines of hundreds of km do not occur (see discussion, Section 6, of the initial submission). As a consequence, small changes that create a supercell at location x or a nearby long squall line at location y may push the upper atmospheric flow differently.

  It is along this line of thinking that the concepts of flow variability as a result of initial condition perturbations or in space and time all get related to the strength and sensitive timing of convective aggregation and storm geometries, as suggested in [4].

  It is exactly here that differential outflow could affect predictability of the flow through the two main mechanisms, the first and second hypothesis.

In the revised manuscript, we would like to move the focus towards the convective organisation, representation and the conceptual understanding from previous works needed as a starting point for our hypotheses. With the latter, we would tie the work much more strongly to the previous works that are the most relevant. Furthermore, we will aim at a much more reader-friendly manuscript structure. However, as this manuscript does not deal with the impact of the convection on the flow-variability itself, the rewritten introduction will mention it mainly as a motivation and shift the focus more towards properties of convection as suggested by the reviewer.

18. *'" 2. Methodology: You're tracking convective systems in the convection-permitting simulations, but you're calculating box averages in the simulations with parameterized convection. This is not a fair or consistent comparison and the results aren't really apples-to-apples comparisons. I recommend focusing on box averages also in the case of PER so you can better compare with PAR."'*

We aim at modifying the methodology section in a way that this point will become more clear: In

both simulations box averages are calculated, and the difference in effective resolution is corrected for. Hence the comparison is fair and consistent. The tracking happens in parallel to the boxes, to fit ellipses and estimate geometry of the convective systems in the convection-permitting simulations. Additionally, we plan to add a workflow-type of figure to described the analysis method in more detail to improve its clarity and the revised manuscript will be modified accordingly.

19. *'" 3. Presentation of figures: As one of the goals of the paper is to compare properties of convection-resolving and parameterized-convection runs, I'd recommend comparing PER and PAR in each individual figure. Fig. 5 is an example where you sort of do this, but in an awkward way (tracks of PAR in Fig. 5a, divergence of PER in Fig. 5b, divergence of PAR in Fig. 5c,d without a clear correspondence to Fig. 5b). A direct comparison between, such as directly comparing Fig. 5b with 5c, would help the reader immensely with digesting the results in terms of what the differences between PER and PAR are. Along similar lines, Fig. 6a and b are difficult to compare because of different axes and markers, making understanding difficult."'*

We agree that the differences between the panels should be clarified and we will improve this in the revised manuscript (see also points #14 and 15 in this reply, to the first reviewer).

Similarly, Figures 6a and 6b are integrated over different vertical levels, because the outflow altitude varies between the two panels (thickness of 200 hPa in convection-permitting runs vs. 250-260 hPa in parameterised configuration!). Therefore, the comparison in one panel would be unfair and suggest conclusions that are not justified. To avoid the suggestion, the two configurations are separated into two plots. We would like to point out this motivation to outline the figure 6 as it was, in the initial submission.

We will work on a clever solution, e.g. to include a line for a weighted average of ICON-PAR in the ICON-PER plot, or some addition along similar lines. This would further assist bringing across the point of Figure 6.

20. *'" 4. Sections 6 and 7 have the biggest potential to bring out some new findings, but in the current manuscript are rather short. I recommend expanding these sections and using them to "build" the paper. Important questions that can be asked are, for example, does the difference in organization between PER and PAR lead to fundamental differences in CMT? How does this matter for predictability? Does it matter at all? This seems to be part of your motivation (L516: "This close connection may lead to perturbation growth in a forecast or spread in an ensemble")"'*

Thank you for these insightful suggestions. We will consider them in the revision process. We think that this point/opportunity will (at least partially) be addressed automatically when we try to create a strong connection between the updated introduction and the updated discussion section, when the connections between acceleration mechanisms, convective aggregation and organisation, the outflow divergence and potential effects on flow predictability are explained in a considerate, clear and accessible way across the updated manuscript.

21. *'" 5. Section 7: I am not sure how the "Dimensionality hypothesis" fits into this study."'*

The "dimensionality hypothesis" comes into play, because we study the variability in divergent outflows from convection and we have shown that this variability can largely be explained by the dimensionality of the convection [4]. However, we have not sufficiently well clarified the connection to our earlier work. This will be improved in a revised version. In particular part of the introduction will be dedicated to explaining theses concepts, providing better context and thereby improve the presentation of the scope of this work.

Part of the study design is that the "dimensionality hypothesis" can (presumably) be thoroughly represented in convection-permitting configurations, which is why we zoom in on the convection-permitting simulations in section 5. However, it is not present in the configuration with convection parameterisation. We agree, that in the initial manuscript version this may not be sufficiently clear to readers, since the connection to [4] was not made clear enough. Therefore, it was difficult to understand the main concepts.

22. *'" 6. Conclusions: Now it seems like you're talking about things that haven't been shown before...for example, "The outflow is responsible for major ensemble spread in the divergent part of the upper-tropospheric wind during a convective event." I don't find any figures showing ensemble spread? It seems buried somewhere but I can't find it. "Using simulations at coarser resolution probably implies assuming a (near-)linear relationship between the outflow and net latent heating." – Does this refer to*

*Fig. 6b? "'*
1) This is revealed by the combination of 250 hPa divergence and precipitation patterns (also, more details on the parameterised simulations regarding this aspect are found in the supplement). It is just in the ensemble spread of our very specific setup, where divergence in Figure 5b, c, and d reveals this spatial pattern, closely connected to the precipitation systems. Large spread in divergence at 250 hPa spatially coincides with the proximity of strong deep convection. They confirm the earlier suggestions based on other studies. 2) This does refer to the difference between Figures 6a and 6b indeed: in a parameterised convection configuration, there is little spread on top of the linear divergence-precipitation rate relationship. Even if we have a "convection-permitting" configuration in at 13 km, this turns out to not change. However, once cloud scale processes and gravity waves are represented better (1 km resolution), we get a large envelope of variability on top of the (here: somewhat non-linear) relation between precipitation rate and mass divergence.

We will clarify both of the main points, mentioned above, as key findings, after the revisions.

The conclusions will be rephrased to make the connection to the results more explicit / obvious.

23. *"' Minor Comments: 2. L26+: You should specifically state the hypotheses. It's unclear what you're after. "'*

They are stated in lines 76-87 of the initial submission. But after a revision, we will carefully make sure that the hypotheses move to a very prominent place and that the introduction of the conceptual ideas behind those hypotheses is thoroughly explained.

24. *"' 3. L30: "based on their linearised gravity wave adjustment model an idealised expression of outflow strength from deep convection was constructed (Nicholls et al., 1991)" – Not clear what this means.*
*4. L32: What do you mean by the "slope of dependency"?*
*5. L35: What is "Outflow dimensionality?"*
*6. L36: " idealised point ("3D") and idealised line ("2D") sources" – I don't understand this concept."'*

As mentioned above in these replies, the conceptual model based on our earlier work [4] will be to be added in thoroughly, with a neat explanation. We agree with the reviewers that this was not done in a sufficiently clear manner in the current manuscript version, which strongly reduced the understandability of the presented research and its scope.

25. *"' 1. L1: What kind of event? This should be made clear (i.e., "convective event")*
*7. L39: "such behavior" – What behavior?*
*8. L134: What day were the PER simulations initialized?*
*9. L138: You should state somewhere before that this is an ensemble study.*
*10. L141: What is the "alternative surface tile dataset"?*
*11. L141: "20 member initial condition ensemble closely" — Does this refer to PER or PAR?"'*
*16. Fig. 6a. The caption says "divergence-latent heating relationship", but the figure shows divergence vs. precip rate. Please reconcile.*

We thank the reviewer for these suggestions; all points will be clarified in the revised manuscript.

26. *"' 12. Methodology (L190+, conditioning on precipitation rate): Do you take into account that the precipitation reflects atmospheric processes from some time before the instantaneous CMT is calculated? What I mean by that is that precipitation takes time to form and to fall and therefore reflects convection from some time ago. Or is there some time averaging going on? Or do you compute precip rate over some time interval?"'*

Thanks for rising this important point. We are aware of this issue. However, in our LES-studies [4, 5] it has been shown that this lag is not of substantial concern (see also the discussion section of [4]).

The output of precipitation (rate) and the winds is available at intervals of 5 minutes (ICON-PER) and 10 minutes (ICON-PAR) and is computed based on the precipitation sum. Hence, the differential of the precipitation sum was taken at the mentioned intervals. With respect to the winds, the output is available at the same frequencies: 5 minutes (ICON-PER) and 10 minutes (ICON-PAR). To further clarify, the dataset of convective systems covers the first +17 hours lead time (starting at +7 hours) of simulations in both ICON-PAR and ICON-PER.

27. *"' 13. L214: "In past times" – Not only in the past, this is still mostly true today. All operational global NWP models as well as climate models use some form of cumulus parameterization."'*

We agree with the reviewer that convection parameterisation are still (and will remain for some time)

a crucial component of numerical weather prediction and climate models. However, we wanted to highlight that at least for regional NWP there is a strong push towards kilometer-scale models, that do not require a deep convection parameterisation anymore. We will rephrase the sentence to make this more clear.

28. *'" 14. L238+: Why do you reject features farther than 25 km from being a match? This seems like a very strict criterion given the low predictability of convective features."'*
Here we do not refer to matches of convective systems between different ensemble members. The "matches" are between the centers of the boxes (used from the volume integration of divergence and corresponding surface precipitation) and those of the detected "convective precipitation" ellipses in the same simulation. The quality of matches is conditional on the ability of a computer vision algorithm to detect the ellipse correctly and consistently (as it is assured by validation algorithms). Essentially, the center of the convective system X in ensemble Y has to match only with itself - according to a center detection by the computer vision algorithm and one of a moving box. However, the "centers" of both metrics could differ and are allowed to differ by a certain number of grid cells, or equivalently, a specific distance in kilometers. This will be more clearly stated in a revised version.

29. *'" 15. L285: "PAR-profiles also reveal a strong divergence maximum directly underneath the tropopause (see supplementary material: Figure S5) – This figure should be in the paper as one of the main points is comparison between PER and PAR...you could make a two-panel figure for example."'*
Thank you for the suggestion. As, in contrast to the ICON-PER results, there is no time evolution in the profiles evident for ICON-PAR we did consider these less interesting. However, we will re-consider the prior decision to put them in the supplement.

30. *'" Editorial Suggestions: 1. L2: no comma after "both" 2. L132: local area model 3. L194: delete "mostly" 4. L319: space is missing before "Consequently" "'*
We thank the reviewer for the suggestions and hope that we will reduce the editorial, grammatical and sentence structure errors in the revised version. Additionally, in case of a final publication, professional copy-editing by Copernicus will also be conducted.

**References**

[1] Edward Groot, Patrick Kuntze, Annette Miltenberger, and Holger Tost. Divergent convective outflow in icon deep convection-permitting and parameterised deep convection simulations. *EGUsphere*, 2023:1–32, 2023.

[2] Referee comment 1 on "divergent convective outflow in icon deep convection-permitting and parameterised deep convection simulations.", 2023.

[3] Referee comment 2 on "divergent convective outflow in icon deep convection-permitting and parameterised deep convection simulations.", 2023.

[4] Edward Groot and Holger Tost. Divergent convective outflow in large-eddy simulations. *Atmospheric Chemistry and Physics*, 23:6065–6081, 2023.

[5] Edward Groot and Holger Tost. Analysis of variability in divergence and turn-over induced by three idealized convective systems with a 3d cloud resolving model. *Atmospheric Chemistry and Physics Discussions*, pages 1–23, 2020.

---

## Author Response (AR1)

**Replies EGUsphere 2023-664**

Edward Groot, Patrick Kuntze, Annette Miltenberger, Holger Tost

November 15, 2023

For further reference, we refer to the earlier author comment, which can be found via `https://doi.org/10.5194/egusphere-2023-664-AC1`.

---

## Referee Report (RR1)

**Review of "Divergent convective outflow in ICON deep convection-permitting and parameterised deep convection simulations", by Edward Groot, Patrick Kuntze, Annette Miltenberger, and Holger Tost.**

January 8, 2024

I find this article a very interesting read and I hope that it is published.

The relation between net latent heating (quantified as the surface precipitation rate) and divergent outflow in the upper troposphere is investigated, using ICON simulations of a real case of severe thunderstorms over Southern Germany. This builds on the author's idealised LES study already published.

Consistent with the earlier LES work, the present study finds that upper-level divergent outflow increases with latent heating, but increases more slowly as the organisation / clustering of the convective updrafts increases. This effect is found in 1km resolution simulations, but is absent from simulations at 13km resolution. It is suggested that resolving small-scale gravity-waves is key to representing the relative slowing of upper-level divergence with increasing convective organisation. The relation between upper-tropospheric outflow and convective momentum transport is also investigated.

However I have a number of fundamental questions about the theory, methodology and how the results are interpreted. I hope that in answering these questions, the paper can be made clearer and stronger. In its current form, it is not clear to me that all of the stated conclusions are proven by the results shown. So I would recommend significant revisions, if that is still possible at this stage?

I also note that this paper was originally submitted 9 months ago, so has been held up for a long time already (I understand one of the previous reviewers is now unavailable, which is why I have been invited to review the article so late in the process). Many apologies if my queries introduce further delay!

A breif summary of my main concerns:

- The divergence profiles shown in figure 6 do not vertically-integrate to zero for the 1km simulations, implying there is strong net convergence of mass into the defined boxes (see my point 6 below for detail). Whilst the model may solve the compressible equations, the magnitude of this mass

imbalance does not seem realistic, which makes me worry that there has been mistake in the analysis?

- The method of defining regions of interest and calculating the mean upper-level divergence differs between the 1km explicit versus 13km parameterised simulations, so it is not a like-for-like comparison (the text implies that the 1km run divergence field was first filtered onto a horizontal scale significantly coarser than the resolution of the parameterised run, yet no filtering was applied to the parameterised run for consistency; also much larger boxes were used in the parameterised run).

- In various places the paper claims that the dependence of outflow strength on convective organisation cannot be represented when using parameterised convection. But the results don't support this conclusion. The organisation quantified using the fitting of ellipses to the precipitation field is largely on scales that are resolvable in the parameterised convection simulation. Any resulting gravity-waves connected with the mesoscale heating structure would similarly be resolvable. And the results in figure 8b show that turning off the convection parameterisation (while keeping all else equal) made no difference to the divergence-precipitation relation.

Below is a more-detailed full list of queries...

1. Introduction: ∼ L20 and onwards.

   A key argument of this paper is that the amount of upper-level divergent outflow driven by a given amount of latent heating within convective clouds varies as a function of storm horizontal structure. However, surely the total (time-integrated) amount of ascent (and hence upper-level divergence) is constrained to be that required to restore the heated air to neutral buoyancy? This is the basis of "Weak Temperature Gradient" scaling argument, which is key to the dynamics of moist convection and its interaction with the stably-stratified free troposphere. Latent heating will temporarily make the cloudy updraft air buoyant relative to its surroundings; it then ascends relative to its stably-stratified surroundings until it reaches its neutral buoyancy level, and the amount of upper-level divergence forced by this must be constrained by mass-continuity. If the time-integrated upper-level divergence was less than this constraint, then the cloud column would remain buoyant after ascent had stopped. This should only be possible if the resulting warm anomaly is held in place by a rotationally-balanced flow, which will not occur at convective scales.

   I think the paper's findings can be reconciled with this argument because it actually considers the *transient* response of the upper-level divergence to the latent heating, not the total *time-integrated* response. We would expect the upper-level divergence to continue after the latent heating has stopped, until the warm anomaly is removed, but this final stage of the outflow is not considered in this study.

So, the authors find significant variation in the transient *rate* at which the upper-level divergence spins-up in response to latent heating, as a function of storm structure. This is a very interesting result, especially the discrepency of this between different model resolutions.

I would just ask the authors to clarify that their results apply for the transient response and not the time-integrated total response of upper-level divergence to latent heating.

e.g. replace the often repeated term "amount of upper tropospheric divergence" with something like "instantaneous rate of upper tropospheric divergence"?

2. One factor not considered in this paper is the influence of the vertical structure of the latent heating profile on the mass divergence rate. If heating is concentrated in a layer of the troposphere with higher (lower) static stability, then less (more) ascent is required to restore the heated air to neutral buoyancy, and hence less (more) upper-level divergence will result. One possible hypothesis that might be worth checking is whether part of the variation in upper-level divergence per unit latent heating can be explained in this way by variation in the vertical structure of the heating profile, rather than storm horizontal morphology. It might be that the coarse simulations have deficient variability in the vertical structure, and hence give less variability in outflow strength.

3. L136: Why would it not be possible to represent the impact of storm morphology on upper-level divergence patterns in a simulation at 13km resolution with parameterised convection? Presumably the gravity-wave dynamics thought to mediate this impact could be at least partially resolved at this resolution, and represented by the model's dynamical core rather than the convection parameterisation? I similarly query the discussion at L439-440: "little or no information on geometry of the convective systems can be represented with a parameterisation"; surely much of the geometry (as quantified by the ellipse fitting in this study) is mesoscale, so occurs on the resolved-scale in the 13km grid PAR simulations. So the gravity-wave response to the storm structure could be captured. The results shown in figure 8b show that the divergence-precipitation relationship is not sensitive to the choice of parameterised versus explicit convection at 13km resolution, so the paper's argument that the overly-linear response in PAR is due to the convection parameterisation (e.g. L680, L710) is not supported.

It might be that the PAR simulation simply fails to produce enough emergent mesoscale organisation on the grid (even though it could be resolved). However, figure 7(c,d) appears to show plenty of mesoscale structure in the divergence field for the PAR simulation.

4. L192: It is stated that the divergence fields in the PER simulation are low-pass-filtered to remove scales below 45km; is the same low-pass filtering also applied to the divergence in the PAR simulations, to ensure fair comparison? Since PAR has a grid-size of 13km, it may have some variability at scales smaller than 45km, so should be filtered in the same way as the PER simulations?

Also, why is it nessecary to low-pass-filter the fields first, given that the analysis only considers the divergence spatially averaged over large boxes anyway? Is there a danger that the low-pass filtering may spread some of the divergence across the box boundaries, so that it is spuriously missed in the analysis?

5. L298 - 305: I am struggling to understand this paragraph (describing the identification of the convective systems in the PAR simulations); consider clarifying? It says that "corresponding precipitation moves together with conditionally unstable or lifted air masses" and "A typical (mesoscale) convective system is easily contained within a box of several to tens of grid cells in each horizontal direction for ICON-PAR" This seems to say that the PAR simulation should produce well-resolved regions of precipitation which move with the flow, but then the text says that only 3 static boxes are used to define the convective systems in PAR. Why not define moving boxes to track the systems, in exactly the same way as is done for the PER simulations?

6. Figure 6: Do the profiles of ensemble-mean box-mean divergence satisfy mass continuity?? In ICON-PER, it looks as though the vertically-integrated lower-troposheric convergence is significantly greater than the upper-tropospheric divergence. Assuming the rate of change in the mass of air contained in the box is small, we should have:

$$\int_0^{z_{top}} \rho \left( \frac{\partial u}{\partial x} + \frac{\partial v}{\partial y} \right) \mathrm{d}z = 0$$

and assuming small deviation from hydrostatic balance ($\mathrm{d}p = -\rho g \, \mathrm{d}z$):

$$\int_{p_{surf}}^{p_{top}} \left( \frac{\partial u}{\partial x} + \frac{\partial v}{\partial y} \right) \mathrm{d}p = 0$$

But if I try to vertically-integrate the divergence in figure 6 over the y-axis pressure coordinate "by eye", it does not balance and there is strong net convergence into the box (especially at earlier times). e.g. considering the time-averaged profile for PER (fig 6b, black line), we have convergence between 900 and 400 hPa averaging around $0.4 \times 10^{-4}$ s$^{-1}$, giving:

$5 \times 10^4$ Pa $\times 4 \times 10^{-5}$ s$^{-1}/g \approx 0.2$ kg m$^{-2}$ s$^{-1}$

And we have divergence between 400 and 180 hPa averaging around 0.5 $\times 10^{-4}$ s$^{-1}$, giving:

$2.2 \times 10^4$ Pa $\times 5 \times 10^{-5}$ s$^{-1}/g \approx 0.1$ kg m$^{-2}$ s$^{-1}$

So the convergence and divergence seem to be out of balance by about a factor of 2. The mass imbalance of the order 0.1 kg m$^{-2}$ s$^{-1}$ would increase the mass of air in the box by $\sim 25\%$ during the 7 hour period shown, which is clearly not realistic.

I can't see this imbalance in the divergence profiles for the PAR simulation (fig 6b, green line, has similar upper tropospheric divergence to PER, but much weaker lower-tropospheric convergence, so looks in-balance). Similar for the PAR divergence profiles in figure S5.

Does this suggest there is something wrong, either with the model or the analysis method in PER? Could the discrepency in PER be somehow due to the low-pass filtering?

7. Comparing figure 4 with figure 7, it appears that the boxes used to define the convective systems (and compute spatially-averaged divergence and precip rate) have quite different size in PER and PAR, so it is not a fair comparison? The example box in PER shown in figure 4 appears to be a square with side length about 1 degree in latitude ($\sim 110$ km), whereas the boxes for PAR shown in figure 7 are rectangles with side lengths 2-4 degrees. The use of systematically larger boxes in PAR might risk including more of the background subsidence within the box, so that the spatially-integrated divergence is reduced?

8. L393 / figure 8: The text says that "from here on vertically integrated values (of upper-level mass divergence) are used". But the quantity plotted on the y-axis says it has units kg m$^{-3}$ s$^{-1}$, which is presumably just $\rho \left( \frac{\partial u}{\partial x} + \frac{\partial v}{\partial y} \right)$? Shouldn't the units be kg m$^{-2}$ s$^{-1}$ if this has been vertically-integrated?

Also the lines shown in the figure for comparing PAR with PER (and the discussion around L435) indicate a correction to the divergence to account for the different thicknesses of the outflow layers, but this should not be needed if vertical integrals were plotted as suggested in the text.

I would recommend calculating the vertical integral of mass divergence for both PER and PAR and comparing those:

$$\int_0^{z_{top}} L(z) \rho \left( \frac{\partial u}{\partial x} + \frac{\partial v}{\partial y} \right) \mathrm{d}z$$

where $L(z)$ is 1 in the outflow layer, zero at other heights. The result then has units kg m$^{-2}$ s$^{-1}$, which can be compared more directly with precipitation rate (which can be expressed in the same units!)

Also Re the discussion around L444 (where the text suggests that the apparent reduction in divergence-precipitation ratio for the 13km simulation with only shallow convection parameterised is due to the outflow layer extending lower-down); this could easily be remedied by plotting the above vertically-integrated divergence, and setting the outflow-layer mask

$L(z)$ interactively so that it always captures all model-levels within the time-varying outflow layer depth (instead of setting $L(z)$ to zero below a fixed height).

9. L537, L580, section 7.2.2 (relation between CMT and outflow strength); The study finds a positive correlation between momentum flux and upper-level divergence (with both normalised by precipitation rate). From this, they conclude that convective momentum transport has some impact on the upper-level divergence. However, correlation does not imply causality. It might be that upper-level divergence (or some aspect of storm structure which is correlated with it) has modified the convective momentum transport? e.g. more organised convective systems may have quite different pressure gradient forces acting on the horizontal momentum of the updrafts, and this will influence the CMT.

10. Minor corrections:

   - L59: missing reference (?) - maybe just need to rerun bibtex and latex?)
   - L230: Seems to say that u'w' and v'w' are *vertically-integrated* up to model-level 25 to obtain the vertical integral of CMT acceleration up to this height? Is this correct? Surely CMT acceleration is obtained by *vertically differentiating* the eddy fluxes u'w' and v'w'? Therefore, the vertical integral of CMT acceleration below level 25 is just the value of u'w', v'w' at level 25, and no vertical integration is needed? (table A1 seems to say that the analysis does indeed just use the mean values at level 25; consider clarifying this in the text?)
   - L247: typo "quantity" should be "quantify"?
   - L397: Typo "symbolsIn" (missing full-stop and space between sentences).
   - L477: says "Figure 9 suggests that low A generally corresponds with low D at high precipitation intensities ($> 6$ mm/h)". However, nearly all the data-points in figure 9 with precip-rate $> 6$ mm/h are in the same bin for A ($< 0.45$), so any dependence of D on A in this region of the plot cannot be seen by the reader?

---

## Referee Report (RR2)

**2nd Review of "Divergent convective outflow in ICON deep convection-permitting and parameterised deep convection simulations", by Edward Groot, Patrick Kuntze, Annette Miltenberger, and Holger Tost.**

**March 5, 2024**

Many thanks for the updated manuscript. The correction to figure 6 has addressed my biggest concern; the divergence profiles plotted for PER just had an erronious factor of air density; there is no sign that anything else has gone wrong.

I am happy to accept the paper, pending some minor corrections discussed below.

My requests for a more "like for like" comparison between the ICON-PER and ICON-PAR simulations, by using identical analysis methods for both, have not been addressed. But I am satisfied that making these changes to the method would be very time-consuming / require rerunning computationally-expensive data-processing, and would be unlikely to significantly affect the results. So I am happy to accept the article without this change.

I thank the author for acknowledging / clarifying that the study did not find sensitivity of the precipitation - divergence relationship to the choice of parameterised versus explicit convection; the sensitivity found appears to be dominated by model-resolution alone. This point just ought to be clarified in a couple other places in the paper (I give some suggestions below).

New reviewer replies to the authors' response are given below in **blue**, following the original numbered list of queries. Below we follow the authors' colour-scheme by quoting the original review queries in **red**, and the authors' responses to these in **black**. Where line numbers are given, these now refer to the authors' track-changes file (not the original or updated manuscript).

1. The request to clarify that the results apply for the transient response of upper-level divergence, not the eventual balanced response.

   Many thanks for the additional discussion L74-83, and for clarifying in various places that the transient response is studied here. This nicely addresses the query.

2. The query about dependence of upper-level divergence on the vertical structure of the latent heating profile, not just the vertically-integrated latent heating.

...if we assume that the warm air mass is relatively homogeneous (to the east of the frontal zone, which is over the French-German border region) and that the convection takes place on the flanks of this warm air mass, there may not be sufficient variation in stability profiles among all the simulations to cover this aspect of vertical variability with our case study (studying the convective systems in the warm air mass over Germany, directly east of the frontal zone).

My argument does not depend on any horizontal variation in the stability profiles, or variation in stability between different models. Rather it is variation in the vertical structure of the heating profile, and how this correlates with *vertical* variation of environment stability. e.g. consider two different convective storms occuring in the same environment and static stability profile, and suppose (as is often observed) that the static stability $N^2$ is greater in the lower free troposphere than it is in the upper troposphere. Now, suppose storm 1 is a relatively young cell, so that most of the latent heating occurs in the lower troposphere, whereas storm 2 is a mature system with a substantial stratiform anvil region generating more latent heating in the upper troposphere. Even if the storms are producing equal precipitation rates, storm 2 will produce stronger upper-level divergent outflow, just because the heating is concentrated at upper-levels where $N^2$ is less, so that stronger ascent is needed to make the buoyancy removal by vertical advection $(-wN^2)$ balance the heating rate $(\frac{g}{T_v}\frac{\partial T_v}{\partial t}_{latent})$. Even under a horizontally homogeneous environment and uniform neutral buoyancy level, storms in different states of organisation may produce differently-shaped vertical profiles of heating rate, which will give different upper-level divergence if $N^2$ varies with height?

We have looked into this in our previous work, [7]. There, we induced variation in the heating profiles by manipulating the constant of latent heat release.

Many thanks for pointing out the interesting experiment in the authors' LES study, where the latent heat of condensation was varied. Note however that the idealised temperature profile used here (from Weisman & Klemp 1982) has arteficially weak variation of $N^2$ with height (potential temperature increases nearly linearly). So this may not have sampled the sensitivity to vertical structure.

I don't think the present study requires a detailed investigation into dependence of outflow strength on vertical structure (this might be an interesting topic for future work). But it might be worth breifly mentioning this possibility in the discussion?

3. Why would it not be possible to represent the impact of storm morphology on upper-level divergence patterns in a simulation at 13km resolution with

...the truncation scales (which will strongly suppress variability at sub-100 km scales in the ICON-PAR)...

...we do not agree that the ellipses are (at least typically) fitted to precipitation fields at scales that are resolvable in the parameterised convection setup: the truncation of the grid scale is typically at wavelengths of about 6-8 grid cells of the grid spacing [e.g. [8]], which translates to about 100 km for ICON-PAR. Everything smaller than this scale is in the grey zone and partially resolved to unresolved, with a rapid decrease of the resolved fraction, presumably especially at 4-8 times grid spacings. The size of large ellipses grows to about 100 km in ICON-PER, but there are plenty of smaller ellipses in our dataset as well. Therefore, we are a little concerned that the reviewer may just be too optimistic about the ability of numerical models to resolve mesoscale processes at lengths less than 8 times grid spacing...

The reviewer's experience with NWP models is that they routinely do produce both gravity-waves and updrafts / downdrafts very near the grid-scale (excessive single-grid-point updrafts are a common problem!) Whilst a model can only simulate these features with significant numerical error, the poor resolvability does not stop them from happening. The idea that models contain a truncation scale of $\sim$8 grid-lengths below-which all features are strongly damped has come from large eddy simulation, where this has been deliberately imposed by specifying a suitably large mixing-length parameter in the 3D Smagorinsky turbulence scheme (or by using a highly diffusive numerical method for advection). Imposing such an $\sim$8 grid-length truncation scale on an NWP model (with 13km grid-size and parameterised convection) would not be done operationally, as it would degrade the model skill. So I don't agree with the author; features significantly smaller than 100km can and will be simulated by the 13km NWP model, but with diminished accuracy. This is evident in figure 7(c,d); the ensemble standard deviation of divergence in ICON-PAR has a lot of structure on scales of around half a degree ($\sim$50km), and the individual ensemble members will presumably contain smaller-scale structure that gets smoothed-out when looking at the standard deviation over all members?

The results shown in figure 8b show that the divergence-precipitation relationship is not sensitive to the choice of parameterised versus explicit convection at 13km resolution, so the paper's argument that the overly-linear response in PAR is due to the convection parameterisation (e.g. L680, L710) is not supported.

We do agree that the statements in the paper could be interpreted as overstatements of the contrasts, which is indeed suggested by the data points in Figure 8 that compare the 13 km ICON with and without convection parameterisation. We have revised the implied causalities and methodological descriptions along this line...

Many thanks for the changes to the text following this. But there are a few other parts of the paper which still imply that the divergence-precipitation relation depends on the choice of parameterised versus explicit convection, even though no evidence for this has been presented (the sensitivity is dominated by model-resolution alone, as is now stated by the added text at L467).

L750-751: maybe change:

"feedback from deep convection to its surroundings at larger scales is likely not accurately represented with parameterised deep convection, even if the precipitation climatology is well represented: parameterised ensembles seem to be underdispersive in terms of corresponding dynamical variability at a given precipitation rate"

to something like:

"feedback from deep convection to its surroundings at larger scales is likely not accurately represented *at coarse resolution*, even if the precipitation climatology is well represented: ensembles *with 13km grid-size* seem to be underdispersive in terms of corresponding dynamical variability at a given precipitation rate"

Maybe also clarify this in the abstract for consistency, e.g. change L3-4:

"Near-linear response of deep convective outflow strength to net latent heating is found for parameterised convection, ..."

to e.g.

"Near-linear response of deep convective outflow strength to net latent heating is found for 13km grid-spacing with either parameterised or explicit convection, ..."

In addition, to be more specific on the implications for resolving gravity waves, the ICON-PAR grid does not allow to represent gravity wave sources and the full spectrum of relevant gravity waves that the divergence consists of. It may mathematically be considered as a coastline problem in 3D: if we represent an island of 100km2 on a 10 km grid, its coast line would typically be 40 km. However, if we can represent it more accurately at refined grids, the length may deviate strongly.

Many thanks for this insight! While a significant amount of mesoscale structure (in both the updrafts and the radiated gravity-waves) will be present on the 13km grid, it might be the smaller (km-scale) variability *within* the wider mesoscale structures that is key? Even though a 50km-long island can be represented on a 13km grid, the bays and inlets along its coast cannot. Refining the grid to 1km may increase the length of the coastline by an order of magnitude. By analogy, the possibility for gravity-waves radiated from within the "island" to collide with eachother from different directions will vastly increase?

But perhaps clarify at L462-463 that with "little or no information on geometry of the convective systems can be represented", this is really

about geometry of smaller-scale structures *within* the convective systems (i.e. km-scale, rather than mesoscale)?

4. Why is low-pass-filtering on a 45km scale applied to ICON-PER (1km) but not ICON-PAR (13km)?

In retrospective, it might possibly seem a better choice to consider multiple filtering operations, including no filtering, and compare a couple of those options between the two simulation methods as well. On the other hand, this would have created a very comprehensive study and data processing procedure, probably a too comprehensive one for just one manuscript (meaning that two to three manuscripts could be needed to disseminate the results).

No, my argument was that the analysis would be simpler and the manuscript shorter if the low-pass filter was just omitted (it should make little difference as long as the box-averages consider areas much bigger than the filter-scale).

Nevertheless, the filtering procedure ought to serve the purpose of creating comparable datasets in simulations of different resolution, rather than diminish comparability (as possibly suggested by the reviewer). This is because the operation should, when applied to ICON at 13 km grid spacing, filter out very little variability: the dynamical spectra of a numerical model drops rapidly at length scales of about 6-8 times grid spacing...

As discussed earlier, this notion is consistent with LES, but is often not the case in NWP models (especially at coarse resolution with parameterised convection, when they typically do not employ a highly diffusive 3D turbulence scheme to enforce the truncation scale). On the otherhand, it might be that ICON is more diffusive than other NWP models that the reviewer is familiar with. This is critically dependent on the model's advection scheme and any horizontal diffusion used (e.g. as part of a 3D turbulence scheme).

We have selected a data processing procedure and are confident that it may slightly affect individual samples in mostly random directions (e.g., as the reviewer mentions, by incidently moving some random divergence just to the other side of box boundaries). The assessed conditional statistical signals, however, will not be affected by such processing issues that average out over the lifetime of the systems and, even further, over the full dataset.

I am happy to take the authors' word that the low-pass filtering does not significantly affect the results anyway. So it is probably not worth rerunning significant parts of the work-flow to pursue this further.

5. Why not define moving boxes in ICON-PAR, to be fully consistent with the analysis of ICON-PER?

We could have made the choice to further investigate the systems in ICON-PAR. However, this has not been done, as from each system, we have analysed several ensemble members (and physically perturbed simulations) to

make sure that the statistical sample size allows for safe conclusions. Furthermore, there are fewer convective systems in these ICON-PAR simulations...

...investigating residual variability and increasing the sample size was clearly not prioritised over the corresponding more detailed analysis of ICON-PER.

This is a reasonable argument; the moving boxes in ICON-PER have been setup manually, repeating this exercise for every ensemble member and sensitivity-test within ICON-PAR would be very laborious, and the effort should be focussed on analysing the more interesting behaviour found in ICON-PER. However, one of the paper's main conclusions is drawn from comparing ICON-PER with ICON-PAR. It would therefore be stronger if more attention was given to ensuring like-for-like comparison and using the same analysis methods when comparing them. e.g. just for the comparison with PAR, the divergence-precip relation in PER could be computed on the same large static boxes as PAR (while still using the smaller moving boxes for the more detailed investigation of relationship to storm structure etc in PER)

But again, at this late stage it may not be worth rerunning the work-flow for a consistency-check which is unlikely to affect the results much.

6. Figure 6: Do the profiles of ensemble-mean box-mean divergence satisfy mass continuity?? In ICON-PER, it looks as though the vertically-integrated lower-troposheric convergence is significantly greater than the upper-tropospheric divergence...

There is indeed a mistake in Figure 6, where the ICON-PER data display mass divergence data rather than divergence data. Therefore, Figure 6 has been corrected accordingly in the updated manuscript, leading to balanced profiles. This issue has not been found in Figure S5, and Figures 6b and S5 should display similar divergence profiles for ICON-PAR, accounting for the respective pressure axis, which means they are at/very close to mass balance. We thank the reviewer for the close and attentive inspection of the figures, which allowed us to correct this inconsistency in Figure 6!

Many thanks for investigating and correcting this. It is very reassuring that it was essentially only the units shown on the colour-scale (6a) and x-axis (6b) that were wrong before! This has been corrected by removing an erronious factor of air density from the plotted data, so that they are now consistent with the stated units. This increases my confidence in the paper overall, by removing my worry that something else had gone wrong with the analysis.

7. Why does the analysis of PAR use much bigger boxes than PER?

The size of the boxes could unfortunately not be chosen smaller than 2-4 degrees in the ICON-PAR; the convective systems inevitably grow larger in this simulation...

...Nevertheless, carefully studying the divergence at upper levels, the convective systems and precipitation rates in ICON-PAR, we have assured that locations with variability in divergence are included in our analysis. This strategy maximises the reliability of the box analysis at the minimum of cost of included remote subsidence (based on the discussion of [7]).

As discussed earlier it would be possible to ensure like-for-like comparison between PAR and PER, by separately computing the precipitation and divergence in PER over the same large static boxes as were used for PAR. But again it is probably not worth rerunning such a large part of the work-flow at this late stage.

I am not overly concerned about the subsidence assumption in the convection parameterisation here. In the reviewer's experience with mass-flux convection schemes, the assumption of "local mass compensation" has little impact, as the model's dynamical core still acts so-as to quickly restore a Weak Temperature Gradient in the horizontal after the parameterised convective heating is applied. This has the effect of immediately undoing the local subsidence imposed by the convection scheme, and redistributing it elsewhere, so that the end result is the same as explicit convection with the equivalent heating profile. The present study's results in figure 8b comparing the divergence response for parameterised versus exlicit convection at 13km grid-size support this.

8. figure 8: The text says that "from here on vertically integrated values (of upper-level mass divergence) are used". But the quantity plotted on the y-axis says it has units kg m$^{-3}$ s$^{-1}$...

Essentially, the above approach is equivalent to what has been done for this work. Vertical integration of divergence over the whole layer depth causes downdraft divergence at lower levels to be included (which is partially visible as mean feature towards the end of the time series in Figure 6). By setting a mask this issue is resolved, as the reviewer states. Exactly this has been done. Nevertheless, substantial divergence features from downdrafts do vertically overlap with outflow from shallow convection in ICON-PER, which means that the perfect vertical mask does not exist. Whether the processing leads to a divergence in kilogram per cubic or squared meter depends on whether one concludes by dividing the whole integral by length scale L or not, to obtain the mean value of the integrand. Both are leading to exactly the same result [while the way it is currently plotted, one can also roughly intercompare the local intensity variation of outflows (integrand), which means it is purely a matter of taste, which of the two is favored]. The manuscript has been corrected accordingly.

Thank you for the edited text at L415, which now clarifies that the quantity shown is the vertical integral divided by outflow depth. Omitting the normalisation by outflow depth would have made the comparison in figure 8 between PER (8a) and PAR (8b) simpler, allowing a slightly more succinct manuscript (the discussion about the correction for differing outflow

depths would not be needed). But it does not matter that much.

9. Unclear direction of causality in the correlation between CMT and outflow strength.

   This is a very interesting comment; there is indeed no complete certainty about causality, based on the correlation signal. We agree that both pressure perturbations and momentum/accelerations by turbulent motion can affect the upper tropospheric flow, next to the dominant cause of divergence - the heating in the middle troposphere, which partially arises from pressure perturbations...

   Many thanks for the extensive discussion around the relation between heating, the pressure field, and vertical momentum transport (and the dependence of the "turbulent" transport on model resolution). This made for an interesting and thought-provoking read, and I agree this wider subject is beyond the scope of the present study.

   I am happy with the slight rewording at L612-613, in the synthesis section. Maybe consider also mentioning the uncertain direction of causality in section 6.3, e.g. at L568-569?

10. Minor corrections.

    Many thanks; all the previously listed minor corrections have been addressed. Except:

    L263: "Ellipse fitting and verification are used to quantity the geometry" should be "Ellipse fitting and verification are used to *quantify* the geometry?"

    I also just spotted one more trivial typo:

    L562: "...while Figure Figure 10b shows..."

---

## Author Response (AR2)

**Replies to all Referee Comments on "Divergent convective outflow in ICON deep convection-permitting and parameterised deep convection simulations. " [1]**

Edward Groot, Patrick Kuntze, Annette Miltenberger, Holger Tost

February 6, 2024

**1 General**

We thank both referees for reading our manuscript and formulating their summary and feedback [2, 3], which we welcome to make further improvements to the manuscript. We think that the reviewers have provided very useful input to do so, which brings us to some purely textual revisions and a figure correction to accomplish an improved manuscript and address the concerns of the reviewers.

**2 Referee 1: reply to RC1 [2]**

1. *'"The authors have made a substantial effort to significantly improve their manuscript on dimensionality, ensemble and convective momentum transport aspects and it is now acceptable, but I suggest to address the remaining minor points below (including shortening section 7)"'*

   We thank the reviewer for this feedback and are very glad that the content of the work has been much clarified after the revisions.

2. *'"l59 typo "In (?) explicit ";*
   *l82 "is inversely proportional to their vertical wavelength" as $m^s = N^2/C^2$, speed $C$ is inversely proportional to vertical wavenumber or proportional to vertical wavelength, please double check and correct;*
   *l247 "to quantity geometry" you mean "quantify the"?*
   *l397 typo "symbolsIn"'*

   The updated manuscript will be corrected accordingly.

3. *'"I found section 7 and Conclusions very long and sometimes repetitive, one could remove 7.3.2 and shorten 7.2.1 by half"'*

   We will have another critical look at Sections 7.3.2 and 7.2.1. We would also just like to remind the reviewers of the earlier round of reviews, in which Section 7 was considered critical and potentially highly relevant to bring up novelties in this study [2, 4], but at the same time a section to extensively work on. Therefore, we will also try to beware when shortening these parts of Section 7, but we agree that 7.2.1 is particularly lengthy and shorten this Section. Section 7.3.2 can probably be merged with the previously existing 7.3.3 in the revised manuscript. We would also like to thank the reviewer for raising this point.

**3 Referee 3: reply to RC3 [3]**

- *'"I find this article a very interesting read and I hope that it is published. The relation between net latent heating (quantified as the surface precipitation rate) and divergent outflow in the upper troposphere is investigated, using ICON simulations of a real case of severe thunderstorms over Southern Germany. This builds on the author's idealised LES study already published. Consistent with the earlier LES work, the present study finds that upper-level divergent outflow increases with latent heating, but*

*increases more slowly as the organisation / clustering of the convective updrafts increases. This effect is found in 1km resolution simulations, but is absent from simulations at 13km resolution. It is suggested that resolving small-scale gravity-waves is key to representing the relative slowing of upper-level divergence with increasing convective organisation. The relation between upper-tropospheric outflow and convective momentum transport is also investigated. "'*

We would like to thank the reviewer for this very encouraging view on this work. Furthermore, we are glad that the key message could be distilled from the work, which had turned out to be rather difficult from the initial submission [2, 4].

- *'"However I have a number of fundamental questions about the theory, methodology and how the results are interpreted. I hope that in answering these questions, the paper can be made clearer and stronger. In its current form, it is not clear to me that all of the stated conclusions are proven by the results shown.*
  *So I would recommend significant revisions, if that is still possible at this stage? I also note that this paper was originally submitted 9 months ago, so has been held up for a long time already (I understand one of the previous reviewers is now unavailable, which is why I have been invited to review the article so late in the process). Many apologies if my queries introduce further delay"'*

We will address the concerns in the following point by point and, wherever we think it is needed, include the relevant considerations into the discussion of a revised version of the manuscript. We think that addressing the reviewer's questions will indeed make the work stronger. Furthermore, we are glad that the reviewer is aware of the specific setting in which a review of our work has been requested.

1. *'"Introduction: L20 and onwards. A key argument of this paper is that the amount of upper-level divergent outflow driven by a given amount of latent heating within convective clouds varies as a function of storm horizontal structure. However, surely the total (time-integrated) amount of ascent (and hence upper-level divergence) is constrained to be that required to restore the heated air to neutral buoyancy? This is the basis of "Weak Temperature Gradient" scaling argument, which is key to the dynamics of moist convection and its interaction with the stably-stratified free troposphere. Latent heating will temporarily make the cloudy updraft air buoyant relative to its surroundings; it then ascends relative to its stably-stratified surroundings until it reaches its neutral buoyancy level, and the amount of upper-level divergence forced by this must be constrained by mass-continuity. If the time-integrated upper-level divergence was less than this constraint, then the cloud column would remain buoyant after ascent had stopped. This should only be possible if the resulting warm anomaly is held in place by a rotationally-balanced flow, which will not occur at convective scales. I think the paper's findings can be reconciled with this argument because it actually considers the transient response of the upper-level divergence to the latent heating, not the total time-integrated response. We would expect the upper-level divergence to continue after the latent heating has stopped, until the warm anomaly is removed, but this final stage of the outflow is not considered in this study. So, the authors find significant variation in the transient rate at which the upper-level divergence spins-up in response to latent heating, as a function of storm structure. This is a very interesting result, especially the discrepency of this between different model resolutions. I would just ask the authors to clarify that their results apply for the transient response and not the time-integrated total response of upper-level divergence to latent heating. e.g. replace the often repeated term "amount of upper tropospheric divergence" with something like "instantaneous rate of upper tropospheric divergence""'*

   We agree that the distinction should be made between a time-integrated and transient response and the terminology is adapted accordingly for the revised manuscript and detailed in lines 74-81 of the revised manuscript. The time-integrated response in the setting of a realistic NWP simulation requires a different methodology and approach, because systems at time lags are indeed thought to produce time-integrated divergence over larger mesoscales with mixed transients, consisting of infragravity wave signals that are in different adjustment states regarding their transients to balanced flow [e.g. [5, 6]].
   We have tried to investigate if there was some lag effect in the response of transient divergence signals in [7] affecting the time-integrated response on long time scales, but have not been able to find substantial evidence for that. However, in these idealised two hour simulations, the Coriolis effect has been switched off, which clearly controls the time-integrated effect [7].
   Given the realistic and more complicated setup of the current study, the integrated effect is not what we aim to analyse here and should indeed be emphasised in this context (also by shortly addressing this point in the introduction and/or discussion).

Indeed, by focusing on the transient, localised and few-hour time scales, we distill responses to convection that are relevant to transient momentum and heating transfer to larger mesoscales (MCS scales and slightly beyond; as quantified by divergence) and their variation in representation.

2. *'"One factor not considered in this paper is the influence of the vertical structure of the latent heating profile on the mass divergence rate. If heating is concentrated in a layer of the troposphere with higher (lower) static stability, then less (more) ascent is required to restore the heated air to neutral buoyancy, and hence less (more) upper-level divergence will result. One possible hypothesis that might be worth checking is whether part of the variation in upper-level divergence per unit latent heating can be explained in this way by variation in the vertical structure of the heating profile, rather than storm horizontal morphology. It might be that the coarse simulations have deficient variability in the vertical structure, and hence give less variability in outflow strength"'*

These hypotheses have not been ignored for our works (both [7] and this study; which will be explained in the following paragraphs). In addition, if we assume that the warm air mass is relatively homogeneous (to the east of the frontal zone, which is over the French-German border region) and that the convection takes place on the flanks of this warm air mass, there may not be sufficient variation in stability profiles among all the simulations to cover this aspect of vertical variability with our case study (studying the convective systems in the warm air mass over Germany, directly east of the frontal zone). It should not be an issue of the different ICON configurations, as the level of neutral buoyancy will typically locate at or very near to the tropopause, with nearly consistent and nearly similar levels of neutral buoyancy and maximum divergence between both configurations (e.g. Figure 6). However, in the below, the impressions based on the two studies are summarised.

We have looked into this in our previous work, [7]. There, we induced variation in the heating profiles by manipulating the constant of latent heat release. We have also tried this procedure in ICON(-PAR simulations - with physics manipulation), but in both [7] and the current case, we could not find structural responses to differences in heating profiles other than those relating to variations in the level of neutral buoyancy. This response was clearly visible in the idealised work of [7] and less clear for certain systems in the current study. Some systems show a clear response of differences in outflow levels and others show a response that is statistically negligible. Also, in both [7] and the current work, the magnitude of (instantaneous) divergence is suggested to not be structurally affected in all those simulations (at least not statistically). It should be noted that the idealised setting allows one to distill the statistical signals in a more confident way than the configuration of the current study.

As a result of the above findings, we think that the way it has been reported in [7] covers what we can confidently say about outflow level/thickness variability among the convective outflows.

3. *'"Presumably the gravity-wave dynamics thought to mediate this impact could be at least partially resolved at this resolution, and represented by the model's dynamical core rather than the convection parameterisation?"'*

First, we need to disentangle the production of the (instantaneous) divergence: the convective parameterisation and/or grid-scale microphysics produce net heating as a consequence of net condensation and precipitation. As a result, the troposphere will warm. The dynamical core is informed by the accompanied heating. Therefore, a response is triggered in the form of the divergence at upper levels. Furthermore, convergence at low levels can be enhanced.

*'"L136: Why would it not be possible to represent the impact of storm morphology on upper-level divergence patterns in a simulation at 13km resolution with parameterised convection? (...) I similarly query the discussion at L439-440: "little or no information on geometry of the convective systems can be represented with a parameterisation"; surely much of the geometry (as quantified by the ellipse fitting in this study) is mesoscale, so occurs on the resolved-scale in the 13km grid PAR simulations. So the gravity-wave response to the storm structure could be captured. The results shown in figure 8b show that the divergence-precipitation relationship is not sensitive to the choice of parameterised versus explicit convection at 13km resolution, so the paper's argument that the overly-linear response in PAR is due to the convection parameterisation (e.g. L680, L710) is not supported. It might be that the PAR simulation simply fails to produce enough emergent mesoscale organisation on the grid (even though it could be resolved). However, figure 7(c,d) appears to show plenty of mesoscale structure in the divergence field for the PAR simulation."'* and *'"In various places the paper claims that the dependence of outflow strength on convective organisation cannot be represented when using parameterised convection. But the results don't support this conclusion. The organisation quantified using the fitting of ellipses to the precipitation field is largely on scales that are resolvable in the parameterised convection simulation. Any resulting gravity-waves connected with the mesoscale heating structure would similarly be resolvable. And the results in figure 8b show that turning off the convection parameterisation (while keeping all else equal) made no difference to the divergence-precipitation relation"'*

We do agree that the statements in the paper could be interpreted as overstatements of the contrasts, which is indeed suggested by the data points in Figure 8 that compare the 13 km ICON with and without convection parameterisation. We have revised the implied causalities and methodological descriptions along this line, correcting from *"that convective organisation is not directly represented in simulation configurations with parameterised deep convection"* to

**"that convective organisation is weakly represented in simulation configurations with parameterised deep convection and weakly coupled to the engine of numerical models (the dynamical cores); we could say it is clearly underrepresented"** (lines $\approx$ 310 of the revised manuscript). Further details are found in the revised manuscript. Nevertheless, the close association between 13 km grid spacing with parameterised convection and storm-resolving (e.g. 1 km) grid spacing with explicit convection exists by the virtue of these being the simulation settings in nearly all cases.

Furthermore, we think it is an important factor that the systems in parameterised ICON (and convective systems in most parameterised configurations) seem to mix dynamical behaviour of stratiform and precipitation convective systems in a biased way, leaning more heavily towards the stratiform side, whereas storm-resolving models seem to (over-?)lean towards strongly convective, especially in ICON. This is likely to a large extent explained by grid spacing, although we cannot be certain from our study whether the parameterisation of convection specifically also has a substantial effect (not ruled out by our results; To study this question explicitly, we would ideally need additional grid spacings of 3-10 km in the grey zone with both explicit and parameterised deep convection, which could anyway further clarify on some of the findings from this work.) Therefore, we (slightly over-)phrased it as (variability in convective organisation and geometry) "cannot be represented" in ICON-PAR versus it being represented very explicitly in convection-permitting set-up. In reality, and in rephrasing the statements, we should rather say that the organisation process can be weakly represented in ICON-PAR/13 km grid spacing compared to good representation (possibly even biased under- or overrepresentation of organised updrafts) simulations with ICON-PER/1 km grid spacing.

In addition, to be more specific on the implications for resolving gravity waves, the ICON-PAR grid does not allow to represent gravity wave sources and the full spectrum of relevant gravity waves that the divergence consists of. It may mathematically be considered as a coastline problem in 3D: if we represent an island of 100km$^2$ on a 10 km grid, its coast line would typically be 40 km. However, if we can represent it more accurately at refined grids, the length may deviate strongly. One could think of gravity wave sources in similar ways, which suggests that their interactions can be represented in much more complex ways at refined grids. Furthermore, as the gravity wave spectrum will extend at sub-grid scales in ICON-PAR, the responses can be more accurately simulated with a setup like ICON-PER. This effect, together with the truncation scales (which will strongly suppress variability at sub-100 km scales in the ICON-PAR) is thought to explain the difference in dispersion between ICON-PAR and ICON-PER, and a corresponding brief explicit discussion included in the revised manuscript.

Furthermore, in line with the closing summary of the previous paragraph, we do not agree that the ellipses are (at least typically) fitted to precipitation fields at scales that are resolvable in the parameterised convection setup: the truncation of the grid scale is typically at wavelengths of about 6-8 grid cells of the grid spacing [e.g. [8]], which translates to about 100 km for ICON-PAR. Everything smaller than this scale is in the grey zone and partially resolved to unresolved, with a rapid decrease of the resolved fraction, presumably especially at 4-8 times grid spacings. The size of large ellipses grows to about 100 km in ICON-PER, but there are plenty of smaller ellipses in our dataset as well. Therefore, we are a little concerned that the reviewer may just be too optimistic about the ability of numerical models to resolve mesoscale processes at lengths less than 8 times grid spacing, which is readdressed in the following.

4. *'"192: It is stated that the divergence fields in the PER simulation are low-pass-filtered to remove scales below 45km; is the same low-pass filtering also applied to the divergence in the PAR simulations, to ensure fair comparison? Since PAR has a grid-size of 13km, it may have some variability at scales smaller than 45km, so should be filtered in the same way as the PER simulations? Also, why is it nessecary to low-pass-filter the fields first, given that the analysis only considers the divergence spatially averaged over large boxes anyway? Is there a danger that the low-pass filtering may spread some of the divergence across the box boundaries, so that it is spuriously missed in the analysis?"'and '"The method of defining regions of interest and calculating the mean upper- level divergence differs between the 1km explicit versus 13km parameterised simulations, so it is not a like-for-like comparison (the text implies that the 1km run divergence field was first filtered onto a horizontal scale significantly coarser than the resolution of the parameterised run, yet no filtering was applied to the parameterised run for consistency; also much larger boxes were used in the parameterised run)."'*

The filtering operation should in our opinion be separated from the box method. The box method on itself is applied to both ICON-PAR and ICON-PER; in the processing, the filtering is essentially applied to in the first place assure that the convection-permitting ICON dataset does not represent variability that the ICON-PAR simulations cannot represent, after which to equivalent datasets the box method is used for integration.

In retrospective, it might possibly seem a better choice to consider multiple filtering operations, including no filtering, and compare a couple of those options between the two simulation methods as well. On the other hand, this would have created a very comprehensive study and data processing procedure, probably a too comprehensive one for just one manuscript (meaning that two to three manuscripts could be needed to disseminate the results).

Nevertheless, the filtering procedure ought to serve the purpose of creating comparable datasets in simulations of different resolution, rather than diminish comparability (as possibly suggested by the reviewer). This is because the operation should, when applied to ICON at 13 km grid spacing, filter out very little variability: the dynamical spectra of a numerical model drops rapidly at length scales of about 6-8 times grid spacing [[8], and also many other studies, which investigate spectra of model simulated dynamics]. At the wavelengths of up to 45 km, the 13 km resolution ICON simulations can be assumed to be heavily truncated and this range falls well below the effective resolution (which is somewhere in the range of 60-120 km)! The scale at which most of the variability gets truncated is just below this effective resolution, in the range of 40-60 km wavelengths. To emphasise why we chose to filter accordingly, we have expanded the paragraph on filtering with a more specific statement in the revised manuscript: **"The filtering step assures that the box integrations that we carry out are applied to datasets with very similar truncation scales."** (lines 204-205)

The choice of using a low-pass filter was further motivated by the different grid spacings of ICON-PAR/ICON-PER and the ability to track mesoscale divergence at upper levels within the ensemble members of ICON-PER, without being contaminated by high-amplitude small scale signals of small-scale (5-20 km) divergence patches (associated with convection and gravity waves at those scales).

We have selected a data processing procedure and are confident that it may slightly affect individual samples in mostly random directions (e.g., as the reviewer mentions, by incidently moving some random divergence just to the other side of box boundaries). The assessed conditional statistical signals, however, will not be affected by such processing issues that average out over the lifetime of the systems and, even further, over the full dataset.

The box boundaries have been inserted with care, but because of events such as merging of convective systems and variations in storm tracks, it can never be assured that individual samples in such a case study are "well-behaved" (as controllable or predictable as in idealised simulations, independently of data processing and filters). The statistical assessment has nevertheless allowed us to assess uncertainty in the findings.

5. *'"298 - 305: I am struggling to understand this paragraph (describing the identification of the convective systems in the PAR simulations); consider clarifying? It says that "corresponding precipitation moves together with conditionally unstable or lifted air masses" and "A typical (mesoscale) convective system is easily contained within a box of several to tens of grid cells in each horizontal direction for ICON-PAR" This seems to say that the PAR simulation should produce well-resolved regions of precipitation which move with the flow, but then the text says that only 3 static boxes are used to define the convective systems in PAR. Why not define moving boxes to track the systems, in exactly the same way as is done for the PER simulations?"'*

We could have made the choice to further investigate the systems in ICON-PAR. However, this has not been done, as from each system, we have analysed several ensemble members (and physically perturbed simulations) to make sure that the statistical sample size allows for safe conclusions. Furthermore, there are fewer convective systems in these ICON-PAR simulations - the three mesoscale systems over Central Europe have all been analysed, where essentially the Central-Germany system is a child of the Alps System. A workflow for data processing equivalent to ICON-PER is not beneficial if only a small set of systems exists and the convection scheme makes the cloud system intrinsically a constant regeneration of itself.

It is possible to analyse the systems over a longer duration, but for the family Alps/Central-Germany, the intensity (rain rate) lowered in an intermediate stage. The strongest diabatic signals from convection will occur when the system is most active, and we include at least one record at low precipitation rate (when one system did not exist for a certain simulation).

Last, but not least, the ICON-PER simulations have this interesting scatter in their ratio D, which ICON-PAR does not have. Therefore, investigating residual variability and increasing the sample size

was clearly not prioritised over the corresponding more detailed analysis of ICON-PER.

6. *'"The divergence profiles shown in figure 6 do not vertically-integrate to zero for the 1km simulations, implying there is strong net convergence of mass into the defined boxes (see my point 6 below for detail). Whilst the model may solve the compressible equations, the magnitude of this mass imbalance does not seem realistic, which makes me worry that there has been mistake in the analysis?"'* and *'"Figure 6: Do the profiles of ensemble-mean box-mean divergence satisfy mass continuity?? In ICON-PER, it looks as though the vertically-integrated lower-trposheric convergence is significantly greater than the upper-tropospheric divergence. Assuming the rate of change in the mass of air contained in the box is small, we should have:*

$$\int_0^{z_{top}} \rho(\frac{\partial u}{\partial x} + \frac{\partial v}{\partial y})dz = 0 \tag{1}$$

*and assuming small deviation from hydrostatic balance ($dp = gdz$):*

$$\int_{p_{surf}}^{p_{top}} (\frac{\partial u}{\partial x} + \frac{\partial v}{\partial y})dp = 0 \tag{2}$$

*But if I try to vertically-integrate the divergence in figure 6 over the y-axis pressure coordinate "by eye", it does not balance and there is strong net convergence into the box (especially at earlier times). e.g. considering the time-averaged profile for PER (fig 6b, black line), we have convergence between 900 and 400 hPa averaging around $0.4 \times 10^{-4}s^{-1}$, giving: $5 \times 10^4$ Pa $\times 4 \times 10^{-5}$ $s^{-1}/g \approx 0.2$ kg $m^{-2}$ $s^{-1}$ And we have divergence between 400 and 180 hPa averaging around $0.5 \times 10^{-4}s^{-1}$, giving: $2.2 \times 10^4$ Pa $\times 5 \times 10^{-5}$ $s^1/g \approx 0.1$ kg $m^2$ $s^1$ So the convergence and divergence seem to be out of balance by about a factor of 2. The mass imbalance of the order 0.1 kg $m^2$ $s^1$ would increase the mass of air in the box by 25% during the 7 hour period shown, which is clearly not realistic. I can't see this imbalance in the divergence profiles for the PAR simulation (fig 6b, green line, has similar upper tropospheric divergence to PER, but much weaker lower-tropospheric convergence, so looks in-balance). Similar for the PAR divergence profiles in figure S5. Does this suggest there is something wrong, either with the model or the analysis method in PER? Could the discrepency in PER be somehow due to the low-pass filtering?"'*

There is indeed a mistake in Figure 6, where the ICON-PER data display mass divergence data rather than divergence data. Therefore, Figure 6 has been corrected accordingly in the updated manuscript, leading to balanced profiles.

This issue has not been found in Figure S5, and Figures 6b and S5 should display similar divergence profiles for ICON-PAR, accounting for the respective pressure axis, which means they are at/very close to mass balance.

We thank the reviewer for the close and attentive inspection of the figures, which allowed us to correct this inconsistency in Figure 6!

7. *'"Comparing figure 4 with figure 7, it appears that the boxes used to define the convective systems (and compute spatially-averaged divergence and precip rate) have quite different size in PER and PAR, so it is not a fair comparison? The example box in PER shown in figure 4 appears to be a square with side length about 1 degree in latitude ( 110 km), whereas the boxes for PAR shown in figure 7 are rectangles with side lengths 2-4 degrees. The use of systematically larger boxes in PAR might risk including more of the background subsidence within the box, so that the spatially-integrated divergence is reduced?"'*

The size of the boxes could unfortunately not be chosen smaller than 2-4 degrees in the ICON-PAR; the convective systems inevitably grow larger in this simulation. It is a process we cannot control, and since we have a large dataset for this case, we could gladly profit from simulations of the same day and event. The box size in ICON-PAR could be considered a small caveat of the case study.

In the discussion of [7], we have described the impact of the box size on the analysis; of course we cannot rule out that subsidence takes place over the chosen (larger) boxes in ICON at 13 km grid spacing, which may not always be completely realistic; this is by construction implied in the structure of most frequently used convective parameterisations, even if it may not always be a realistic representation of mesoscale convective systems.

Nevertheless, carefully studying the divergence at upper levels, the convective systems and precipitation rates in ICON-PAR, we have assured that locations with variability in divergence are included in our analysis. This strategy maximises the reliability of the box analysis at the minimum of "cost" of included remote subsidence (based on the discussion of [7]).

The trade-off of a highly dynamical case is the displacement of some of the upper tropospheric divergence variability, by strong upper-level winds, to the northwestern flank of the convective systems,

which inevitably resulted in the box size we used in ICON-PAR. As opposed to an idealised case study, where the storm-relative winds are controllable, this inevitably is the best we can do in the current study (at least for a mid-latitude case where organisation is predominantly controlled by wind shear).

8. *'"393 / figure 8: The text says that "from here on vertically integrated values (of upper-level mass divergence) are used". But the quantity plotted on the y-axis says it has units kg m3 s1, which is presumably just $\rho(\frac{\partial u}{\partial x} + \frac{\partial v}{\partial y})$ Shouldn't the units be kg m2 s1 if this has been vertically-integrated? Also the lines shown in the figure for comparing PAR with PER (and the discussion around L435) indicate a correction to the divergence to account for the different thicknesses of the outflow layers, but this should not be needed if vertical integrals were plotted as suggested in the text. I would recommend calculating the vertical integral of mass divergence for both PER and PAR and comparing those:*

$$\int_0^{z_{top}} L(z)\rho(\frac{\partial u}{\partial x} + \frac{\partial v}{\partial y})dz \tag{3}$$

*where L(z) is 1 in the outflow layer, zero at other heights. The result then has units kg m2 s1, which can be compared more directly with precipitation rate (which can be expressed in the same units!) Also Re the discussion around L444 (where the text suggests that the apparent reduction in divergence-precipitation ratio for the 13km simulation with only shallow convection parameterised is due to the outflow layer extending lower-down); this could easily be remedied by plotting the above vertically-integrated divergence, and setting the outflow-layer mask L(z) interactively so that it always captures all model-levels within the time-varying outflow layer depth (instead of setting L(z) to zero below a fixed height)."'*

Essentially, the above approach is equivalent to what has been done for this work. Vertical integration of divergence over the whole layer depth causes downdraft divergence at lower levels to be included (which is partially visible as mean feature towards the end of the time series in Figure 6). By setting a mask this issue is resolved, as the reviewer states. Exactly this has been done. Nevertheless, substantial divergence features from downdrafts do vertically overlap with outflow from shallow convection in ICON-PER, which means that the perfect vertical mask does not exist.

Whether the processing leads to a divergence in kilogram per cubic or squared meter depends on whether one concludes by dividing the whole integral (above, equation 3, as inserted by the reviewer) by length scale L or not, to obtain the mean value of the integrand. Both are leading to exactly the same result [while the way it is currently plotted, one can also roughly intercompare the local intensity variation of outflows (integrand), which means it is purely a matter of taste, which of the two is favored]. The manuscript has been corrected accordingly.

9. *'"L537, L580, section 7.2.2 (relation between CMT and outflow strength); The study finds a positive correlation between momentum flux and upper- level divergence (with both normalised by precipitation rate). From this, they conclude that convective momentum transport has some impact on the upper-level divergence. However, correlation does not imply causality. It might be that upper-level divergence (or some aspect of storm structure which is correlated with it) has modified the convective momentum transport? e.g. more organised convective systems may have quite different pressure gradient forces acting on the horizontal momentum of the updrafts, and this will influence the CMT"'*

This is a very interesting comment; there is indeed no complete certainty about causality, based on the correlation signal.

We agree that both pressure perturbations and momentum/accelerations by turbulent motion can affect the upper tropospheric flow, next to the dominant cause of divergence - the heating in the middle troposphere, which partially arises from pressure perturbations.

Now, pressure perturbations are resolved differently between say, the simulation at 13 km grid versus the other at 1 km grid spacing - which is a result of differences in the gravity wave spectra that can be represented. Hence, our view is that the role of pressure perturbations for CMT have been interpreted (in existing literature, e.g. [9]) in a way that is not very clear to us. It might (as the reviewer suggests) imply a relation of CMT to storm structure, but we could not discern a relation from the current study as a consequence of the chosen methodology, and neither from [7] as a consequence of no relation between CMT and upper tropospheric divergence rates - CMT is correlated to convective organisation, neither to storm structure nor to upper-level divergence in that work. Furthermore, which part of the gravity wave spectra is resolved is affected by the numerical model and its resolution; hence the pressure perturbations that excite the resolved momentum acceleration depends on resolution by definition. Now to us, the question arise, what do we define as

- CMT momentum tendencies

- Divergence response
- Pressure perturbations

Since pressure perturbations are described to (at least partially) drive both CMT tendencies and divergence - [9] and our work. As the grid refines, we will generally resolve mesoscale convective pressure perturbations better and could re-attribute some momentum tendencies to the pressure fluctuations around mesoscale convective systems.

We would argue however, that pressure perturbations are also largely a result of the localised heating source, and they partially propagate as gravity waves (which affect pressure themselves).

So it is fair to ask the question: which part of the divergence is aliased onto a process classically called convective momentum transport and arising from sub-grid turbulence at 1 km and finer grid spacings and which part is driven by heating; which part is driven by the convective updrafts and how? Or, at any grid spacing, which part of it is arising from sub-grid turbulence and which part is heating-induced? Is much of what we are often calling CMT essentially sub-grid dipoles of divergence/convergence, arising from gravity wave interactions, that can propagate into the surroundings of updrafts and downdrafts? This is a question well beyond the scope of this work and ultimately also more closely connected to turbulence theory than to deep convection.

Nevertheless, this more fundamental question could challenge the views on CMT, but does not imply that one may not be able to fully differentiate between the two (at least in our view) - given that the differentation is a function of grid spacing. Therefore, in our view, it cannot be stated with certainty that the causality question does actually matter, nor that the causality may indeed be wrongly described in our study.

We would say it is an interesting philosophical question to address in a separate work, one day (about which reviewers or readers would be welcome to exchange thoughts).

Based on the reviewer's suggestion, we have chosen for an even safer formulation than before to describe the relation between CMT and the found correlation, avoiding the suggestion of proven causality in the updated manuscript completely, by correcting *"Given a certain precipitation rate, it has been found here that flow perturbations induced as a result of convective momentum transport have some slight impact the mass divergence slightly in ICON-PER. "* to **"Given a certain precipitation rate, it is suggested by the findings here that flow perturbations induced as a result of convective momentum transport can likely impact the mass divergence rate slightly in ICON-PER"** (lines 605-606). However, the earlier formulations already did not seem to suggest proven causality in our opinion (e.g. choice of "probably" rather than certainly).

10. *'"Minor corrections:*

    - *L59: missing reference (?) - maybe just need to rerun bibtex and latex?)*
    - *L247: typo "quantity" should be "quantify"?*
    - *L397: Typo "symbolsIn" (missing full-stop and space between sentences).*

    *"'*

    Correct; consistently with replies to the other reviewer, these have been corrected for the revisions.

    - *'"L230: Seems to say that u'w' and v'w' are vertically-integrated up to model-level 25 to obtain the vertical integral of CMT acceleration up to this height? Is this correct? Surely CMT acceleration is obtained by vertically differentiating the eddy fluxes u'w' and v'w' ? Therefore, the vertical integral of CMT acceleration below level 25 is just the value of u'w', v'w' at level 25, and no vertical integration is needed? (table A1 seems to say that the analysis does indeed just use the mean values at level 25; consider clarifying this in the text?) "'*
    The analysis of the reviewer is correct; We have clarified the text further (L246-247), regarding this point.

    - *'"L477: says "Figure 9 suggests that low A generally corresponds with low D at high precipitation intensities ( 6 mm/h)". However, nearly all the data-points in figure 9 with precip-rate 6 mm/h are in the same bin for A ( 0.45), so any dependence of D on A in this region of the plot cannot be seen by the reader?*

    *"'*

    Indeed, we will correct the text by removing the practically useless statement here.

**References**

[1] Edward Groot, Patrick Kuntze, Annette Miltenberger, and Holger Tost. Divergent convective outflow in icon deep convection-permitting and parameterised deep convection simulations. [original pre-print and revised version]. *EGUsphere*, 2023:1–32, 2023.

[2] The two referee comments of referee 1 on "divergent convective outflow in icon deep convection-permitting and parameterised deep convection simulations.", 2023.

[3] Referee comment 3 on "divergent convective outflow in icon deep convection-permitting and parameterised deep convection simulations.", 2024.

[4] Referee comment 2 on "divergent convective outflow in icon deep convection-permitting and parameterised deep convection simulations.", 2023.

[5] L. Bierdel, T. Selz, and G.C. Craig. Theoretical aspects of upscale error growth through the mesoscales: an analytical model. *Quarterly Journal of the Royal Meteorological Society*, 143(709):3048–3059, October 2017.

[6] L. Bierdel, T. Selz, and G. C. Craig. Theoretical aspects of upscale error growth on the mesoscales: Idealized numerical simulations. *Quarterly Journal of the Royal Meteorological Society*, 144(712):682–694, April 2018.

[7] Edward Groot and Holger Tost. Divergent convective outflow in large-eddy simulations. *Atmospheric Chemistry and Physics*, 23:6065–6081, 2023.

[8] William C. Skamarock. Evaluating mesoscale nwp models using kinetic energy spectra. *Monthly Weather Review*, 132(12):3019–3032, 2004.

[9] Robert A. Houze. Mesoscale convective systems. *Reviews of Geophysics*, 42(4), 2004.

---

## Author Response (AR3)

**Replies to all Referee Comments on "Divergent convective outflow in ICON deep convection-permitting and parameterised deep convection simulations. " [1]**

Edward Groot, Patrick Kuntze, Annette Miltenberger, Holger Tost

March 27, 2024

**1 General**

We thank the referees for reading our updated manuscript and formulating the feedback [2]. In this document, the **latest version** of the referee's comments are copied in red, with corresponding responses directly below in black.

**2 Referee 3: reply to RC3 [2]**

*"" I am happy to accept the paper, pending some minor corrections discussed below. My requests for a more "like for like" comparison between the ICON-PER and ICON-PAR simulations, by using identical analysis methods for both, have not been addressed. But I am satisfied that making these changes to the method would be very time-consuming / require rerunning computationally-expensive data-processing, and would be unlikely to significantly affect the results. So I am happy to accept the article without this change. I thank the author for acknowledging / clarifying that the study did not find sensitivity of the precipitation - divergence relationship to the choice of parameterised versus explicit convection; the sensitivity found appears to be dominated by model-resolution alone. This point just ought to be clarified in a couple other places in the paper (I give some suggestions below)."'*

We thank the reviewer for this feedback.
The suggestions of the reviewer are taken into account for the next revision and further addressed in the below.

1. *'"Many thanks for the additional discussion L74-83, and for clarifying in various places that the transient response is studied here. This nicely addresses the query "'*

    We would like to thank the reviewer once more for raising the issue and appreciate that the issue has been resolved in the revised manuscript.

2. *'"My argument does not depend on any horizontal variation in the stability profiles, or variation in stability between different models. Rather it is variation in the vertical structure of the heating profile, and how this correlates with vertical variation of environment stability. E.g. consider two different convective storms occuring in the same environment and static stability profile, and suppose (as is often observed) that the static stability $N^2$ is greater in the lower free troposphere than it is in the upper troposphere. Now, suppose storm 1 is a relatively young cell, so that most of the latent heating occurs in the lower troposphere, whereas storm 2 is a mature system with a substantial stratiform anvil region generating more latent heating in the upper troposphere. Even if the storms are producing equal precipitation rates, storm 2 will produce stronger upper- level divergent outflow, just because the heating is concentrated at upper-levels where $N^2$ is less, so that stronger ascent is needed to make the buoyancy removal by vertical advection ($wN^2$) balance the heating rate ( $\frac{g}{T_v}\frac{\partial T_v}{\partial t}$ ). Even under a horizontally homogeneous environment and latent uniform neutral buoyancy level, storms in different states of organisation may produce differently-shaped vertical profiles of heating rate, which will give different upper-level divergence if $N^2$ varies with height?"' and '"Many thanks for pointing out the interesting experiment in the authors' LES study, where the latent heat of condensation was varied. Note however that the idealised temperature profile used here (from Weisman & Klemp 1982) has arteficially*

*weak variation of $N^2$ with height (potential temperature increases nearly linearly). So this may not have sampled the sensitivity to vertical structure. I don't think the present study requires a detailed investigation into dependence of outflow strength on vertical structure (this might be an interesting topic for future work). But it might be worth breifly mentioning this possibility in the discussion?"'*

We agree that this effect of differences in heating profiles as a function of differential convective organisation affects the local magnitude, and hence sharpness, of the divergence pulse, i.e., outflow at the tropopause (with large $N^2$). This localised pulse will strongly amplify with increased $N^2$, but be mostly locally confined to a thin near-tropopause layer for certain heating profiles. On the other hand, deep outflows of $300 - 400$ hPa or half the tropospheric depth (typically corresponding to substantial positive net heating throughout the troposphere) will typically reduce the sharpness of the near-tropopause peak of divergence and replace it with more gentle outflow at lower levels. Also, both our current work and the LES-simulations of [3] show broadly similar divergence profiles across the ensemble members (and over time), which could be seen as a small limitation.

We agree that the vertical distribution of mass divergence will be strongly affected by the heating profiles and hence by factors such as convective organisation, but consistent with earlier studies, we do not see a substantial variation of the vertically integrated magnitude of the divergence with the heating profile in our results. This is supported by consistency of the LES study and the current study (e.g. Figure 6a and some of the left panels of Figure S5).

Nevertheless, the current study (and the LES-study) does not span a large enough variation in vertical stratification and heating profiles to draw definitive conclusions on this aspect - we just intend to cover a range of deep convection variability and - in this regard - come to the conclusion we have stated.

We thank the reviewer for raising this discussion point and have included some discussion of this point in section 7.3.2 of the revised manuscript.

3. *'"The reviewer's experience with NWP models is that they routinely do produce both gravity-waves and updrafts / downdrafts very near the grid- scale (excessive single-grid-point updrafts are a common problem!) Whilst a model can only simulate these features with significant numerical error, the poor resolvability does not stop them from happening. The idea that models contain a truncation scale of 8 grid-lengths below-which all features are strongly damped has come from large eddy simulation, where this has been deliberately imposed by specifying a suitably large mixing-length parameter in the 3D Smagorinsky turbulence scheme (or by using a highly diffusive numerical method for advection). Imposing such an 8 grid-length truncation scale on an NWP model (with 13km grid-size and parameterised convection) would not be done operationally, as it would degrade the model skill. So I don't agree with the author; features significantly smaller than 100km can and will be simulated by the 13km NWP model, but with diminished accuracy. This is evident in figure 7(c,d); the ensemble standard deviation of divergence in ICON-PAR has a lot of structure on scales of around half a degree (50km), and the individual ensemble members will presumably contain smaller-scale structure that gets smoothed-out when looking at the standard deviation over all members?"'and (now from item number 4!)"'As discussed earlier, this notion is consistent with LES, but is often not the case in NWP models (especially at coarse resolution with parameterised convection, when they typically do not employ a highly diffusive 3D turbulence scheme to enforce the truncation scale). On the other hand, it might be that ICON is more diffusive than other NWP models that the reviewer is familiar with. This is critically dependent on the model's advection scheme and any horizontal diffusion used (e.g. as part of a 3D turbulence scheme"')*

The authors do agree that near-grid features as updrafts, downdrafts and gravity waves do widely occur in storm-resolving simulations. However, the fact that they may - regularly and excessively - occur does not mean that the representation and interactions with nearby features are realistic and never spurious. Furthermore, if we look at spectral analysis, even of storm-resolving simulations, it can be seen that these spectra are still strongly suppressed below a certain wavelength [4], corresponding to about 5-10 grid spacing. Furthermore, representation of wavelike perturbation features smaller than 4 times the grid spacing gets virtually impossible (the perturbations cannot propagate downstream based on simple numerics, but rather will tend to dissipate by nature), so that propagation and scale interactions can only become realistic in the 5-10 grid spacings range.

Nevertheless, we do agree, reduced or sometimes potentially excessive variability (relative to the real atmosphere) at small near-minimal length scales (say waves of about 6-8 times grid spacing) can significantly differ between different models and configurations, depending on details in treatment of advection and turbulence schemes and other means of exchange across scales.

In the end, there is the intrinsic property of a truncation scale, which strictly dissipates (hence: removes) all perturbations that cannot propagate with the flow. As to what we see in our ensemble members, we do agree that perturbations of order 50 km exist in the ensemble means. Given the small

ensemble size in ICON-PAR, this certainly (to some extent) reflects the effective size of features in individual ensemble members (which can just have different amplitudes locally, across the ensemble). In the ICON-PER ensemble there is somewhat more effective smoothing across the ensemble, but we believe that spectral properties of smallest resolved scales (like those in [4]) also affect our simulations.

*'"Many thanks for the changes to the text following this. But there are a few other parts of the paper which still imply that the divergence-precipitation relation depends on the choice of parameterised versus explicit convection, even though no evidence for this has been presented (the sensitivity is dominated by model-resolution alone, as is now stated by the added text at L467). L750-751: maybe change: "feedback from deep convection to its surroundings at larger scales is likely not accurately represented with parameterised deep convection, even if the precipitation climatology is well represented: parameterised ensembles seem to be underdispersive in terms of corresponding dynamical variability at a given precipitation rate" to something like: "feedback from deep convection to its surroundings at larger scales is likely not accurately represented at coarse resolution, even if the precipitation climatology is well represented: ensembles with 13km grid-size seem to be underdispersive in terms of corresponding dynamical variability at a given precipitation rate" Maybe also clarify this in the abstract for consistency, e.g. change L3-4: "Near-linear response of deep convective outflow strength to net latent heating is found for parameterised convection, ..." to e.g. "Near-linear response of deep convective outflow strength to net latent heating is found for 13km grid-spacing with either parameterised or explicit convection, ..."'*

We will carefully reconsider the wording. However, we would also like to point out that causality may seem to be implied to a very quick reader, but it is not really strictly implied by our wording: we do agree about dominance by model resolution, but also want to emphasise that a parameterisation often follows the choice of a certain resolution automatically.

We think that it would not be bad to use the statements as we do for the short and concise abstract, as usually coarse resolution directly correlates with parameterised deep convection. Nothing in the abstract is untrue, we highlight typical configuration A and typical configuration B, without mentioning alternative configuration C yet (i.e. explicit convection at 13 km grid spacing).

In the body of the paper, readers will realise that this is a simplification, and you could possibly try to resolve the convection explicitly at a grid spacing where this is typically not done, but this is rather atypical. In the discussion section, a sentence has been added to (re-)emphasise the results for explicit convection experiments at 13 km resolution in the revised manuscript.

*"Many thanks for this insight! While a significant amount of mesoscale structure (in both the updrafts and the radiated gravity-waves) will be present on the 13km grid, it might be the smaller (km-scale) variability within the wider mesoscale structures that is key? Even though a 50km- long island can be represented on a 13km grid, the bays and inlets along its coast cannot. Refining the grid to 1km may increase the length of the coastline by an order of magnitude. By analogy, the possibility for gravity-waves radiated from within the "island" to collide with eachother from different directions will vastly increase? But perhaps clarify at L462-463 that with "little or no information on geometry of the convective systems can be represented", this is really about geometry of smaller-scale structures within the convective systems (i.e. km-scale, rather than mesoscale)?"*

We highly appreciate that the analogy has clarified the content here. The reviewer is right: we agree that the (heating) structures of individual cells within convective systems will be of high importance. We will use the discussion here to strengthen the content of the manuscript further and utilise the analogy better!

4. *'"No, my argument was that the analysis would be simpler and the manuscript shorter if the low-pass filter was just omitted (it should make little difference as long as the box-averages consider areas much bigger than the filter-scale)."'* and *'"I am happy to take the authors' word that the low-pass filtering does not significantly affect the results anyway. So it is probably not worth rerunning significant parts of the work-flow to pursue this further."'* and

5. *'"This is a reasonable argument; the moving boxes in ICON-PER have been setup manually, repeating this exercise for every ensemble member and sensitivity-test within ICON-PAR would be very laborious, and the effort should be focussed on analysing the more interesting behaviour found in ICON-PER. However, one of the paper's main conclusions is drawn from comparing ICON-PER with ICON-PAR. It would therefore be stronger if more attention was given to ensuring like-for-like comparison and using the same analysis methods when comparing them. e.g. just for the comparison with PAR, the divergence-precip relation in PER could be computed on the same large static boxes as PAR (while still using the smaller moving boxes for the more detailed investigation of relationship to storm structure etc in PER) But again, at this late stage it may not be worth rerunning the work-flow for a consistency-check which is unlikely to affect the results much."'*

We are glad that the reviewer is happy with the analysis as is and shares the thought that the existing workflow is at least in practice beneficial for the study. Furthermore, we thank the reviewer for further elaborating on their thoughts.

6. *'"Many thanks for investigating and correcting this. It is very reassuring that it was essentially only the units shown on the colour-scale (6a) and x-axis (6b) that were wrong before! This has been corrected by removing an erronious factor of air density from the plotted data, so that they are now consistent with the stated units. This increases my confidence in the paper overall, by removing my worry that something else had gone wrong with the analysis."'*
   We are glad that the reviewer is reassured.

7. *'"As discussed earlier it would be possible to ensure like-for-like comparison between PAR and PER, by separately computing the precipitation and divergence in PER over the same large static boxes as were used for PAR. But again it is probably not worth rerunning such a large part of the work-flow at this late stage. I am not overly concerned about the subsidence assumption in the convection parameterisation here. In the reviewer's experience with mass-flux convection schemes, the assumption of "local mass compensation" has little impact, as the model's dynamical core still acts so-as to quickly restore a Weak Temperature Gradient in the horizontal after the parameterised convective heating is applied. This has the effect of immediately undoing the local subsidence imposed by the convection scheme, and redistributing it elsewhere, so that the end result is the same as explicit convection with the equivalent heating profile. The present study's results in figure 8b comparing the divergence response for parameterised versus exlicit convection at 13km grid-size support this."'*
   We agree that having the same convective system in the same model would be the most **optimal** fair comparison and have mentioned a couple of reasons at the top of this reply (and in the previous reply) why this is apparently too challenging and unfeasible in practice for the selected case.
   We are happy that the reviewer shares very little concern about subsidence effects and are glad that the opinion on this has been shared in this thread of replies.

8. *'"Thank you for the edited text at L415, which now clarifies that the quantity shown is the vertical integral divided by outflow depth. Omitting the normalisation by outflow depth would have made the comparison in figure 8 between PER (8a) and PAR (8b) simpler, allowing a slightly more succinct manuscript (the discussion about the correction for differing outflow depths would not be needed). But it does not matter that much. "'*
   We are glad that the referee's opinion closely agrees with ours and would like to thank the reviewer for elaborating on this point.

9. *'"Many thanks for the extensive discussion around the relation between heating, the pressure field, and vertical momentum transport (and the dependence of the "turbulent" transport on model resolution). This made for an interesting and thought-provoking read, and I agree this wider subject is beyond the scope of the present study. I am happy with the slight rewording at L612-613, in the synthesis section. Maybe consider also mentioning the uncertain direction of causality in section 6.3, e.g. at L568-569?"'*
   We are also happy about the exchange on this particular point and thank the reviewer for an active participation in this discussion.
   In our opinion, the direction of causality is something for the synthesis and discussion, so that the focus in Section 6.3 is (as it has been in the latest revision) at the corresponding statistics. We would argue that this is a choice to streamline the content of the paper and keep it focused; any of the two choices could probably be preferred by an arbitrary subset of readers.

10. *'"Many thanks; all the previously listed minor corrections have been addressed. Except: L263: "Ellipse fitting and verification are used to quantity the geometry" should be "Ellipse fitting and verification are used to quantify the geometry?" I also just spotted one more trivial typo: L562: "...while Figure Figure 10b shows..."'*
    Correct; we thank the reviewer for carefully reading the revised manuscript and correct the manuscript accordingly.

**References**

[1] Edward Groot, Patrick Kuntze, Annette Miltenberger, and Holger Tost. Divergent convective outflow in icon deep convection-permitting and parameterised deep convection simulations. [original pre-print and revised version]. *EGUsphere*, 2023:1–32, 2023.

[2] Referee comment 3 on "divergent convective outflow in icon deep convection-permitting and parameterised deep convection simulations.", 2024.

[3] Edward Groot and Holger Tost. Divergent convective outflow in large-eddy simulations. *Atmospheric Chemistry and Physics*, 23:6065–6081, 2023.

[4] F. Judt. Atmospheric predictability of the tropics, middle latitudes, and polar regions explored through global storm-resolving simulations. *Journal of the Atmospheric Sciences*, 77(1):257 – 276, 2020.